# CANDOR: Counterfactual ANnotated DOubly Robust Off-Policy Evaluation

## Abstract

When applying contextual bandit algorithms in high-stakes settings (e.g., medical treatment), practitioners rely on off-policy evaluation (OPE) methods that use historical data to evaluate the behavior of novel policies prior to deployment. Unfortunately, OPE techniques are inherently limited by the breadth of the available data, which may not reflect distribution shifts resulting from the application of a new policy. Recent work attempts to address this challenge by leveraging domain experts to increase dataset coverage by annotating counterfactual samples. However, such annotations are not guaranteed to be free of errors, and incorporating imperfect annotations can lead to worse policy value estimates than not using the annotations at all. To make use of imperfect annotations, we propose a family of OPE estimators based on the doubly robust (DR) principle, which combines importance sampling (IS) with a reward model (direct method, DM) for better statistical guarantees. We introduce three opportunities within the DR estimation framework to incorporate counterfactual annotations. Under mild assumptions, we prove that using annotations within just the DM component yields the most desirable results, providing an unbiased estimator even under noisy annotations. We validate our approaches in several settings, including a real-world medical domain, observing that the theoretical advantages of using annotations within just the DM component hold in practice under realistic conditions. By addressing the challenges posed by imperfect annotations, this work broadens the applicability of OPE methods and facilitates safer and more effective deployment of decision-making systems.

## 1 Introduction

Contextual bandit methods have been successfully applied to learn optimal decision-making policies across several domains, including healthcare (Yao et al., 2021), recommendation systems (Li et al., 2010), and education (Lan & Baraniuk, 2016). In high-stakes decision-making scenarios, such as designing patient treatment policies in clinical settings, it is critical for practitioners to assess the performance of a new policy prior to deployment. To do so, standard practice consists of applying off-policy evaluation (OPE) methods (Sutton & Barto (2018), Chapter 5), which estimate the value of a new (target) policy using a behavior dataset collected from a different policy. By facilitating policy evaluations without risky real-world experiments, OPE methods represent a crucial tool for safe policy deployment.

However, OPE is inherently limited by the quality and coverage of the behavior dataset. For instance, the current treatment policy in a hospital may have never recommended a recently developed drug, so no OPE method can reliably evaluate a policy that recommends this drug as a treatment. To address this issue, Tang & Wiens (2023) proposed an importance sampling (IS)-based OPE estimator called C-IS (referred to in this work as IS$^+$), in which experts provide annotations (i.e., predicted rewards) for counterfactual actions of samples observed in the behavior dataset. However, their approach relied on the **strong assumption that annotations are free of errors**. Realistically, even expert-generated annotations are prone to imperfections. Determining the optimal way to incorporate potentially imperfect counterfactual annotations into an OPE estimator remains an open challenge.

To address this challenge, we propose a family of OPE estimators based on the doubly robust (DR) principle (Cassel et al., 1976). Compared to IS estimators, DR estimators offer provable reductions in variance while remaining unbiased. It is not immediately obvious how to use the additional data coverage provided by potentially imperfect counterfactual annotations while retaining the desirable properties of DR estimators. In this work, we introduce three ways of modifying DR estimators to include counterfactual annotations, each of which impacts the estimator performance in a different way. Through a rigorous analysis of the bias-variance trade-off of each approach, **in the face of imperfect annotations**, we identify one estimator that successfully leverages information from counterfactual annotations to improve coverage without compounding error in those annotations. In contrast, the other two estimators compound error proportionally to the annotation error, resulting in worse policy estimates than ignoring the annotations altogether.

In summary, our contributions are the following:

- **We propose a family of OPE estimators** inspired by the DR principle that incorporate counterfactual annotations while accounting for potential errors in the annotations. We perform a thorough theoretical analysis of our proposed estimators, finding that how annotations are incorporated into the estimator has a substantial impact on the estimator's performance (Section 3).
- **We evaluate our estimators on three synthetic contextual bandit environments and a real medical dataset.** We use the synthetic settings to empirically verify our theoretical insights, and use the medical domain to demonstrate the potential utility of our proposed approaches in high-stakes problems (Section 4).
- **We provide practical considerations for choosing the best OPE estimator** in the presence of imperfect counterfactual annotations, which, to our knowledge, is currently missing from the OPE literature. This systematic guide further facilitates the deployment of contextual bandit policies in high-stakes settings (Section 5).

## 2 Background

We consider a contextual bandit setting defined by $(\mathcal{S}, \mathcal{A}, R, d_0)$, where $\mathcal{S}$ is the discrete context space, $\mathcal{A}$ is the discrete action space, $R : \mathcal{S} \times \mathcal{A} \to \Delta(\mathbb{R})$ is the reward function, and $d_0$ is the initial context distribution. Given a behavior dataset $D = \{(s_i, a_i, r_i)\}_{i=1}^N$ generated from a behavior policy $\pi_b$, we aim to evaluate a different target policy $\pi_e$ by estimating its value $v(\pi_e) = \mathbb{E}_{s \sim d_0, a \sim \pi_e(\cdot|s), r \sim R(s,a)}[r]$.

### 2.1 Off-Policy Evaluation

We give an overview of three common types of OPE approaches in the context of contextual bandit. Importance sampling (IS), $\hat{V}^{\text{IS}} = \frac{1}{N} \sum_{i=1}^N \rho_{s_i}(a_i) r_i$, assigns an inverse propensity score (IPS), $\rho_s(a) = \frac{\pi_e(a|s)}{\pi_b(a|s)}$, to each sample $(s_i, a_i, r_i)$ in the behavior dataset (Horvitz & Thompson, 1952; Precup et al., 2000). Similar to prior work, we assume that the IPS ratio $\rho$ is known (Farajtabar et al., 2018; Thomas & Brunskill, 2016). IS results in an unbiased estimate of the value of the target policy, $v(\pi_e)$, when $\pi_e$ is well supported by the behavior dataset (Precup et al., 2000), i.e., has sufficient "coverage" (Assumption 1).

**Assumption 1** (Common support). $\pi_e(a|s) > 0 \to \pi_b(a|s) > 0$.

The variance of the IS estimator is (Tang & Wiens, 2023)

$$N \cdot \mathbb{V}[\hat{V}^{\text{IS}}] = \mathbb{V}_{s \sim d_0}[v^{\pi_e}(s)] + \mathbb{E}_{s \sim d_0}[\mathbb{V}_{a \sim \pi_b(\cdot|s)}[\rho_s(a)\bar{R}(s,a)]] \\ + \mathbb{E}_{s \sim d_0}[\mathbb{E}_{a \sim \pi_b(\cdot|s)}[\rho_s(a)^2 \sigma_R(s,a)^2]].$$

where $\bar{R}(s,a) = \mathbb{E}[R(s,a)]$ and $\sigma_R(s,a)^2 = \mathbb{V}[R(s,a)]$ are the mean and variance of the reward distribution, respectively.

Another approach to OPE is the direct method (DM) (Li et al., 2010; Beygelzimer & Langford, 2009; van Seijen et al., 2009; Harutyunyan et al., 2016; Le et al., 2019; Voloshin et al., 2021). DM first uses the behavior dataset to estimate a reward model, $\hat{R} : \mathcal{S} \times \mathcal{A} \to \mathbb{R}$, to predict the mean reward, and then uses $\hat{R}$ to directly compute the target policy value as $\hat{V}^{\text{DM}} = \sum_s d_0(s) \sum_a \pi_e(a|s)\hat{R}(s,a)$. $\hat{R}$ can vary in complexity, ranging

from regression models to neural networks. If the reward model is fully realizable and there is full coverage in the behavior dataset, then DM has zero bias and favorable variance in its estimate of the target policy value. Typically, DM estimators have a lower variance than IS (Dudik et al., 2011) when the size of the behavior dataset is sufficiently large to learn an accurate reward model.

The last category of OPE approaches consists of doubly robust (DR) methods (Dudik et al., 2011; Dudík et al., 2014; Farajtabar et al., 2018; Jiang & Li, 2016). These methods are termed "doubly robust" because they maintain strong theoretical guarantees even when either the IPS ratio $\rho$, or the estimated reward function $\hat{R}$, is inaccurate. As such, the DR estimator is robust to two sources of error (the IPS ratio and the reward model). A related line of literature is in one-step estimators, which strongly resemble DR estimators (Kennedy, 2023). The asymptotic properties of these estimators is studied in semiparametric efficient statistics and efficient influence functions. The standard DR estimator is

$$\hat{V}^{\mathrm{DR}} = \frac{1}{N} \sum_{i=1}^{N} \underbrace{\hat{R}(s_i, \pi_e)}_{\text{DM part}} + \underbrace{\rho_{s_i}(a_i)(r_i - \hat{R}(s_i, a_i))}_{\text{IS part}}, \tag{1}$$

where $\hat{R}(s, \pi_e) = \sum_{a \in \mathcal{A}} \pi_e(a|s)\hat{R}(s, a)$ is the estimated value of state $s$ under the target policy $\pi_e$ using the reward model $\hat{R}$. We refer to the first and second term in Equation (1) as the *DM part* and the *IS part*, respectively. Under standard coverage assumptions (Assumption 1), the DR estimator produces an unbiased estimate of $v(\pi_e)$ under standard sample splitting assumptions (e.g., the reward model and OPE estimate are learned using independent splits of the data). DR methods also see a reduction in variance in comparison to IS-based methods; the variance can be written as

$$N \cdot \mathbb{V}[\hat{V}^{\mathrm{DR}}] = \mathbb{V}_{s \sim d_0}[v^{\pi_e}(s)] + \mathbb{E}_{s \sim d_0} \left[ \mathbb{V}_{a \sim \pi_b(\cdot|s)} \left[ \rho_s(a)(\bar{R}(s, a) - \hat{R}(s, a)) \right] \right]$$
$$+ \mathbb{E}_{s \sim d_0} \left[ \mathbb{E}_{a \sim \pi_b(\cdot|s)} \left[ \rho_s(a)^2 \sigma_R(s, a)^2 \right] \right].$$

The reduction in variance relative to the IS estimator rests in the second term, in which $\rho$ is scaled by $\bar{R}(s, a) - \hat{R}(s, a)$ instead of $\bar{R}(s, a)$, which is close to 0 if the estimated reward model $\hat{R}$ is accurate.

## 2.2 Counterfactual Annotations

In our work, we consider incorporating counterfactual annotations to increase data coverage. Suppose that we are given a behavior dataset of size $N$, $D = \{(s_i, a_i, r_i)\}_{i=1}^{N}$. Each factual sample $(s_i, a_i)$ in the behavior dataset is associated with a set of counterfactual annotations $\mathbf{g}_i = \{g_i^{\tilde{a}} \mid \tilde{a} \in \mathcal{A} \setminus \{a_i\}\}$. Note that $\mathbf{g}_i$ may be empty. We assume that the annotation of the counterfactual action $\tilde{a}$ is drawn from some distribution $G : \mathcal{S} \times \mathcal{A} \to \Delta(\mathbb{R})$, $g_i^{\tilde{a}} \sim G(s_i, \tilde{a})$. We assume that there are a total of $M$ counterfactual annotations. In practice, we expect to collect a small subset of all possible counterfactual annotations because they may be expensive to obtain. We refer to the dataset that combines factual samples and counterfactual annotations as the counterfactual-annotated dataset and denote it by $D^+$. A simple example of a counterfactual-annotated dataset with two contexts and two actions is visualized in Figure 1. In this example, we observe two factual samples and only one of them has a counterfactual annotation.

In Section 3.2, we discuss three scenarios for the function $G$ (perfect, biased, or noisy annotations). For simplicity, we use $c_i^a$ to refer to either the reward or the counterfactual annotation of the factual sample $(s_i, a_i)$, i.e., $c_i^a = r_i$ when $a = a_i$ and $c_i^a = g_i^a$ when $a \neq a_i$.

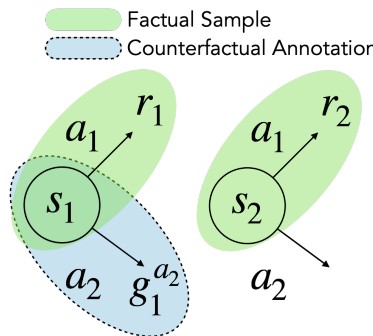

Figure 1: **Counterfactual-annotated dataset with two contexts and two actions.** There are two factual samples, $(s_1, a_1, r_1)$ and $(s_2, a_1, r_2)$. For the first (left) factual sample, we have a corresponding counterfactual annotation $(s_1, a_2, g_1^{a_2})$. For the second (right), the annotation is missing.

### 2.3 The IS$^+$ Estimator

To incorporate counterfactual annotation, Tang & Wiens (2023) introduced IS$^+$, defined as

$$\hat{V}^{\text{IS}^+} = \frac{1}{N} \sum_{i=1}^{N} \sum_{a \in \mathcal{A}} w_i^a \rho_{s_i}^+(a) c_i^a,$$

where $\{w_i^a\}$ is a set of user-defined weights for the $i$-th factual sample $(s_i, a_i)$ and its associated counterfactual annotations. This method requires that $\sum_{a \in \mathcal{A}} w_i^a = 1$ to ensure that IS$^+$ is a convex combination of factual and counterfactual samples. We set $w_i^{\tilde{a}} = 0$ if the annotation of the counterfactual action $\tilde{a}$ for the $i'th$ state is not available. $\rho_s^+(a) = \frac{\pi_e(a|s)}{\pi_b^+(a|s)}$ is the augmented IPS ratio, where the augmented behavior policy $\pi_b^+(a|s)$ is defined as

$$\pi_b^+(a|s) = \bar{W}(a|s, a)\pi_b(a|s) + \sum_{\check{a} \in \mathcal{A} \setminus \{a\}} \bar{W}(a|s, \check{a})\pi_b(\check{a}|s),$$

and $\bar{W}(\tilde{a}|s, a) = \mathbb{E}[w^{\tilde{a}}]$ is the average weight of action $\tilde{a}$ for the factual context-action pair $(s, a)$. The weights and augmented IPS ratios are critical in the construction of IS$^+$, as they ensure the context distribution in the counterfactual-annotated dataset $D^+$ remains identical to the context distribution in the original factual dataset $D$. This is because the weights $\{w_i^a\}$ associated with the factual and counterfactual actions for each sample $s_i$ must sum to 1; otherwise, the estimator is no longer unbiased.

## 3 Methods

When the behavior dataset has limited coverage (which tends to be true in practice), IS estimators are known to have high variance (Jiang & Li, 2016). In contrast, DM estimators have high bias when the reward model is misspecified. Thus, we explore how to introduce counterfactual annotations into a DR estimator, which retains beneficial theoretical properties even when the reward model is misspecified–a situation that frequently occurs in practice. While DR is well-understood in a setting with only factual samples, we aim to incorporate counterfactual annotations such that we can overcome the limitations of the coverage of the behavior dataset. The most naive approach is to directly use the counterfactual-annotated dataset $D^+$ in a standard DR estimator, viewing the counterfactual annotations as additional samples. However, as we discuss in Appendix C, this approach can produce arbitrarily biased estimates of $v(\pi_e)$ depending on the number of annotations used, because it alters the context distribution of the behavior dataset, regardless of annotation quality. As a result, we focus on developing new estimators that build on the DR principle. Below, we present three new estimators along with a rigorous theoretical analysis of their bias and variance properties in the presence of imperfect annotations.

### 3.1 Proposed DR Estimators with Counterfactual Annotations

The standard DR estimator, as shown in Equation (1), can be broken down into two components: the direct method (DM) part and the importance sampling (IS) part. We observe that counterfactual annotations can be independently leveraged in either of these components. Based on this insight, we propose **three** new DR-inspired estimators leveraging counterfactual annotations. First, **DM$^+$-IS** (Equation (2)) uses the counterfactual-annotated dataset to estimate the reward model and combines it with standard IS. Next, **DM-IS$^+$** (Equation (3)) uses counterfactual annotations to augment the IS part (as in IS$^+$) and combines it with a standard DM estimator. Finally, **DM$^+$-IS$^+$** (Equation (4)) uses counterfactual annotations in both the DM and IS parts.

$$\hat{V}^{\text{DM}^+\text{-IS}} = \frac{1}{N} \sum_{i=1}^{N} \left( \hat{R}^+(s_i, \pi_e) + \rho_{s_i}(a_i)(r_i - \hat{R}^+(s_i, a_i)) \right) \tag{2}$$

$$\hat{V}^{\text{DM-IS}^+} = \frac{1}{N} \sum_{i=1}^{N} \left( \hat{R}(s_i, \pi_e) + \sum_{a \in A} w_i^a \rho_{s_i}^+(a)(c_i^a - \hat{R}(s_i, a)) \right) \tag{3}$$

$$\hat{V}^{\text{DM}^+\text{-IS}^+} = \frac{1}{N} \sum_{i=1}^{N} \left( \hat{R}^+(s_i, \pi_e) + \sum_{a \in A} w_i^a \rho_{s_i}^+(a)(c_i^a - \hat{R}^+(s_i, a)) \right). \tag{4}$$

Here, $\hat{R}^+$ is the reward function estimate learned using the counterfactual-annotated dataset $D^+$ (see further discussion in appendix E).

### 3.2 Theoretical Analyses under Imperfect Annotations

Now, we examine the performance of our proposed estimators in the presence of imperfect annotations, offering insights into how these limitations affect the estimators. This analysis also provides theoretical support for our guidance on selecting a robust OPE estimator, which we discuss further in Section 5. In our problem setting, there are three possible sources of error: incorrect estimates of the behavior policy $\pi_b$, a misspecified reward model, and imperfect (biased or noisy) annotations. Like prior work (Farajtabar et al., 2018; Thomas & Brunskill, 2016) we assume that the IPS ratio $\rho$ is known, and instead focus on identifying an OPE estimator that is robust to the last two sources of error. These theoretical results inform our hypotheses and help ensure that our empirical findings align with the expected behavior (Section 4.2).

The novelty of our theoretical results are two-fold. First, prior work on DR estimators provided expectation and variance derivations assuming a prespecified error term in $\hat{R}$ (e.g., (Dudik et al., 2011) assumed that $\hat{R}(s,a) = \bar{R}(s,a) + \epsilon(s,a)$). In contrast, our analysis accounts for the stochasticity in $\hat{R}$ arising from the dataset used to fit the reward model, since our proposed approaches explicitly modify what data is used to fit the reward model. We assume that the reward model is estimated from a separate dataset, which we refer to as $D_{\hat{R}}$ or $D_{\hat{R}^+}$, depending on if counterfactual annotations are incorporated. We assume that $D_{\hat{R}}$ and $D_{\hat{R}^+}$ are drawn from the same data distributions as $D$ and $D^+$, respectively. Second, we derive the bias and variance of our proposed DR estimators under imperfect annotations, which is arguably more realistic in practice. We do not make assumptions about the reward model class being well specified. We use the following three assumptions to quantify the quality of counterfactual annotations.

**Assumption 2** (Perfect annotations). $\mathbb{E}_{g^a \sim G(s,a)}[g^a] = \bar{R}(s, a)$, $\mathbb{V}_{g^a \sim G(s,a)}[g^a] = \sigma_R^2(s, a)$.

**Assumption 3** (Biased annotations). $\mathbb{E}_{g^a \sim G(s,a)}[g^a] = \bar{R}(s, a) + \epsilon_G(s, a)$, $\epsilon_G(s, a) \neq 0$.

**Assumption 4** (Noisy annotations). $\mathbb{V}_{g^a \sim G(s,a)}[g^a] = \sigma_R(s, a)^2 + \Delta_G(s, a)$, $\Delta_G(s, a) > 0$.

Assumption 3 and Assumption 4 are used to study the effect of biased and noisy (i.e., higher variance) annotations. The additional bias and noise are captured in the terms $\epsilon_G$ and $\Delta_G$, respectively. Assumption 2 is contrary to Assumptions 3 and 4, and as such never invoked together. Similar to Tang & Wiens (2023), we use the following assumption on dataset support, which is a relaxed version of Assumption 1.

**Assumption 5** (Common support with annotations). $\pi_e(a|s) > 0 \rightarrow \pi_b^+(a|s) > 0$.

First, we show that, with perfect annotations (Assumption 2) and appropriate coverage assumptions (Assumption 1 or 5), all three proposed estimators are unbiased (Propositions 12, 14 and 17 in Appendix G). Additionally, when all counterfactual actions are annotated and $w^a = 1/|\mathcal{A}|$, DM-IS$^+$ and DM$^+$-IS$^+$ are both equivalent to IS$^+$(Corollaries 20 and 21 in Appendix H).

Now, we derive the bias of the proposed estimators when Assumption 2 (perfect annotations) is violated. These derivations only rely on Assumption 3 (annotation bias) but not Assumption 4 (annotation variance).

**Proposition 1** (Unbiasedness of DM$^+$-IS under imperfect annotations). *Under biased annotations (Assumption 3) and common support (Assumption 1),* $\mathbb{E}[\hat{V}^{\text{DM}^+\text{-IS}}] = v(\pi_e)$.

**Theorem 2** (Bias of DM-IS$^+$ and DM$^+$-IS$^+$ under imperfect annotations). *Under biased annotations (Assumption 3) and common support (Assumption 5), the two estimators have the same expectation:*

$$\mathbb{E}[\hat{V}^{\text{DM-IS}^+}] = \mathbb{E}[\hat{V}^{\text{DM}^+\text{-IS}^+}] = v(\pi_e) + \mathbb{E}_{\substack{s\sim d_0 \\ a\sim\pi_e(s)}}\left[\left(1 - \frac{\bar{W}(a|s,a)\pi_b(a|s)}{\pi_b^+(a|s)}\right)\epsilon_G(s,a)\right]. \tag{5}$$

*Remark.* Proposition 1 establishes that, with biased annotations, DM$^+$-IS is an unbiased estimator of the target policy value $v(\pi_e)$. In contrast, Theorem 2 shows that both DM-IS$^+$ and DM$^+$-IS$^+$ will produce biased estimates of $v(\pi_e)$. Note that the last term in Equation (5) is identical to the expectation derivation for IS$^+$ (Tang & Wiens, 2023).

**Theorem 3** (Variance of DM$^+$-IS under imperfect annotations). *Under Assumptions 1, 3 and 4,*

$$N \cdot \mathbb{V}[\hat{V}^{\text{DM}^+\text{-IS}}] = \mathbb{V}_{s\sim d_0}[v^{\pi_e}(s)] + \mathbb{E}_{s\sim d_0}\mathbb{E}_{a\sim\pi_b(s)}[\rho_s(a)^2\sigma_R^2(s,a)]$$
$$+ \mathbb{E}_{s\sim d_0}\mathbb{E}_{a\sim\pi_b}\left[\left(\rho_s(a)^2 - \tfrac{1}{\pi_b(a|s)}\right)\Delta_{\hat{R}^+}(s,a)\right] + \mathbb{E}_{s\sim d_0}\left[\mathbb{E}_{a\sim\pi_b}[\rho_s(a)^2\varepsilon_{\hat{R}^+}(s,a)^2] - \varepsilon_{\hat{R}^+}^{\pi_e}(s)^2\right],$$

*where* $\Delta_{\hat{R}^+}(s,a) = \mathbb{V}_{D_{\hat{R}^+}}[\hat{R}^+(s,a)]$, $\varepsilon_{\hat{R}^+}(s,a) = \mathbb{E}_{D_{\hat{R}^+}}[\hat{R}^+(s,a)] - \bar{R}(s,a)$, *and* $\varepsilon_{\hat{R}^+}^{\pi_e}(s) = \mathbb{E}_{a\sim\pi_e}[\varepsilon_{\hat{R}^+}(s,a)]$.

*Remark.* Theorem 3 characterizes the variance of DM$^+$-IS under biased and noisy counterfactual annotations. The first two terms of the variance remain identical to those derived under perfect annotations (see Appendix Proposition 13). However, the third term (highlighted in purple), which depends on $\hat{R}^+$, can be dominant when noisy annotations introduce additional variance in the estimate of the reward model. The last term highlighted in green emerges from the possible estimation error of the reward model due to imperfect annotations.

We derive the variance of DM-IS$^+$ and DM$^+$-IS$^+$ under imperfect annotations in Proposition 16 and Proposition 19 respectively. Despite these derivations, we find it difficult to compare the variance of the three estimators under imperfect annotations theoretically. In particular, the comparison relies on knowing properties of certain terms such as the sign and magnitude of annotation error, which are not testable in practice. As such, we compare the three estimators empirically.

We summarize our theorems in Appendix Table 3 with full proofs provided in Appendix G. In short, under perfect annotations, all of our proposed DR estimators are unbiased. Under imperfect annotations, DM$^+$-IS$^+$ and DM-IS$^+$ share the same bias, while DM$^+$-IS remains unbiased. We expect imperfect annotations to increase the variance of all three proposed estimators due to the increased bias and variance of the reward function estimate.

## 4 Experiments

We now empirically evaluate the performance of our proposed estimators, focusing on settings with imperfect annotations and a misspecified reward model, which reflect real-world scenarios. Prior work demonstrated that counterfactual annotations can improve the variance of an IS-based OPE estimator, suggesting that incorporating these annotations into both the IS and DM part of the DR estimator should lead to an even larger reduction in the estimator error (Tang & Wiens, 2023). However, our findings demonstrate that the improvement in estimator error depends both on the quality of the annotations as well as how they are incorporated into the estimator.

Our experiments seek to answer the following questions: **1)** How do imperfect annotations empirically affect the proposed OPE methods? **2)** How do our proposed methods perform with compounding errors from imperfect annotations and a misspecified reward model?

### 4.1 Experimental Setup

To answer these questions, we investigated three synthetic settings with progressively increasing state and action space sizes, and one real-world medical domain. Key characteristics of these domains are summarized in Appendix Table 1 with further details in Appendix B.

### 4.1.1 Synthetic Domains

**Two Context Bandit** (Tang & Wiens, 2023): This setting is visualized in Figure 1 and has two contexts, and two actions. Without loss of generality, the reward of taking either action from the first context is sampled from a normal distribution and set to 0 for the second context.

**Heartsteps** (Mandyam et al., 2024): This realistic mobile health simulator models the user's physical activities given mobile interventions based on the Heartsteps study (Klasnja et al., 2019). The context is a 3-dimensional vector that includes a treatment effect term and the step count of the previous day. There are two actions (either *send an intervention* or *do nothing*) at each decision time, and the reward is drawn from a normal distribution with the mean being the square root of the user's observed step count.

**Sepsis** (Oberst & Sontag, 2019): In this setting, we adapt the sepsis simulator in (Oberst & Sontag, 2019), which is originally built for a Markov Decision Process (MDP) setting, to a contextual bandit setting by interacting with the environment for only one step. The patient context is an 8-dimensional vector that contains information about vitals and ongoing treatments. There are 8 treatment options, and the reward is an indicator function of whether the patient is under treatment and has stable vitals.

To produce perfect counterfactual annotations of state $s$ and counterfactual action $\tilde{a}$, we sample from the true reward model, i.e. $G(s, \tilde{a}) = \mathcal{N}(\bar{R}(s, \tilde{a}), \sigma_R(s, \tilde{a}))$. To produce biased and noisy counterfactual annotation, we sample from $\mathcal{N}(\bar{R}(s, \tilde{a}) + \epsilon_G(s, a), \sigma_R(s, \tilde{a}) + \Delta_G(s, a))$, where $\epsilon_G$ and $\Delta_G$ refer to the additional bias and variance that compromise the quality of the annotations.

In addition to imperfect annotations, we study the compounding error of misspecified reward models. A misspecified reward model cannot perfectly capture the environment's true reward function, regardless of the training data size. Such misspecification is common in practice. For instance, in a clinical setting, the true reward model guiding a clinician's treatment decisions is often unknown and approximated by a simpler model (e.g., maintaining the patient's vitals within a safe range). In our experiments, we create misspecified reward models across all three synthetic settings by either partially observing the state or altering the state representation (Table 1 and Appendix B).

For these three synthetic domains, we consider various combinations of stochastic behavior and target policies (details in Appendix B). Specifically, the behavior policies vary in their coverage of the action space. We present the averaged results across these combinations. We calculate the value of the target policy using Monte Carlo estimates, and report the root mean squared error (RMSE) of estimated policy values.

### 4.1.2 Real-World Clinical Data

**MIMIC-IV, Potassium Administration** (Johnson et al., 2020; Goldberger et al., 2000): MIMIC-IV contains electronic health records for over 65,000 admitted patients. In this domain, we study a subset of patients from MIMIC-IV that received potassium repletion through an intravenous line. Potassium repletion is a common task in critical care settings; imbalanced potassium levels can have severe side effects including cardiac arrest (Prasad et al., 2022). We created two splits of the dataset based on whether a patient has renal disease (we refer to these splits as "renal" and "non-renal"). The behavior policy is the clinician's treatment policy for the "non-renal" patients and the target policy is the clinician's policy for "renal" patients. In Appendix Figure 14, we see that patients with renal disease are given lower dosages to account for their impaired kidney function (Shrimanker & Bhattarai, 2025). Our goal is to estimate the value of the target policy using data from the behavior policy.

**Domain Setup**: The patient context is a 20-dimensional vector containing information about vitals, administered medications, and static covariates; the actions are five possible dosages of potassium; and the reward is an indicator function of whether the patient's lab potassium value is within the reference range 2 hours after administering a given dosage. Distinct from the synthetic settings, $\pi_b$ and $\pi_e$ are not given and are instead estimated using behavior cloning. We use linear regression to fit our estimated reward model. We measure estimators' performance using RMSE.

**Counterfactual Annotations**: We randomly selected a subset of state-action pairs in the behavior data split and generated counterfactual annotations for those samples. The annotations are produced using OpenAI

"o1" (OpenAI et al., 2024), which is prompted to predict a patient's blood potassium level after administering a dosage that is different from what that patient actually received in the behavior dataset. This procedure mimics a setting where counterfactual annotations may be imperfect. Further details regarding the dataset and annotation construction are in Appendix B.

### 4.1.3 Baselines

We compare our proposed estimators to standard OPE estimators that do not use counterfactual annotations (IS, DM, and DR) and IS$^+$, which uses counterfactual annotations. We also compare to a direct method estimator that estimates the reward model using the counterfactual-annotated dataset, defined as $\hat{V}^{\text{DM}^+} = \sum_s d_0(s) \sum_a \pi_e(a|s) \hat{R}^+(s, a)$.

### 4.2 Results

#### 4.2.1 Imperfect annotations and well-specified reward models

**Biased annotations affect the RMSE of OPE estimators more than higher variance annotations.** First, we examine the impact of biased and noisy annotations under a well-specified reward model. Focusing on the two context bandit setting, we demonstrate that biased annotations have a greater effect on the RMSE of the proposed estimators than noisy annotations (Figure 2). Error metrics for this setting are provided in Appendix A.1. While Figure 2 reports only the proposed OPE methods, this trend holds across all OPE methods that use counterfactual annotations, across all synthetic environments. The effect is most pronounced in methods that incorporate annotations in the IS component (IS$^+$, DM-IS$^+$, DM$^+$-IS$^+$).

This observation aligns with our theoretical results (Proposition 1, Theorem 2), which suggest that estimators using IS$^+$ are biased with biased annotations. The RMSE of DM$^+$-IS remains far more stable across biased and noisy annotations, and is in general more invariant to annotation quality than baselines. From Theorem 3, we know that noisier annotations increase the RMSE of DM$^+$-IS; empirically, we find that this increase is not substantial.

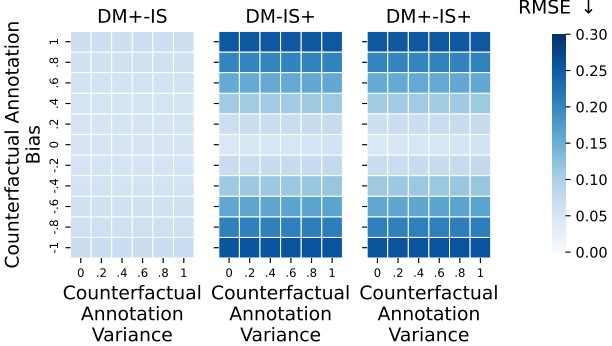

Figure 2: **Heatmaps of mean RMSE with a well-specified reward model on Two-Context Bandit (lower RMSE is represented lighter):** The bias of the counterfactual annotations has a larger impact on RMSE than the variance. The $x, y$-axis represents the variance ($\Delta_G$) and the bias ($\epsilon_G$) of the annotations, respectively. The RMSE hardly varies across the x-axis, but increases proportionally to the magnitude of the annotation bias. This trend is particularly noticeable in DM-IS$^+$ and DM$^+$-IS$^+$. The RMSE of DM$^+$-IS is far more consistent regardless of the annotation bias and variance.

#### 4.2.2 Imperfect annotations and misspecified reward models

In many realistic settings, such as those involving clinical data, we are likely to have both a misspecified reward model and imperfect annotations. **Our results demonstrate that, across all synthetic datasets, DM$^+$-IS is most robust to these two sources of error.** Intuition suggests that, under a misspecified reward model, DM and DM$^+$ will suffer, since a misspecified reward violates the accurate model assumption (Dudik et al., 2011). However, a misspecified reward model should not substantially affect any of the approaches that uses the DR principle, because these estimators can rely on the their IS component to still produce favorable results.

In Section 4.2.1, we noted that the annotation variance does not highly impact the RMSE of the proposed estimators. Thus, we focus on the effect of biased annotations in this set of results. We report mean RMSE across all three synthetic environments with varying degrees of annotation bias. Error metrics are available in Appendix A.2. Our results show that DM$^+$-IS is most resilient to the compounding errors from both

imperfect annotations and a misspecified reward model (Figure 3), showing the lowest RMSE across all magnitudes of annotation bias. Across all synthetic domains, we see that DM$^+$-IS consistently has either the lowest RMSE, or performs comparably to the best performing method. We hypothesize that this is because DM$^+$-IS is the only proposed estimator that is unbiased in the presence of imperfect annotations, and does not suffer from a misspecified reward model due to its DR properties.

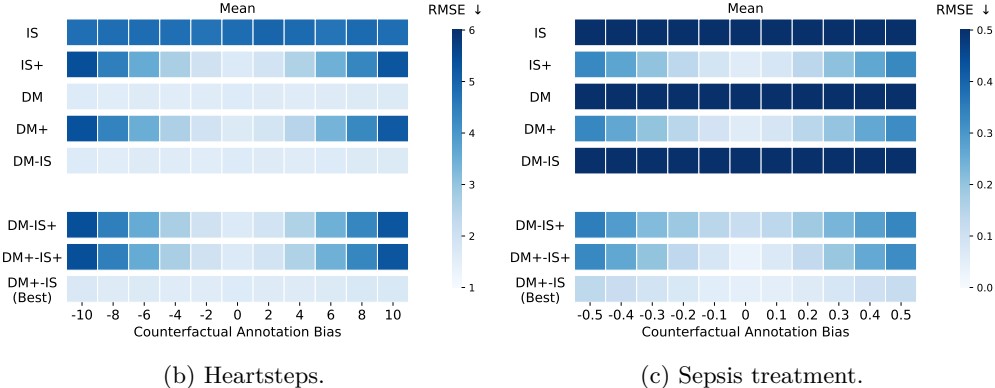

(b) Heartsteps.                    (c) Sepsis treatment.

Figure 3: **Heatmaps of mean RMSE with a misspecified reward model and imperfect annotations (lower RMSE is represented lighter):** The $x$-axis represents annotation bias, $\epsilon_G$. Across all datasets (two context results reported in Appendix A), DM$^+$-IS performs either better than all baselines, or comparably to the best-performing baseline. Among all methods that use counterfactual annotations, DM$^+$-IS is most robust to biased annotations and a misspecified reward model. In comparison to baselines that do not use counterfactual annotations, DM$^+$-IS frequently produces a lower RMSE.

### 4.3 DM$^+$-IS outperforms baselines on MIMIC-IV data

Finally, we evaluate our methods using offline data from MIMIC-IV. With 100 counterfactual annotations, we find that DM$^+$-IS outperforms all baselines (Figure 4a). Notably, IS$^+$ exhibits the highest error, suggesting that the counterfactual annotations may be imperfect. The relatively small difference in performance between DM$^+$ and DM$^+$-IS implies that the estimated reward model is reasonably accurate. We also examine how the performance of key OPE estimators varies as the number of counterfactual annotations increases (Figure 4b). While the RMSE of DM-IS$^+$ and DM$^+$-IS$^+$ remains high with additional annotations—indicating that the incorporation of imperfect annotations introduces bias—the RMSE of DM$^+$-IS initially decreases and then plateaus. These trends are consistent with our observations in the synthetic experiments (Figures 2 and 3).

## 5 Selecting an OPE Estimator

As discussed in Section 4.2, we find the choice of OPE estimator depends most on (1) whether the reward model is misspecified, and (2) the bias of the counterfactual annotations. In the case that the reward model and annotation quality are known, our recommendations are summarized in Figure 5. The empirical results reported in the main text focus on the misspecified reward model and imperfect annotations case; we report results supporting the other three settings in Appendix A.1 and Appendix A.2. However, in the vast majority of real-world settings, the reward model and annotation quality are unknown a priori. In these settings, we recommend using DM$^+$-IS, which is most robust to the compounding errors from imperfect annotations and a misspecified reward model (Figure 3). Particularly, any further use of imperfect annotations (such as in the IS part), can lead to larger compounding errors.

To further emphasize the utility of DM$^+$-IS, we explore the consequences of choosing DM$^+$-IS when both the annotation and reward model quality is unknown in the sepsis treatment environment (Appendix Figure 9). Our results indicate that choosing DM$^+$-IS regardless of annotation or reward model quality will provide OPE estimates that are within a small margin of the best possible OPE method. The best performing OPE method is either DM or DM$^+$ (according to Figure 5), both with a well-specified reward model. $\Delta$, the difference between the DM$^+$-IS estimate and the DM or DM$^+$ estimate (depending on annotation quality) is

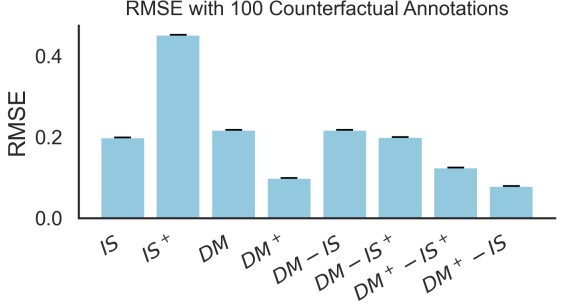
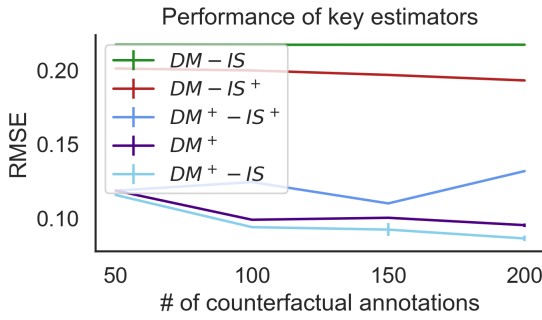

(a) DM$^+$-IS outperforms all estimators with 100 counterfactual annotations. Error bars represent 95% confidence intervals, and DM$^+$-IS outperforms baselines with no overlapping intervals.

(b) As the number of counterfactual annotations increases, the performance of DM$^+$-IS initially improves and then stays consistent. Error bars represent 95% confidence intervals.

Figure 4: **DM$^+$-IS performs best on MIMIC-IV**, a setting where annotations are likely imperfect.

small relative to the range of possible reward in the environment. That is, DM$^+$-IS produces estimates of $v(\pi_e)$ that are close to those of the best performing OPE method. This result suggests that, in a setting where it is difficult to assess reward model or annotation quality, DM$^+$-IS is the natural choice.

## 6 Conclusion

In this work, we address the open problem of incorporating imperfect counterfactual annotations into an OPE estimator and present a practical guide for their integration. We systematically explore various design options for incorporating annotations into a DR-based OPE estimator, and we find that imperfect counterfactual annotations are most beneficial when incorporated into the DM part of a DR estimator. Through comprehensive theoretical analyses and empirical evaluations, we find that selecting the best OPE method hinges on two critical factors: (1) whether the reward model is well-specified, and (2) the annotation quality. We conclude that, under the most realistic conditions (i.e., a misspecified reward model and imperfect annotations), our DM$^+$-IS estimator is most robust.

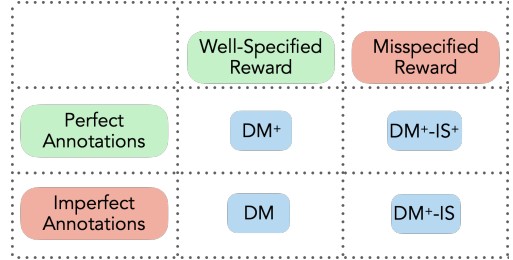

Figure 5: **Lookup table capturing the practical considerations when choosing an OPE estimator:** The most critical factors include (1) whether the reward model is well-specified and (2) the quality of the annotations. If these factors are known a priori, the best OPE estimator can be easily identified.

**Limitations and Future Work.** This work focuses on the contextual bandit setting, with future directions including extensions to the MDP setting. Additionally, this work considers a subset of possible reward function parameterizations. A promising avenue for future work includes optimizing the use of a limited budget of counterfactual annotations to improve OPE performance. Furthermore, future work can consider identifying practical ways to preprocess a set of counterfactual annotations to mitigate the effect of bias and noise. Overall, our approach relaxes restrictive assumptions about annotation quality, enabling more practical use of bandit algorithms in high-stakes applications.

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

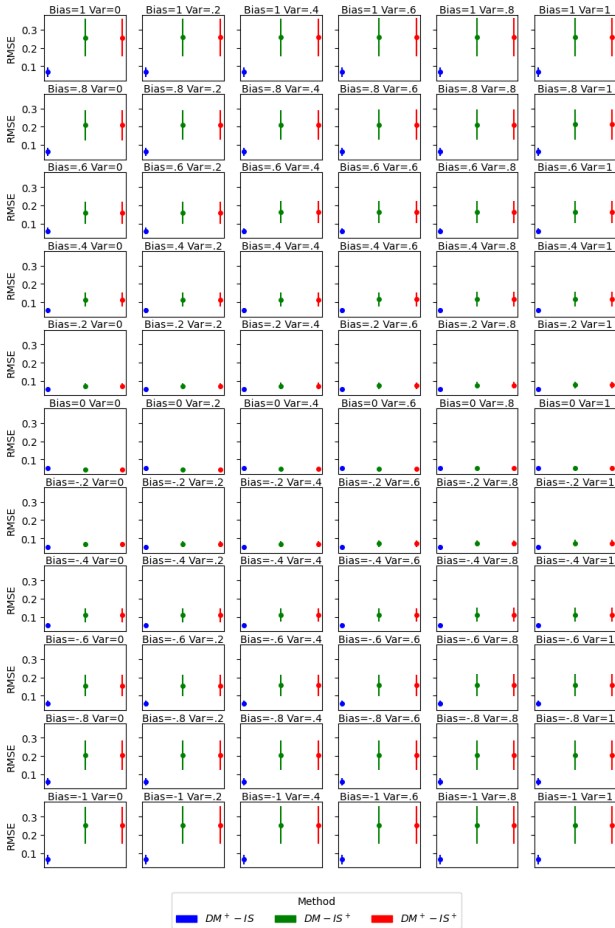

Figure 6: **Investigating the performance of the proposed estimators under imperfect annotations and a well-specified reward:** Here, we report the results observed in Figure 2 in terms of a line plot. Each plot depicts the performance in terms of RMSE of the three proposed estimators under a particular value of annotation bias and variance. Error bars represents standard deviation.

# A    Additional Empirical Results

In the main text, we primarily focus on the case when the annotations are imperfect and the reward model is misspecified. We characterize the performance of the baseline OPE methods and our proposed approaches in four settings: either a well-specified or misspecified reward model, and either perfect or imperfect annotations. Here, we report additional results for each setting including error metrics on plots reported in the main text.

## A.1    Well-Specified Reward

In Figure 2, we note that the bias of the counterfactual annotation plays a larger role in affecting the RMSE of the proposed methods than the variance of the counterfactual annotation. We report the same heatmap observed in Figure 2 in terms of a line plot in Figure 6. Here, we report the mean standard deviation of the RMSE across all datasets (Figure 7). Our results indicate that $DM^+$-IS has the least fluctuation in standard deviation across the range of counterfactual annotation bias and variance in comparison to all methods that use counterfactual annotations.

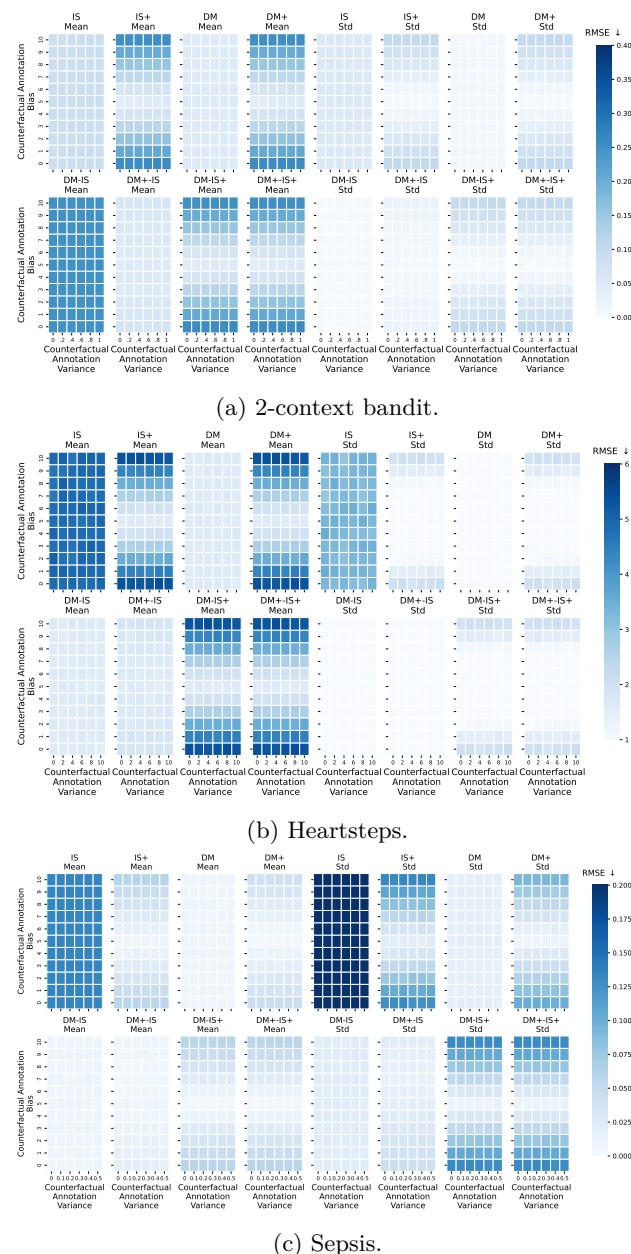

(a) 2-context bandit.

(b) Heartsteps.

(c) Sepsis.

Figure 7: **Heatmap of mean and standard deviation of RMSE with a well-specified reward model (lower mean/standard deviation is represented lighter)**: The $x, y$-axis represents the variance ($\Delta_G$) and the bias ($\epsilon_G$) of the annotations, respectively. In a well-specified reward setting, DM and DR perform comparably and best. Of note, IS sometimes performs well in this setting largely due to high coverage in the behavior policy for most pairings. In terms of standard deviation of RMSE, all methods that use counterfactual annotations with the exception of DR experience high standard deviation for the most imperfect annotations.

## A.2 Misspecified Reward

In Figure 3 we report mean RMSE across a variety of counterfactual annotation bias values in the Heartsteps and Sepsis settings (Figure 8). Here, we report the mean and standard deviation of RMSE across all three datasets.

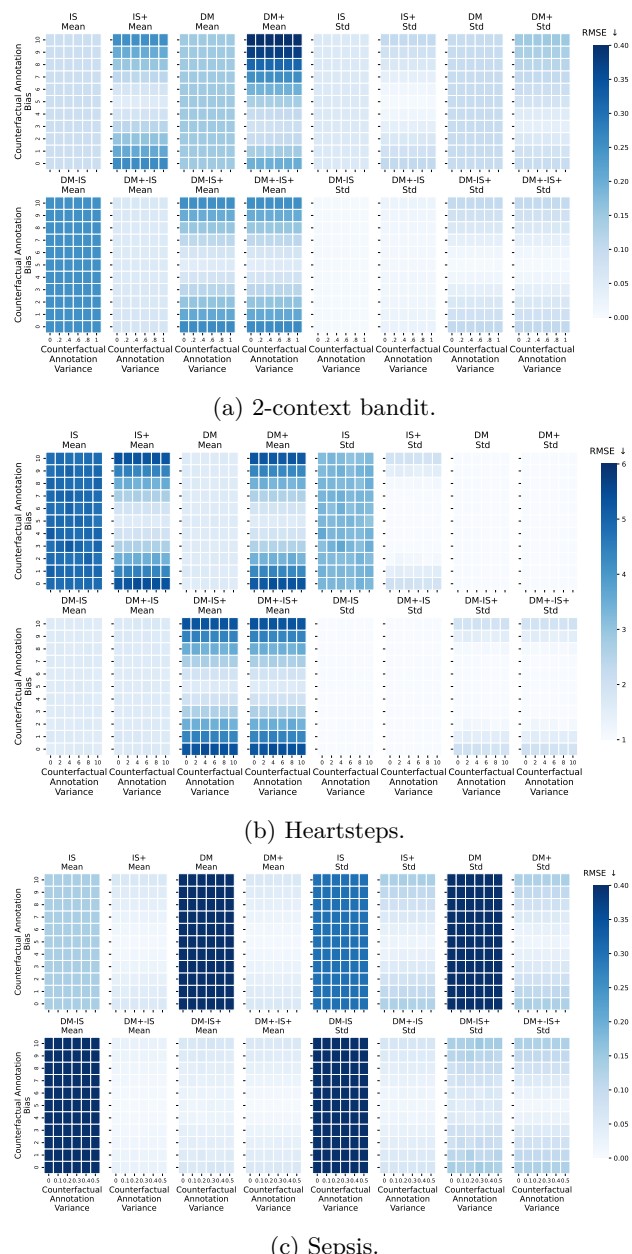

(a) 2-context bandit.

(b) Heartsteps.

(c) Sepsis.

Figure 8: **Heatmap of mean and standard deviation of RMSE with a misspecified reward model (lower mean/standard deviation is represented lighter)**: The $x, y$-axis represents the variance ($\Delta_G$) and the bias ($\epsilon_G$) of the annotations, respectively. In a misspecified reward setting, DM and DM$^+$ tend to suffer. In comparison, DM$^+$-IS outperforms all method or performs comparably to the best performing methods for both RMSE mean and standard deviation.

In Figure 9, we claim that across all approaches that use counterfactual annotations, DM$^+$-IS is best suited to perform OPE when we do not know the annotation or reward model quality. To support this claim, we report the same plot for DM-IS$^+$ and DM$^+$-IS$^+$ in Figure 13.

We also report the results from the heatmaps observed in Figure 3, presented in terms of a line plot Figure 10. Our conclusions remain identical to those discussed in Section 4.

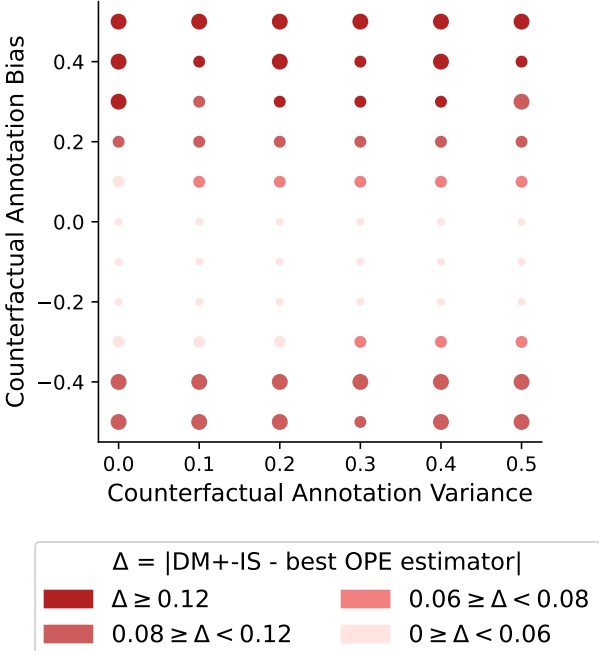

Figure 9: **Exploring the consequences of always using DM$^+$-IS in the sepsis treatment environment:** Usually, the reward model or annotation quality is unknown. Regardless, we choose DM$^+$-IS to estimate $v(\pi_e)$ and compare it to the best performing OPE method. We find that regardless of how imperfect the annotations are, DM$^+$-IS produces estimates with low mean $\Delta$ relative to the range of reward in the sepsis environment, which is 5.2. The color of the dots represents the magnitude of mean $\Delta$, and the size is proportional to the variance of $\Delta$. $\Delta$ increases in magnitude proportional to the magnitude of the annotation bias ($\epsilon_G$), though even in extreme cases, $\Delta$ is relatively small.

Finally, as discussed in Section 3, our goal is to define an estimator that is "doubly robust" to two sources of error, namely the error of the annotation and the error of the reward model. As such, we do not account for imperfect IPS ratios and assume that the estimates of IPS ratios are fairly accurate. In the case that the IPS ratio is inaccurate, all proposed estimators will be biased. The estimation error of the IPS ratio will thus propagate through the bias and variance reductions introduced in our work. To further illustrate this, we report the performance of DM$^+$-IS with varying degrees of incorrectly estimated behavior policies $\epsilon$ (Figure 11). We leave further evaluation of this setting to future work.

### A.3 Additional MIMIC-IV Results

In Section 4.2, we compared the performance of all estimators with 100 counterfactual annotations. Now, we demonstrate that DM$^+$-IS has the most favorable performance across a wider range of $m$, the number of counterfactual annotations available **??**. In particular, we note that the performance of DM$^+$-IS stays consistently better in comparison to the other estimators across all values of $m$ and that the performance either improves or remains consistent, but does not degrade if you increase the number of counterfactual annotations.

## B  Simulator environments

In all of our experiments, we use a separate random seed to generate datasets. We report results across 50 runs, each of which is a sampled dataset. Our results across all experiments take approximately 100 hours of compute, which was run on a local university cluster. Our code is attached as a portion of the supplement material and will be made public upon acceptance via a Github link. Our code is covered under the MIT

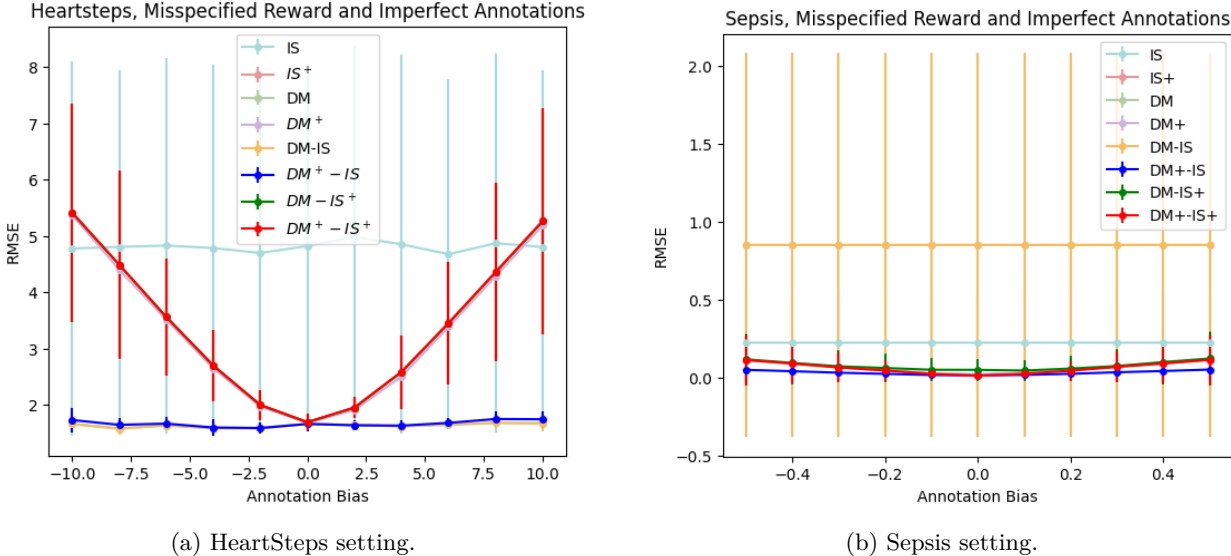

(a) HeartSteps setting.

(b) Sepsis setting.

Figure 10: **DM$^+$-IS performs best on the HeartSteps and Sepsis environments with imperfect annotations and misspecified reward models**. Here, we report the same results observed in Figure 3, except represented as a line plot.

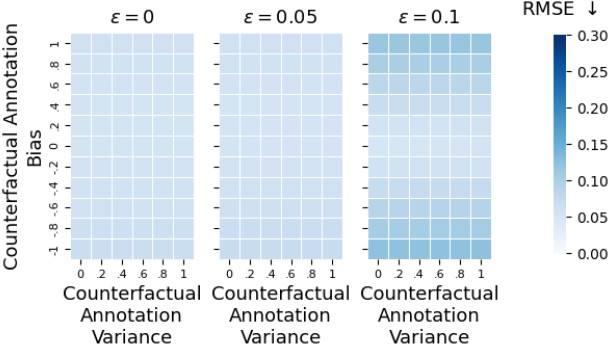

Figure 11: **Analyzing the consequences of incorrect IPS ratios in DM$^+$-IS in the Two-context Bandit:** The $x, y$ axes correspond to the annotation variance and bias respectively. The heatmap color corresponds to the mean RMSE of the DM$^+$-IS estimator. We study three settings of $\epsilon$, where $\hat{\pi}_b = \pi_b + \epsilon$ is the estimated behavior policy. As $\epsilon$ increases, DM$^+$-IS becomes a more biased estimator.

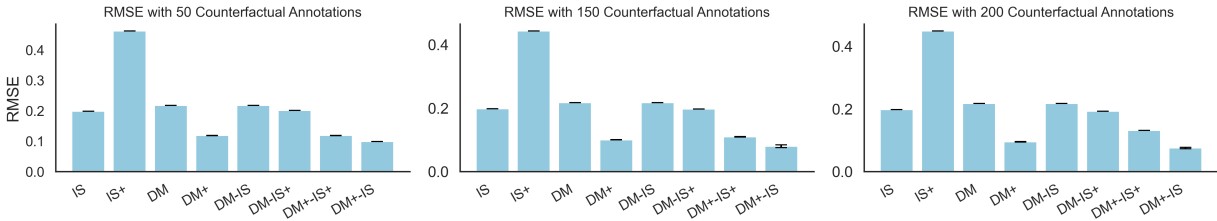

Figure 12: **Comparing the performance of all estimators in MIMIC-IV as we increase the number of counterfactual annotations.** The x-axis enumerates the possible estimators, and the y-axis measures the RMSE of the learned policy estimate. Error bars represent standard deviation. Each plot corresponds to the performance at different number of counterfactual annotations. We note that DM$^+$-IS has the most favorable performance across all settings.

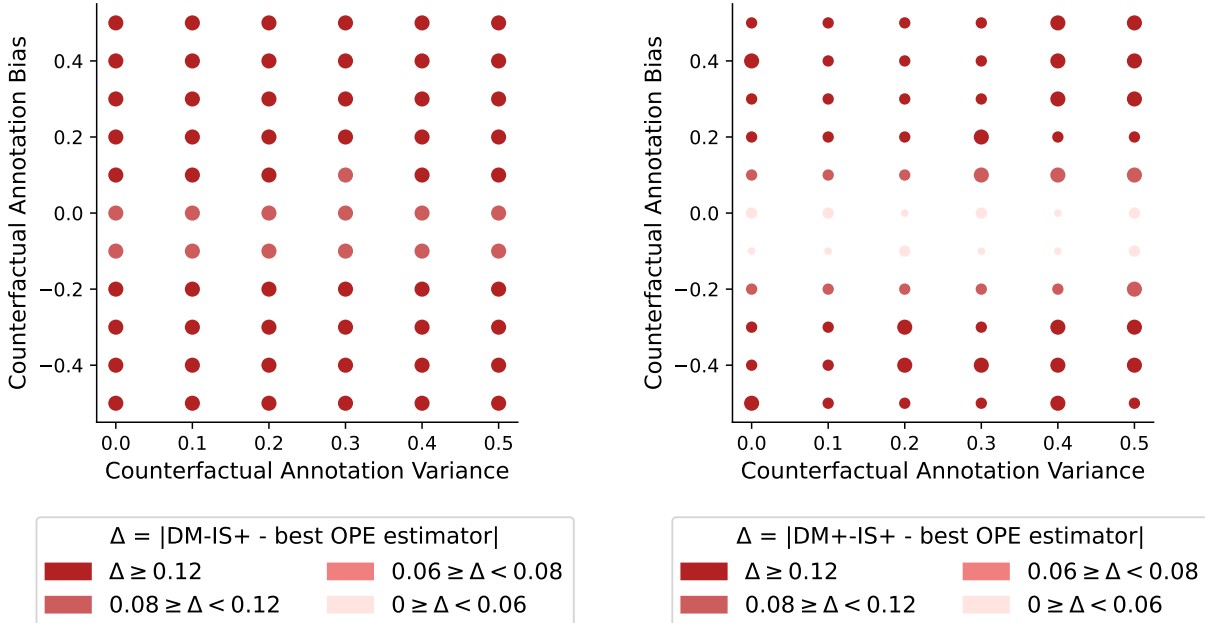

(a) Exploring the consequences of choosing DM-IS$^+$ when annotation and reward model quality are unknown.

(b) Exploring the consequences of choosing DM$^+$-IS$^+$ when annotation and reward model quality are unknown.

Figure 13: Here we compare the consequences of choosing the other two proposed estimators (DM-IS$^+$ and DM$^+$-IS$^+$). Note that across the range of possible annotation quality, DM-IS$^+$ has a larger error (darker color) in comparison to DM$^+$-IS as displayed in Figure 9. While DM$^+$-IS$^+$ has smaller $\Delta$ when the counterfactual annotations have low variance, $\Delta$ more rapidly increases as the annotations become more imperfect.

license. Here we discuss the implementation details for the three simulator settings we use in our empirical results.

## B.1 Two-context bandit

The two-context bandit setting contains two contexts, each with two actions. The bandit receives reward for taking either action from the first context, and no reward for taking any action from the second context. In particular, the reward is sampled from the normal distribution $\mathcal{N}(1, 0.5)$ for the first action in the first context and the distribution $\mathcal{N}(2, 0.5)$ for the second action in the first context. In this domain, we sample a counterfactual annotation for every sample. Factual rewards are sampled from the reward distribution. We use a uniform initial context distribution, and report results over 100 runs for all estimators. We use a dataset of size $N = 100$ to fit the OPE estimate and $M = 100$ counterfactual annotations. The same dataset size and distribution is used to learn a reward function estimate if necessary depending on the OPE method. We equally weight all samples (0.5). The reward function is represented as a sample mean. A misspecified reward function uses a partially observed context. In particular, for 50% of the samples, the context is randomly selected. There are no hyperparameters to be tuned. We report results averaged across 9 possible combinations of $\pi_b, \pi_e$. We use all combinatorial combinations of the policies $[0.1, 0.9], [0.5, 0.5], [0.9, 0.1]$.

## B.2 Heartsteps

The Heartsteps simulator is a step count simulator based off earlier work (Mandyam et al., 2024). All samples are sampled randomly and independently from an initial context distribution $d_0$ where each context is the square root of the prior day's step count. There are two possible actions: send a notification, or do not send a notification. The context and action are projected into $\mathbb{R}^3$ using a function $\phi(s, a)$, which outputs a vector that contains a scalar to represent the eventual decrease in step count over time, the previous day's

step count, and a treatment effect term that is nonzero when the action is nonzero. Then, the step count is calculated as $\phi(s, a) \cdot \theta^T$ where $\theta = [-0.04, 0.9999, 0.3]$. The well-specified reward model is a linear function of $\phi(s, a)$. The misspecified reward model is a linear function of only the first two indices of the vector $\phi(s, a)$. We report results averaged across 9 possible combinations of $\pi_b, \pi_e$. We use all combinatorial combinations of the policies $[0.1, 0.9], [0.5, 0.5], [0.9, 0.1]$.

## B.3   Sepsis Treatment

The sepsis treatment simulator is based off prior work (Oberst & Sontag, 2019). While the original simulator assumes a Markov Decision Process (MDP) setting, we adapt the simulator to accommodate a contextual bandit setting. To do this, we sample one-step transitions rather than full trajectories. There are 1442 possible contexts and 8 possible actions in the environment. The initial context distribution $d_0$ samples uniformly across the possible contexts. Each context is represented as a vector of length 8, where the features describe patient heart rate, systolic blood pressure, blood glucose level, percentage oxygen, and the presence of three treatments including antibiotics, vasopressors, and ventilation. The 8 possible actions represent every combinatorial combination of three binary treatments: antibiotics, vasopressors, and ventilation. The well-specified reward function is a linear function of the number of abnormal vitals, and whether the patient is on treatment or not, with $\theta = [-1, -1]$. The misspecified reward model one-hot-encodes the context and action and projects into a vector of length 168. The reward function is then a linear function of the vector of length 168. In this setting, we use one target policy, $\pi_e = [0.3, 0.2, 0, 0, 0.2, 0.1, 0.1, 0.1]$. We report results averaged across the following behavior policies:

$$\pi_{b1} = [0.1, 0.1, 0.4, 0.3, 0.1, 0.0, 0.0, 0.0]$$
$$\pi_{b2} = [0.1, 0.1, 0.4, 0.2, 0.1, 0.1, 0.0, 0.0]$$
$$\pi_{b3} = [0.1, 0.1, 0.4, 0.1, 0.1, 0.1, 0.0, 0.1]$$
$$\pi_{b4} = [0.1, 0.1, 0.3, 0.1, 0.1, 0.1, 0.1, 0.1]$$
$$\pi_{b5} = [0.2, 0.1, 0.2, 0.1, 0.1, 0.1, 0.1, 0.1]$$
$$\pi_{b6} = [0.3, 0.1, 0.2, 0.0, 0.1, 0.1, 0.1, 0.1]$$

## B.4   MIMIC-IV

Here, we use data from MIMIC-IV (Johnson et al., 2020), an Electronic Health Records dataset sourced from the Beth Israel Deaconess Medical Center in Boston, MA. We consider a subset of the patients who receive potassium repletion. That is, we include all patients who have received at least one instance of potassium administration through an IV. Furthermore, we treat this setting as a one-step contextual bandit setting. A patient context is represented as a 20-dimensional vector containing information about static covariates (e.g., age, gender), aggregated lab values observed in the previous four hour window, aggregated medicines administered in the previous four hour window, and any indication of procedures that were undertaken (e.g., the patient was placed on a ventilator). There are five possible actions, each corresponding to a dosage of potassium (units are mEq). After administering a dosage of potassium, we observe a reward that is a function of the patient's updated context.

In our other empirical results, we report RMSE as the key metric; this hinges on the fact that we know the reward function for this setting. To emulate this in MIMIC-IV, a setting in which there is no reported reward signal, we construct a specific reward function. In particular, this reward function is a binary indicator of whether a patient's potassium lab value observed after potassium administration is within the potassium reference range (3.5-5 mmol/L). That is, the reward function $R(s_{t+1})$ is a function of the patient's next observed context. It is reasonable to assume that the reward function is shared across all patients because the potassium reference range is shared across all patients.

To emulate a setting in which we have a behavior policy and a target policy, we further split the cohort of patients that receive potassium repletion into two sub-cohorts. In particular, one sub-cohort does not have renal disease, and one sub-cohort does have renal disease. The repletion policies are different between these

cohorts because the policy must consider the inability for the renal disease patients' kidneys to properly filter potassium. We treat the non-renal population as the behavior cohort and the renal population as the target cohort. The value of the target policy can be calculated by treating the target samples as Monte-Carlo samples. We estimate the policies from the data.

We expect the repletion policies between the two groups to differ since the policy for renal patients must account for the inability of their kidneys to efficiently absorb potassium. This setup allows us to calculate the ground-truth value of the target policy using the returns of the target trajectories. As shown in Figure 14, patients with renal disease are administered a variety of dosages, including ones that are under-observed in the behavior dataset (e.g., 10 mEq); as a result, this is a setting in which OPE can be useful. Finally, we

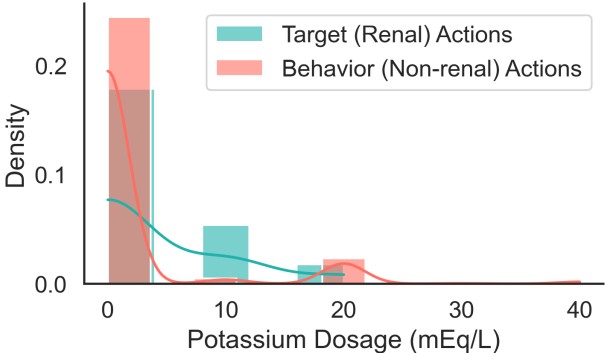

Figure 14: **Repletion policies differ between the behavior and target cohorts**, making this a setting in which OPE can be helpful. In particular, we note that the target dataset samples actions (e.g., 10 mEq of potassium) that are under-observed in the behavior dataset.

construct annotations using LLMs for the behavior cohort. The annotations are sourced from a powerful LLM, OpenAI 'o1'. LLMs have been successful in reasoning about medical domains, presenting a cheaper alternative to soliciting an expert (i.e., doctor). To obtain annotations, we ask the LLM to complete two tasks. The first is to summarize the patient context, highlighting covariates relevant for predicting blood potassium level. Then, we ask the LLM to predict the patient's blood potassium if they were administered a given dosage. Note that we have ground truth lab values for the dosages administered in MIMIC-IV; in our experiments we found 'o1' predictions to largely align with ground truth lab values. Annotations are obtained by soliciting lab value predictions for unobserved dosages and feeding these values into the known reward function.

## C Naive Doubly Robust Estimator

As discussed in Section 1, a naive doubly robust estimator uses the dataset $D^+$ within the context of a standard doubly robust estimator. The definition of the naive doubly robust estimator is

$$\hat{V}^{\text{Naive-DR}} = \sum_{i=1}^{N+M} \left( \hat{R}^+(s_i, \pi_e) + \rho_{s_i}(a_i)(c_i^{a_i} - \hat{R}^+(s_i, a_i)) \right),$$

where $c_i^{a_i}$ is the factual reward or counterfactual annotation depending on if $(s_i, a_i)$ is a factual or counterfactual sample. We claim in the main text that this approach will always lead to an arbitrarily biased estimate of $v(\pi_e)$ because it does not preserve the context distribution in $D$. To demonstrate this, we report results (Table 2) for the naive doubly robust estimator in the 2-context bandit environment under a well-specified reward model and a perfect annotation setting (Assumption 2). Note that the standard doubly robust estimator (DR) is unbiased here, and in contrast, the naive doubly robust estimator is biased. This is because the naive doubly robust estimator alters the context distribution $d_0$, which is assumed to be constant between the behavior dataset and the dataset used to calculate the target policy value $v(\pi_e)$.

| Dataset | N | M | % annotated | $|\mathcal{A}|$ | $|\mathcal{S}|$ | # of $\pi_b$, $\pi_e$ | Well-specified reward | Misspecified reward |
|---|---|---|---|---|---|---|---|---|
| 2-context Bandit | 100 | 100 | 100 | 2 | 2 | 9 | Sample mean | A random state is observed 50% of the time, creating partial observability like ModelFail in Thomas & Brunskill (2016). |
| Heartsteps | 200 | 200 | 100 | 2 | 80 | 9 | Linear regression (3 features) | Linear regression (2 features) |
| Sepsis | 700 | 700 | 12.5 | 8 | 1442 | 6 | Linear regression (top 2 features) | Linear regression (168 least relevant features) |
| MIMIC-IV | 652 | Varies | Varies | 5 | > 13000 | 1 | N/A | Indicator function of whether the lab is within the reference range |

Table 1: Key characteristics of contextual bandits settings used in empirical results.

Table 2: Naive doubly robust estimator performance. We report mean ± standard error for all values.

| Method | RMSE | Bias | Std |
|---|---|---|---|
| Naive DR | 0.317 ± 0.0007 | 0.3038 ± 0.0922 | 0.092 ± 0.0001 |
| DR | 0.108 ± 0.0002 | -0.016± 0.1066 | 0.1067 ± 0.0001 |

## D   Table of theoretical results

We provide a summary for our theoretical results to guide the reader to the appropriate proofs (Table 3). Earlier work (Tang & Wiens, 2023) introduced an IS-based estimator that incorporates counterfactual annotations, and found that re-weighting the samples was required to maintain the context-action distribution defined by the original factual dataset. In our work, we note that this re-weighting is not necessary when building a reward function, which we discuss in Appendix E. On the journey to constructing a doubly robust estimator, we derive the bias and variance of the standard direct method OPE estimator in Appendix F. Our work identifies three opportunities to incorporate counterfactual annotations into a doubly robust estimator. In this work, we investigate the theoretical properties of these estimators under three annotation settings: perfect annotations, biased annotations, and higher variance annotations. We prove expectation and variance terms for each of our three estimators with and without assumptions on the annotation quality in Appendix G. Note that we avoid deriving the variance for DM-IS$^+$ and DM$^+$-IS$^+$ because these terms are very complicated. Instead, we analyze these variance terms empirically in our simulated settings. Finally, we establish an equivalence between two of our doubly robust approaches and IS$^+$ (Tang & Wiens, 2023) under an equal weighting scheme in Appendix H.

## E   Weighted vs. unweighted reward function

In the main text, we claim that we do not need to re-weight samples when constructing a reward function to produce an unbiased estimate, like earlier work had to do with IS$^+$ (Tang & Wiens, 2023). To demonstrate this, we first derive the expectation and variance of the weighted reward function, and identify that the unweighted (or equally weighted) reward function is a special case. We then compare the variance terms for the unweighted and weighted reward functions, and prove that the variance of the unweighted reward function is lower. This result suggests that not weighting the samples when constructing the reward function is a superior strategy in the case that the annotations are perfect.

Table 3: Summary of theoretical results and associated proofs. Note that only Assumption 3 is required for the corresponding bias proofs.

| Method | Assumption Annotation | Assumption Coverage | Bias | Std |
|---|---|---|---|---|
| DM$^+$-IS | 2 | 1 | Proposition 12 | Proposition 13 |
| DM-IS$^+$ | 2 | 5 | Proposition 14 | Proposition 15 |
| DM$^+$-IS$^+$ | 2 | 5 | Proposition 17 | Proposition 18 |
| DM$^+$-IS | 3,4 | 1 | Proposition 1 | Theorem 3 |
| DM-IS$^+$ | 3,4 | 5 | Theorem 2 | Proposition 16 |
| DM$^+$-IS$^+$ | 3,4 | 5 | Theorem 2 | Proposition 19 |

## E.1   Weighted Reward Function

The weighted reward function is

$$\hat{R}^+(s,a) = \frac{\sum_{i=1}^N \mathbb{1}(s_i = s, a_i = a) * w_i^a * r_i + \sum_{j=1}^M \mathbb{1}(s_j = s, a_j = a) * w_j^a * g_j}{\sum_{i=1}^N \mathbb{1}(s_i = s, a_i = a) * w_i^a + \sum_{j=1}^M \mathbb{1}(s_j = s, a_j = a) * w_j^a}$$

where $w_i^a \sim W(s_i, a_i)$ is a known weight associated with the sample $s_i, a_i$ that arises from some function $W$.

### E.1.1   Bias

**Proposition 4** (Unbiasedness of weighted reward function). *Under Assumption 2, the weighted reward function is unbiased.* $\mathbb{E}[\hat{R}^+(s,a)] = \bar{R}(s,a)$.

Proof: Let $N(s,a) = \sum_{i=1}^{M+N} \mathbf{1}(s_i = s, a_i = a)$ be the number of times the sample $s, a$ appears in the counterfactual-annotated dataset $D^+$. We can re-write the weighted reward function as a function of $N(s,a)$. Let $c_i = r_i$ if the sample is in the factual dataset, and $c_i = g_i$ if the sample is a counterfactual annotation.

$$\hat{R}^+(s,a) = \frac{1}{\sum_{i=1}^{N(s,a)} w_i^a} \sum_{i=1}^{N(s,a)} w_i^a c_i$$

To calculate expectation, we first use the law of total expectation and the expectation of the scalar reward or annotation.

$$\mathbb{E}_{D^+ \sim \mathcal{D}^+}[\hat{R}^+(s,a)] = \mathbb{E}_{\substack{N_{s,a} \sim D^+, c_i \sim R(s_i, a_i), \\ w_i^a \sim W(s_i, a)}} \left[ \frac{1}{\sum_{i=1}^{N(s,a)} w_i^a} \sum_{i=1}^{N(s,a)} w_i^a c_i \right] \tag{6}$$

$$= \mathbb{E}_{N_{s,a} \sim D^+} \left[ \mathbb{E}_{c_i \sim R(s_i, a_i), w_i^a \sim W(s_i, a)} \left[ \frac{1}{\sum_{i=1}^{N(s,a)} w_i^a} \sum_{i=1}^{N(s,a)} w_i^a c_i \right] \right] \tag{7}$$

$$= \mathbb{E}_{N_{s,a} \sim D^+} \left[ \mathbb{E}_{w_i^a \sim W(s_i, a)} \left[ \frac{1}{\sum_{i=1}^{N(s,a)} w_i^a} \sum_{i=1}^{N(s,a)} w_i^a \mathbb{E}_{c_i \sim R(s_i, a_i)}[c_i] \right] \right] \tag{8}$$

$$= \mathbb{E}_{N_{s,a} \sim D^+} \left[ \mathbb{E}_{w_i^a \sim W(s_i, a)} \left[ \frac{1}{\sum_{i=1}^{N(s,a)} w_i^a} \sum_{i=1}^{N(s,a)} w_i^a \bar{R}(s,a) \right] \right] \tag{9}$$

Let $W_i^a = \frac{w_i^a}{\sum_{i=1}^{N(s,a)} w_i^a}$. Because each $w_i^a$ is normalized by the sum of all the weights across all the samples, the sum of all weights is 1. Now, we can write the sum as a weighted mean:

$$= \mathbb{E}_{N_{s,a} \sim D^+} \left[ \mathbb{E}_{w_i^a \sim W(s_i, a)} \left[ \sum_{i=1}^{N(s,a)} W_i^a \bar{R}(s,a) \right] \right] \tag{10}$$

$$(11)$$

Because $\bar{R}(s_i, a_i)$ is a constant, we can pull it out of the expectation.

$$= \mathbb{E}_{N_{s,a} \sim D^+} \left[ \mathbb{E}_{w_i^a \sim W(s_i, a)} \left[ \bar{R}(s, a) \sum_{i=1}^{N(s,a)} W_i^a \right] \right] \tag{12}$$

$$= \mathbb{E}_{N_{s,a} \sim D^+} \left[ \mathbb{E}_{w_i^a \sim W(s_i, a)} \left[ \bar{R}(s, a) \right] \right] \tag{13}$$

$$= \bar{R}(s, a) \tag{14}$$

### E.1.2  Variance

**Proposition 5.** *Under [Assumption 2](), the weighted reward function has variance,* $\mathbb{V}[\hat{R}^+(s, a)] = \mathbb{E}_{N_{s,a} \sim D^+} \left[ \sum_{i=1}^{N(s,a)} (W_i^a)^2 \sigma_R^2(s, a) \right]$.

Proof: We use the law of total variance and the definition of $W_i^a$.

$$\mathbb{V}_{D^+ \sim \mathcal{D}^+}[\hat{R}^+(s, a)] = \mathbb{V}_{\substack{N_{s,a} \sim D^+, \\ c_i \sim R(s_i, a_i)}} \left[ \sum_{i=1}^{N(s,a)} \frac{w_i^a c_i}{\sum_{i=1}^{N(s,a)} w_i^a} \right] \tag{15}$$

$$= \mathbb{E}_{N_{s,a} \sim D^+} \left[ \mathbb{V}_{c_i \sim R(s_i, a_i)} \left[ \sum_{i=1}^{N(s,a)} \frac{w_i^a c_i}{\sum_{i=1}^{N(s,a)} w_i^a} \right] \right] \tag{16}$$

$$+ \mathbb{V}_{N_{s,a} \sim D^+} \left[ \mathbb{E}_{c_i \sim R(s_i, a_i)} \left[ \sum_{i=1}^{N(s,a)} \frac{w_i^a c_i}{\sum_{i=1}^{N(s,a)} w_i^a} \right] \right] \tag{17}$$

$$= \mathbb{E}_{N_{s,a} \sim D^+} \left[ \mathbb{V}_{c_i \sim R(s_i, a_i)} \left[ \sum_{i=1}^{N(s,a)} \frac{w_i^a c_i}{\sum_{i=1}^{N(s,a)} w_i^a} \right] \right] \tag{18}$$

$$= \mathbb{E}_{N_{s,a} \sim D^+} \left[ \mathbb{V}_{c_i \sim R(s_i, a_i)} \left[ \sum_{i=1}^{N(s,a)} W_i^a c_i \right] \right] \tag{19}$$

$$= \mathbb{E}_{N_{s,a} \sim D^+} \left[ \sum_{i=1}^{N(s,a)} (W_i^a)^2 \mathbb{V}_{c_i \sim R(s,a)} [c_i] \right] \tag{20}$$

$$= \mathbb{E}_{N_{s,a} \sim D^+} \left[ \sum_{i=1}^{N(s,a)} (W_i^a)^2 \sigma_R^2(s, a) \right] \tag{21}$$

$$(22)$$

### E.2  Unweighted Reward Function

Here the unweighted reward function is identical to the weighted one, except without weights. This can also be seen as a special case of the weighted reward function with equal weights. The unweighted reward function is :

$$\hat{R}^+(s, a) = \frac{\sum_{i=1}^{N} \mathbb{1}(s_i = s, a_i = a) * r_i + \sum_{j=1}^{M} \mathbb{1}(s_j = s, a_j = a) * g_j}{\sum_{i=1}^{N} \mathbb{1}(s_i = s, a_i = a) + \sum_{j=1}^{M} \mathbb{1}(s_j = s, a_j = a)} \tag{23}$$

### E.2.1 Bias

**Proposition 6.** *Under Assumption 3, the unweighted reward function has expectation* $\mathbb{E}[\hat{R}^+(s,a)] = \mathbb{E}_{N_{s,a}, M_{s,a}} \left[ \bar{R}(s_i, a_i) + \frac{M_{s,a}\epsilon_G}{N_{s,a}+M_{s,a}} \right]$ *where* $N_{s,a}, M_{s,a}$ *are the number of factual and counterfactual samples with context-action equivalent to* $s, a$ *in the counterfactual augmented dataset.*

Proof: Define $N_{s,a} = \sum_{i=1}^{N} \mathbb{1}(s_i = s, a_i = a)$ and $M_{s,a} = \sum_{j=1}^{M} \mathbb{1}(s_j = s, a_j = a)$ where $N$ is the number of total factual samples and $M$ is the total number of counterfactual annotations.

Now, we can re-write the reward function as $\hat{R}^+(s,a) = \frac{\sum_{i=1}^{N_{s,a}} r_i + \sum_{j=1}^{M_{s,a}} g_j}{N_{s,a}+M_{s,a}}$.

We calculate the expectation of this reward function under the dataset distribution, which is the joint distribution across $N_{s,a} \sim D^+, M_{s,a} \sim D^+, r \sim R(s,a), g \sim G(s,a)$.

$$\mathbb{E}_{D^+ \sim \mathcal{D}^+}[\hat{R}^+(s,a)] = \mathbb{E}_{N_{s,a}, M_{s,a} \sim D^+, r \sim R(s,a), g \sim G(s,a)} \left[ \frac{\sum_{i=1}^{N_{s,a}} r_i + \sum_{j=1}^{M_{s,a}} g_j}{N_{s,a} + M_{s,a}} \right] \tag{24}$$

We can use the law of total expectation to separate out the joint distribution:

$$= \mathbb{E}_{\substack{N_{s,a} \\ \sim D^+}} \left[ \mathbb{E}_{M_{s,a} \sim D^+, r \sim R(s,a), g \sim G(s,a)} \left[ \frac{\sum_{i=1}^{N_{s,a}} r_i + \sum_{j=1}^{M_{s,a}} g_j}{N_{s,a} + M_{s,a}} \right] \right] \tag{25}$$

$$= \mathbb{E}_{\substack{N_{s,a} \\ \sim D^+}} \left[ \mathbb{E}_{M_{s,a} \sim D^+} \left[ \mathbb{E}_{r \sim R(s,a), g \sim G(s,a)} \left[ \frac{\sum_{i=1}^{N_{s,a}} r_i + \sum_{j=1}^{M_{s,a}} g_j}{N_{s,a} + M_{s,a}} \right] \right] \right] \tag{26}$$

$$= \mathbb{E}_{\substack{N_{s,a} \\ \sim D^+}} \left[ \mathbb{E}_{M_{s,a} \sim D^+} \left[ \frac{1}{N_{s,a} + M_{s,a}} \sum_{i=1}^{N_{s,a}} \bar{R}(s_i, a_i) + \sum_{j=1}^{M_{s,a}} \bar{R}(s_j, a_j) + \epsilon_G \right] \right] \tag{27}$$

$$= \mathbb{E}_{N_{s,a} \sim D^+} \left[ \mathbb{E}_{M_{s,a} \sim D^+} \left[ \frac{1}{N_{s,a} + M_{s,a}} \sum_{i=1}^{N_{s,a}} \bar{R}(s_i, a_i) + \sum_{j=1}^{M_{s,a}} \bar{R}(s_j, a_j) + \epsilon_G \right] \right] \tag{28}$$

$$= \mathbb{E}_{N_{s,a} \sim D^+} \left[ \mathbb{E}_{M_{s,a} \sim D^+} \left[ \frac{1}{N_{s,a} + M_{s,a}} N_{s,a} \times \bar{R}(s_i, a_i) + M_{s,a} \times (\bar{R}(s_j, a_j) + \epsilon_G) \right] \right] \tag{29}$$

$$= \mathbb{E}_{N_{s,a} \sim D^+} \left[ \mathbb{E}_{M_{s,a} \sim D^+} \left[ \bar{R}(s_i, a_i) + \frac{M_{s,a}\epsilon_G}{N_{s,a} + M_{s,a}} \right] \right] \tag{30}$$

$$= \mathbb{E}_{N_{s,a}, M_{s,a} \sim D^+} \left[ \bar{R}(s_i, a_i) + \frac{M_{s,a}\epsilon_G}{N_{s,a} + M_{s,a}} \right] \tag{31}$$

$$\tag{32}$$

**Proposition 7.** *Under Assumption 2, the unweighted reward function is unbiased,* $\mathbb{E}[\hat{R}^+(s,a)] = \bar{R}(s,a)$.

Proof: If we set $\epsilon_G = 0$ in Proposition 6, the expected value of the unweighted reward function is $\bar{R}(s,a)$, which means that it is an unbiased estimator.

### E.2.2 Variance

**Proposition 8.** *Under Assumption 2, the unweighted reward function has variance,* $\mathbb{V}[\hat{R}^+(s,a)] = \sigma_R^2(s,a) \mathbb{E}_{N_{s,a}} \left[ \frac{1}{N_{s,a}} \right]$.

Proof: Let $N_{s,a}$ denote how many samples in the total dataset have the same state-action as $s, a$. We now use the law of total variance, the earlier bias result, and the expectation of $c_i$.

$$\mathbb{V}_{D^+ \sim \mathcal{D}^+}[\hat{R}^+(s,a)] = \mathbb{V}_{\substack{N_{s,a} \\ c_i \sim R(s,a)}} \left[ \frac{1}{N_{s,a}} \sum_{i=1}^{N_{s,a}} c_i \right] \tag{33}$$

$$= \mathbb{E}_{N_{s,a}} \left[ \mathbb{V}_{c \sim R} \left[ \frac{1}{N_{s,a}} \sum_{i=1}^{N_{s,a}} c_i \right] \right] \tag{34}$$

$$+ \mathbb{V}_{N_{s,a}} \left[ \mathbb{E}_{c_i \sim R(s,a)} \left[ \frac{1}{N_{s,a}} \sum_{i=1}^{N_{s,a}} c_i \right] \right] \tag{35}$$

$$= \mathbb{E}_{N_{s,a}} \left[ \mathbb{V}_{c_i \sim R(s,a)} \left[ \frac{1}{N_{s,a}} \sum_{i=1}^{N_{s,a}} c_i \right] \right] \tag{36}$$

$$= \mathbb{E}_{N_{s,a}} \left[ \frac{1}{N_{s,a}^2} \sum_{i=1}^{N_{s,a}} \mathbb{V}_{c_i \sim R(s,a)} \left[ c_i \right] \right] \tag{37}$$

$$= \mathbb{E}_{N_{s,a}} \left[ \frac{1}{N_{s,a}^2} \sum_{i=1}^{N_{s,a}} \sigma_R^2(s,a) \right] \tag{38}$$

$$= \mathbb{E}_{N_{s,a}} \left[ \frac{1}{N_{s,a}^2} \times N_{s,a} \times \sigma_R^2(s,a) \right] \tag{39}$$

$$= \mathbb{E}_{N_{s,a}} \left[ \frac{\sigma_R^2(s,a)}{N_{s,a}} \right] \tag{40}$$

$$= \sigma_R^2(s,a) \mathbb{E}_{N_{s,a}} \left[ \frac{1}{N_{s,a}} \right] \tag{41}$$

$$\tag{42}$$

### E.3 Comparing variance terms

**Proposition 9.** *The variance of the unweighted reward function is less than or equal to the variance of the weighted reward function.*

Proof: We can prove that the variance of the weighted reward function is higher than that of the unweighted reward function using Jensen's inequality. Since both the variance terms for the weighted and unweighted reward function have a $\sigma_R^2(s,a)$, we consider only the coefficient. We start with the coefficient for the weighted reward function on the LHS of the first line. We know by Jensen's inequality that:

$$\frac{1}{N_{s,a}} \sum_{i=1}^{N_{s,a}} (W_i^a)^2 \geq \frac{1}{N_{s,a}^2} \left( \sum_{i=1}^{N_{s,a}} W_i^a \right)^2 \tag{43}$$

$$\frac{1}{N_{s,a}} \sum_{i=1}^{N_{s,a}} (W_i^a)^2 \geq \frac{1}{N_{s,a}^2} \times 1 \tag{44}$$

$$\sum_{i=1}^{N_{s,a}} (W_i^a)^2 \geq \frac{1}{N_{s,a}} \tag{45}$$

Note that the coefficient of the variance term for the unweighted reward function is $\frac{1}{N_{s,a}}$. This result suggests that the variance of the randomly weighted reward function is always at least as high as the variance of the uniformly weighted reward function. As a result, we recommend using a uniformly weighted (or unweighted) reward function estimator when we have perfect annotations.

In the case that we know the annotations are imperfect, we recommend using a weighted reward function that down-weights the imperfect annotation (and consequently up-weights the corresponding factual sample). We hypothesize that this procedure will result in an improved estimator in both the misspecified and well-specified reward function scenarios because it limits the effect of the imperfect annotation.

We include a baseline which augments the standard DM estimator using counterfactual annotations. This estimator is $\hat{V}^{DM^+} = \sum_s d_0(s) \sum_a \pi(a|s) \hat{R}^+(s,a)$.

## F   Bias and variance of the standard direct method (DM) OPE estimator

In the main text, we reference the variance term for the standard direct method (DM) estimator. Here, we derive the expectation and variance of the DM estimator, which is defined as:

$$\hat{V}^{\mathrm{DM}} = \frac{1}{N} \sum_{s_i \in D_{\hat{R}}} \sum_{a \in A} \pi_e(a|s_i) \hat{R}(s_i, a) \tag{46}$$

For the purposes of the expectation and variance derivations below, we assume that the reward model is fully realizable according to Assumption 6.

**Assumption 6** (Realizability). $R^* \in \mathcal{F}$.

We derive the expectation and variance with respect to two datasets. $D_0$ is used to estimate the OPE value, and $D_{\hat{R}}$ is used to estimate the reward function.

**Proposition 10** (Unbiasedness of $\hat{V}^{DM}$). *Under Assumption 6, $\mathbb{E}_{D_0 \sim \mathcal{D}, D_{\hat{R}} \sim \mathcal{D}}[\hat{V}^{DM}] = v(\pi_e)$ if $N_{s,a} > 0$ for all $(s,a)$ where $d(s) > 0$ and $\pi_e(a|s) > 0$ in the dataset $D_{\hat{R}}$.*

Proof:

$$\mathbb{E}_{D_0 \sim \mathcal{D}, D_{\hat{R}} \sim \mathcal{D}}[\hat{V}^{\mathrm{DM}}] = \mathbb{E}_{D_0, D_{\hat{R}} \sim \mathcal{D}} \left[ \frac{1}{N} \sum_{s_i \in D_0} \sum_{a \in A} \pi_e(a|s_i) \hat{R}(s_i, a) \right] \tag{47}$$

$$= \mathbb{E}_{D_{\hat{R}} \sim \mathcal{D}} \left[ \frac{1}{N} \sum_{i=1}^{N} \sum_{a \in A} \pi_e(a|s_i) \mathbb{E}_{D_{\hat{R}} \sim \mathcal{D}}[\hat{R}(s_i, a)] \right] \tag{48}$$

$$= \frac{1}{N} \sum_{i=1}^{N} \mathbb{E}_{D_{\hat{R}} \sim \mathcal{D}} \left[ \sum_{a \in A} \pi_e(a|s_i) \mathbb{E}_{D_{\hat{R}} \sim \mathcal{D}}[\hat{R}(s_i, a)] \right] \tag{49}$$

$$\tag{50}$$

Now, because we consider every sample as independent, we consider just the term inside the summation without $\frac{1}{N}$.

$$= \sum_{i=1}^{N} \sum_{s \in \mathcal{S}} d(s) \left( \sum_{a \in A} \pi_e(a|s) \mathbb{E}_{D_{\hat{R}} \sim \mathcal{D}}[\hat{R}(s, a)] \right) \tag{51}$$

$$= \sum_{i=1}^{N} \sum_{s \in \mathcal{S}} d(s) \left( \sum_{a \in A} \pi_e(a|s) \bar{R}(s, a) \right) \tag{52}$$

$$= \sum_{i=1}^{N} \sum_{s \in \mathcal{S}} d(s) v^{\pi_e}(s_i) \tag{53}$$

$$= \sum_{i=1}^{N} v(\pi_e) \tag{54}$$

$$= v(\pi_e) \tag{55}$$

where we apply linearity of expectation, definition of expectation over $D_{\hat{R}}$, substitute the expectation of $\hat{R}(s,a)$ where $d(s) > 0$ and $\pi_e(a|s) > 0$, definition of value function and policy value.

**Proposition 11** (Variance of $\hat{V}^{DM}$). *Under Assumption 6, $\mathbb{V}_{D \sim \mathcal{D}, D_{\hat{R}} \sim \mathcal{D}}[\hat{V}^{DM}] = \frac{1}{N} \mathbb{V}_{s \sim d_0}[V^{\pi_e}(s)] + \mathbb{E}_{s \sim d_0} \mathbb{E}_{a \sim \pi_e} \left[ \left( \frac{1}{N} + (1 - \frac{1}{N}) d(s) \right) \pi_e(a|s) \sigma_R^2(s, a) \mathbb{E}_{D_{\hat{R}} \sim \mathcal{D}} \left[ \frac{1}{N_{s,a}(D_{\hat{R}})} \right] \right].$*

Proof: By law of total variance, we can decompose the variance into two terms:

$$\mathbb{V}_{D_{\hat{R}}\sim\mathcal{D},D_{\hat{R}}\sim\mathcal{D}}[\hat{V}^{\mathrm{DM}}] = \underbrace{\mathbb{V}_{D_{\hat{R}}\sim\mathcal{D}}\mathbb{E}_{D_{\hat{R}}\sim\mathcal{D}}\left[\hat{V}^{\mathrm{DM}}\right]}_{(1)} + \underbrace{\mathbb{E}_{D_{\hat{R}}\sim\mathcal{D}}\mathbb{V}_{D_{\hat{R}}\sim\mathcal{D}}\left[\hat{V}^{\mathrm{DM}}\right]}_{(1')} \tag{56}$$

We can substitute the intermediate result from the proof of Proposition 10 into (1):

$$(1) = \mathbb{V}_{D_{\hat{R}}\sim\mathcal{D}}\left[\mathbb{E}_{D_{\hat{R}}\sim\mathcal{D}}\left[\hat{V}^{\mathrm{DM}}\right]\right] = \mathbb{V}_{D_{\hat{R}}\sim\mathcal{D}}\left[\hat{R}(d,\pi_e)\right] \tag{57}$$

For $(1')$, we first consider the inner variance with respect to $D_{\hat{R}}$ assuming $\hat{R}$ is given:

$$\mathbb{V}_{D_{\hat{R}}\sim\mathcal{D}}\left[\hat{V}^{\mathrm{DM}}\right] = \mathbb{V}_{D_{\hat{R}}\sim\mathcal{D}}\left[\frac{1}{N}\sum_{i=1}^{N}\sum_{a\in A}\pi_e(a|s_i)\hat{R}(s_i,a)\right] \tag{58}$$

$$= \frac{1}{N^2}\sum_{i=1}^{N}\mathbb{V}_{s_i\sim d_0}\left[\hat{R}(s_i,\pi_e)\right] \qquad\qquad \text{var of sum of iid} \tag{59}$$

$$= \frac{1}{N}\mathbb{V}_{s\sim d_0}\left[\hat{R}(s,\pi_e)\right] \qquad\qquad \text{iid sample average} \tag{60}$$

Substituting this into $(1')$:

$$(1') = \mathbb{E}_{D_{\hat{R}}\sim\mathcal{D}}\mathbb{V}_{D_{\hat{R}}\sim\mathcal{D}}\left[\hat{V}^{\mathrm{DM}}\right] \tag{61}$$

$$= \mathbb{E}_{D_{\hat{R}}\sim\mathcal{D}}\left[\frac{1}{N}\mathbb{V}_{s\sim d_0}\left[\hat{R}(s,\pi_e)\right]\right] \tag{62}$$

$$= \frac{1}{N}\mathbb{E}_{D_{\hat{R}}\sim\mathcal{D}}\left[\mathbb{E}_{s\sim d_0}\left[\hat{R}(s,\pi_e)^2\right] - \mathbb{E}_{s\sim d_0}\left[\hat{R}(s,\pi_e)\right]^2\right] \quad \text{definition of var} \tag{63}$$

$$= \frac{1}{N}\left(\underbrace{\mathbb{E}_{D_{\hat{R}}\sim\mathcal{D}}\mathbb{E}_{s\sim d_0}\left[\hat{R}(s,\pi_e)^2\right]}_{(2)} - \underbrace{\mathbb{E}_{D_{\hat{R}}\sim\mathcal{D}}\left[\mathbb{E}_{s\sim d_0}\left[\hat{R}(s,\pi_e)\right]^2\right]}_{(2')}\right) \tag{64}$$

$$(2) = \mathbb{E}_{D_{\hat{R}}\sim\mathcal{D}}\mathbb{E}_{s\sim d_0}\left[\hat{R}(s,\pi_e)^2\right] \tag{65}$$

$$= \mathbb{E}_{s\sim d_0}\mathbb{E}_{D_{\hat{R}}\sim\mathcal{D}}\left[\hat{R}(s,\pi_e)^2\right] \tag{66}$$

$$= \mathbb{E}_{s\sim d_0}\left[v^{\pi_e}(s)^2 + \mathbb{V}_{D_{\hat{R}}\sim\mathcal{D}}[\hat{R}(s,\pi_e)]\right] \qquad \text{substitute corollary} \tag{67}$$

$$= \mathbb{E}_{s\sim d_0}\left[v^{\pi_e}(s)^2\right] + \mathbb{E}_{s\sim d_0}\left[\mathbb{V}_{D_{\hat{R}}\sim\mathcal{D}}[\hat{R}(s,\pi_e)]\right] \qquad \text{linearity of expectation} \tag{68}$$

$$= \mathbb{E}_{s\sim d_0}\left[v^{\pi_e}(s)\right]^2 + \mathbb{V}_{s\sim d_0}\left[v^{\pi_e}(s)\right] + \mathbb{E}_{s\sim d_0}\left[\mathbb{V}_{D_{\hat{R}}\sim\mathcal{D}}[\hat{R}(s,\pi_e)]\right] \quad \text{definition of variance} \tag{69}$$

$$= v(\pi_e)^2 + \mathbb{V}_{s\sim d_0}\left[v^{\pi_e}(s)\right] + \mathbb{E}_{s\sim d_0}\left[\mathbb{V}_{D_{\hat{R}}\sim\mathcal{D}}[\hat{R}(s,\pi_e)]\right] \quad \text{definition of value function} \tag{70}$$

$$\tag{71}$$

$$(2') = \mathbb{E}_{D_{\hat{R}}\sim\mathcal{D}}\left[\mathbb{E}_{s\sim d_0}\left[\hat{R}(s,\pi_e)\right]^2\right] \tag{72}$$

$$= \mathbb{E}_{D_{\hat{R}}\sim\mathcal{D}}\left[\hat{R}(d,\pi_e)^2\right] \tag{73}$$

$$= v(\pi_e)^2 + \mathbb{V}_{D_{\hat{R}}\sim\mathcal{D}}[\hat{R}(d,\pi_e)] \tag{74}$$

Thus,

$$(2) - (2') = \mathbb{V}_{s \sim d_0} \left[ v^{\pi_e}(s) \right] + \mathbb{E}_{s \sim d_0} \left[ \mathbb{V}_{D_{\hat{R}} \sim \mathcal{D}} [\hat{R}(s, \pi_e)] \right] - \mathbb{V}_{D_{\hat{R}} \sim \mathcal{D}} \left[ \hat{R}(d, \pi_e) \right] \tag{75}$$

Putting everything together, we have:

$$\mathbb{V}_{D_{\hat{R}} \sim \mathcal{D}, D_{\hat{R}} \sim \mathcal{D}} [\hat{V}^{\mathrm{DM}}] = (1) + (1') \tag{76}$$

$$= \mathbb{V}_{D_{\hat{R}} \sim \mathcal{D}} \left[ \hat{R}(d, \pi_e) \right] \tag{77}$$

$$+ \frac{1}{N} \left( \mathbb{V}_{s \sim d_0} \left[ v^{\pi_e}(s) \right] + \mathbb{E}_{s \sim d_0} \left[ \mathbb{V}_{D_{\hat{R}} \sim \mathcal{D}} [\hat{R}(s, \pi_e)] \right] - \mathbb{V}_{D_{\hat{R}} \sim \mathcal{D}} \left[ \hat{R}(d, \pi_e) \right] \right) \tag{78}$$

$$= \frac{1}{N} \mathbb{V}_{s \sim d_0} \left[ v^{\pi_e}(s) \right] + \frac{1}{N} \mathbb{E}_{s \sim d_0} \left[ \mathbb{V}_{D_{\hat{R}} \sim \mathcal{D}} [\hat{R}(s, \pi_e)] \right] + \left( 1 - \frac{1}{N} \right) \mathbb{V}_{D_{\hat{R}} \sim \mathcal{D}} \left[ \hat{R}(d, \pi_e) \right] \tag{79}$$

$$= \frac{1}{N} \mathbb{V}_{s \sim d_0} \left[ v^{\pi_e}(s) \right] + \frac{1}{N} \mathbb{E}_{s \sim d_0} \left[ \mathbb{E}_{a \sim \pi_e(\cdot|s)} \left[ \pi_e(a|s) \, \sigma_R^2(s, a) \, \mathbb{E}_{D_{\hat{R}} \sim \mathcal{D}} \left[ \frac{1}{N_{s,a}(D_{\hat{R}})} \right] \right] \right] \tag{80}$$

$$+ \left( 1 - \frac{1}{N} \right) \mathbb{E}_{s \sim d_0} \mathbb{E}_{a \sim \pi_e(\cdot|s)} \left[ d(s) \, \pi_e(a|s) \, \sigma_R^2(s, a) \, \mathbb{E}_{D_{\hat{R}} \sim \mathcal{D}} \left[ \frac{1}{N_{s,a}(D_{\hat{R}})} \right] \right] \tag{81}$$

$$= \frac{1}{N} \mathbb{V}_{s \sim d_0} \left[ v^{\pi_e}(s) \right] \tag{82}$$

$$+ \mathbb{E}_{s \sim d_0} \mathbb{E}_{a \sim \pi_e(\cdot|s)} \left[ \left( \frac{1}{N} + \left( 1 - \frac{1}{N} \right) d(s) \right) \pi_e(a|s) \, \sigma_R^2(s, a) \, \mathbb{E}_{D_{\hat{R}} \sim \mathcal{D}} \left[ \frac{1}{N_{s,a}(D_{\hat{R}})} \right] \right] \tag{83}$$

## G   Expectation and variance of doubly robust estimators under different annotation conditions

Now we derive the expectation and variance of the introduced doubly robust estimators under different annotation conditions. In the main text, we note that the annotations can be perfect (i.e. $\mathbb{E}[g_i] = \mathbb{E}[r_i]$, $\mathbb{V}[g_i] = \mathbb{V}[r_i]$), biased (i.e. $\mathbb{E}[g_i] = \mathbb{E}[r_i] + \epsilon_G(s_i, a_i)$), and have higher variance (i.e. $\mathbb{V}[g_i] = \mathbb{V}[r_i] + \Delta_G(s_i, a_i)$). In our proofs, we first derive the expectation and variance under the imperfect annotations condition. Then, we show that the perfect annotation condition is a special case of these derivations.

### G.1   Expectation and variance of DM$^+$-IS

The DR estimator is defined as

$$\hat{V}^{DM^+ - IS} = \frac{1}{N} \sum_{i=1}^{N} \left( \hat{R}^+(s_i, \pi_e) + \frac{\pi_e(a_i|s_i)}{\pi_b(a_i|s_i)} (r_i - \hat{R}^+(s_i, a_i)) \right)$$

#### G.1.1   Expectation

We now prove Proposition 1, which is restated below.

**Proposition** (Expectation of DM$^+$-IS under imperfect annotations). *Under Assumptions 1 and 3,* $\mathbb{E}[\hat{V}^{\mathrm{DM}^+\text{-IS}}] = v(\pi_e)$.

Proof: The expectation is taken over the dataset $D_0$, which is used to fit the OPE estimate, and $D_{\hat{R}}$, which is used to learn the reward function estimate. We first use the linearity of expectation.

$$\mathbb{E}_{D_0, D_{\hat{R}} \sim \mathcal{D}} [\hat{V}^{DM^+ - IS}] \tag{84}$$

$$= \mathbb{E}_{\substack{D_{\hat{R}}, s \sim d_0 \\ a \sim \pi_b(\cdot|s), r \sim R(s,a)}} \left[ \frac{1}{N} \sum_{i=1}^{N} \left( \hat{R}^+(s_i, \pi_e) + \frac{\pi_e(a_i|s_i)}{\pi_b(a_i|s_i)} (r_i - \hat{R}^+(s_i, a_i)) \right) \right] \tag{85}$$

$$= \frac{1}{N} \sum_{i=1}^{N} \mathbb{E}_{\substack{D_{\hat{R}}, s \sim d_0 \\ a \sim \pi_b(\cdot|s), r \sim R(s,a)}} \left[ \left( \hat{R}^+(s_i, \pi_e) + \frac{\pi_e(a_i|s_i)}{\pi_b(a_i|s_i)} (r_i - \hat{R}^+(s_i, a_i)) \right) \right] \tag{86}$$

We now split the expectation into two terms:

$$= \frac{1}{N} \sum_{i=1}^{N} \mathbb{E}_{D_{\hat{R}}, s \sim d_0} [\hat{R}^+(s_i, \pi_e)] + \mathbb{E}_{\substack{D_{\hat{R}}, s \sim d_0 \\ a \sim \pi_b(\cdot|s), r \sim R(s,a)}} \left[ \frac{\pi_e(a_i|s_i)}{\pi_b(a_i|s_i)} (r_i - \hat{R}^+(s_i, a_i)) \right] \tag{87}$$

$$= \frac{1}{N} \sum_{i=1}^{N} \mathbb{E}_{D_{\hat{R}}, s \sim d_0} [\hat{R}^+(s_i, \pi_e)] + \mathbb{E}_{\substack{D_{\hat{R}}, s \sim d_0 \\ r \sim R(s,a)}} \left[ \pi_b(a_i|s_i) \frac{\pi_e(a_i|s_i)}{\pi_b(a_i|s_i)} (r_i - \hat{R}^+(s_i, a_i)) \right] \tag{88}$$

$$= \frac{1}{N} \sum_{i=1}^{N} \mathbb{E}_{s \sim d_0} [\hat{R}^+(s_i, \pi_e)] + \mathbb{E}_{s \sim d_0 a \sim \pi_e(\cdot|s), r \sim R(s,a)} \left[ (r_i - \hat{R}^+(s_i, a_i)) \right] \tag{89}$$

$$= \frac{1}{N} \sum_{i=1}^{N} \mathbb{E}_{D_{\hat{R}}, s \sim d_0} [\hat{R}^+(s_i, \pi_e)] + \mathbb{E}_{D_{\hat{R}}, s \sim d_0 a \sim \pi_e(\cdot|s)} \left[ (\bar{R}(s_i, a_i) - \hat{R}^+(s_i, a_i)) \right] \tag{90}$$

$$= \frac{1}{N} \sum_{i=1}^{N} \mathbb{E}_{s \sim d_0 a \sim \pi_e} [\bar{R}(s_i, a_i)] \tag{91}$$

$$= v(\pi_e) \tag{92}$$

The expectation of DM$^+$-IS when the annotations are biased is the value of the target policy. The variance of the annotation has no effect on the expectation of this estimator.

**Proposition 12** (Unbiasedness of DM$^+$-IS)**.** *If both Assumptions 2 and 5 hold, the* DM$^+$-IS *estimator is unbiased,* $\mathbb{E}[\hat{V}^{\mathrm{DM}^+\text{-IS}}] = v(\pi_e)$.

Proof: Under imperfect annotations, the estimator is an unbiased estimator of the value of the target policy. If we have additional assumptions about perfect annotations, this estimator is also unbiased.

### G.1.2 Variance

We now prove Theorem 3, which is restated below.

**Theorem** (Variance of DM$^+$-IS under annotations with higher variance)**.** *Under Assumption 4 and Assumption 1,*

$$N \cdot \mathbb{V}[\hat{V}^{\mathrm{DM}^+\text{-IS}}] = \mathbb{V}_{s \sim d_0}[v^{\pi_e}(s)] + \mathbb{E}_{s \sim d_0} \mathbb{E}_{a \sim \pi_b(s)} [\rho_s(a)^2 \sigma_R^2(s, a)]$$

$$+ \mathbb{E}_{s \sim d_0} \left[ \mathbb{E}_{a \sim \pi_b} [\rho_s(a)^2 \varepsilon_{\hat{R}^+}(s, a)^2] - \varepsilon_{\hat{R}^+}^{\pi_e}(s)^2 \right] + \mathbb{E}_{s \sim d_0} \mathbb{E}_{a \sim \pi_b} \left[ \left( \rho_s(a)^2 - \frac{1}{\pi_b(a|s)} \right) \Delta_{\hat{R}^+}(s, a) \right]$$

*where* $\varepsilon_{\hat{R}}^+(s, a) = \mathbb{E}_{D_{\hat{R}^+}} [\hat{R}^+(s, a)] - \bar{R}(s, a)$, $\varepsilon_{\hat{R}^+}^{\pi_e}(s) = \mathbb{E}_{a \sim \pi_e} [\varepsilon_{\hat{R}^+}(s, a)]$, *and* $\Delta_{\hat{R}^+}(s, a) = \mathbb{V}_{D_{\hat{R}^+}} [\hat{R}^+(s, a)]$.

We use the law of total variance, the earlier expectation term, independence of samples, and the law of total variance.

$$\mathbb{V}_{D_0, D_{\hat{R}^+}} [\hat{V}^{\mathrm{DM}^+\text{-IS}}] = \mathbb{V}_{D_0, D_{\hat{R}^+}} [\frac{1}{N} \sum_{i=1}^{N} \hat{R}^+(s_i, \pi_e) + \rho_{s_i}(a_i)(r_i - \hat{R}^+(s_i, a_i))] \tag{93}$$

$$= \mathbb{E}_{D_{\hat{R}^+}} [\mathbb{V}_{D_0} [\frac{1}{N} \sum_{i=1}^{N} \hat{R}^+(s_i, \pi_e) + \rho_{s_i}(a_i)(r_i - \hat{R}^+(s_i, a_i))]] \tag{94}$$

$$+ \mathbb{V}_{D_{\hat{R}^+}} [\mathbb{E}_{D_0} [\frac{1}{N} \sum_{i=1}^{N} \hat{R}^+(s_i, \pi_e) + \rho_{s_i}(a_i)(r_i - \hat{R}^+(s_i, a_i))]] \tag{95}$$

$$= \mathbb{E}_{D_{\hat{R}^+}} [\frac{1}{N} \sum_{i=1}^{N} \mathbb{V}_{s_i \sim d_0 a_i \sim \pi_b, r_i \sim R(s_i, a_i)} [\hat{R}^+(s_i, \pi_e) + \rho_{s_i}(a_i)(r_i - \hat{R}^+(s_i, a_i))]] \tag{96}$$

$$= \mathbb{E}_{D_{\hat{R}^+}}[\frac{1}{N}\sum_{i=1}^{N}\mathbb{E}_{s_i\sim d_0}[\mathbb{V}_{a_i\sim\pi_b, r_i\sim R(s_i,a_i)}[\hat{R}^+(s_i,\pi_e) + \rho_{s_i}(a_i)(r_i - \hat{R}^+(s_i,a_i))]] \tag{97}$$

$$+ \mathbb{V}_{s_i\sim d_0}[\mathbb{E}_{a_i\sim\pi_b, r_i\sim R(s_i,a_i)}[\hat{R}^+(s_i,\pi_e) + \rho_{s_i}(a_i)(r_i - \hat{R}^+(s_i,a_i))]]] \tag{98}$$

$$= \mathbb{E}_{D_{\hat{R}^+}}[\frac{1}{N}\sum_{i=1}^{N}\underbrace{\mathbb{E}_{s_i\sim d_0}[\mathbb{V}_{a_i\sim\pi_b, r_i\sim R(s_i,a_i)}[\hat{R}^+(s_i,\pi_e) + \rho_{s_i}(a_i)(r_i - \hat{R}^+(s_i,a_i))]]}_{1} \tag{99}$$

$$+ \mathbb{V}_{s_i\sim d_0}[v(\pi_e)]] \tag{100}$$

$$\text{Term } 1 = \mathbb{E}_{D_{\hat{R}^+}}[\frac{1}{N}\sum_{i=1}^{N}\mathbb{E}_{s_i\sim d_0}[\mathbb{E}_{a_i\sim\pi_b}[\mathbb{V}_{r_i\sim R(s_i,a_i)}[\hat{R}^+(s_i,\pi_e) + \rho_{s_i}(a_i)(r_i - \hat{R}^+(s_i,a_i))]] \tag{101}$$

$$+ \mathbb{E}_{s_i\sim d_0}[\mathbb{V}_{a_i\sim\pi_b}[\mathbb{E}_{r_i\sim R(s_i,a_i)}[\hat{R}^+(s_i,\pi_e) + \rho_{s_i}(a_i)(r_i - \hat{R}^+(s_i,a_i))]] \tag{102}$$

$$= \mathbb{E}_{D_{\hat{R}^+}}[\frac{1}{N}\sum_{i=1}^{N}\mathbb{E}_{s_i\sim d_0}[\mathbb{E}_{a_i\sim\pi_b}[\rho_{s_i}(a_i)^2\sigma_R^2]] \tag{103}$$

$$+ \mathbb{E}_{s_i\sim d_0}[\mathbb{V}_{a_i\sim\pi_b}[\rho_{s_i}(a_i)(\bar{R} - \hat{R}^+)]] \tag{104}$$

The total variance term is now:

$$= \frac{1}{N}\sum_{i=1}^{N}\mathbb{V}_{s_i\sim d_0}[v(\pi_e)]] + \mathbb{E}_{s_i\sim d_0}[\mathbb{E}_{a_i\sim\pi_b}[\rho_{s_i}(a_i)^2\sigma_R^2]] \tag{105}$$

$$+ \mathbb{E}_{D_{\hat{R}^+}}[\mathbb{E}_{s_i\sim d_0}[\mathbb{V}_{a_i\sim\pi_b}[\rho_{s_i}(a_i)(\bar{R} - \hat{R}^+)]]] \tag{106}$$

We simplify the last term, the only term that involves $D_{\hat{R}^+}$.

$$\mathbb{E}_{D_{\hat{R}^+}}[\mathbb{E}_{s_i\sim d_0}[\mathbb{V}_{a_i\sim\pi_b}[\rho_{s_i}(a_i)(\bar{R} - \hat{R}^+)]]] \tag{107}$$

$$= \mathbb{E}_{D_{\hat{R}^+}}[\mathbb{E}_{s_i\sim d_0}[\mathbb{E}_{a_i\sim\pi_b}[\rho_{s_i}(a_i)^2(\bar{R} - \hat{R}^+)^2] - \mathbb{E}_{a_i\sim\pi_b}[\rho_{s_i}(a_i)(\bar{R} - \hat{R}^+)]^2]] \tag{108}$$

$$= \underbrace{\mathbb{E}_{D_{\hat{R}^+}}\mathbb{E}_{s_i\sim d_0}\mathbb{E}_{a_i\sim\pi_b}[\rho_{s_i}(a_i)^2(\bar{R} - \hat{R}^+)^2]}_{1} - \underbrace{\mathbb{E}_{D_{\hat{R}^+}}\mathbb{E}_{s_i\sim d_0}[\mathbb{E}_{a_i\sim\pi_b}[\rho_{s_i}(a_i)(\bar{R} - \hat{R}^+)]^2]}_{2} \tag{109}$$

$$\text{Term } 1 = \mathbb{E}_{s\sim d_0}\mathbb{E}_{a\sim\pi_b}[\rho_s(a)^2\mathbb{E}_{D_{\hat{R}^+}}[(\bar{R} - \hat{R}^+)^2]] \tag{110}$$

$$= \mathbb{E}_{s\sim d_0}\mathbb{E}_{a\sim\pi_b}[\rho_s(a)^2\mathbb{E}_{D_{\hat{R}^+}}[\bar{R}(s,a)^2 - 2\bar{R}(s,a)\hat{R}^+(s,a) + \hat{R}(s,a)^2]] \tag{111}$$

$$= \mathbb{E}_{s\sim d_0}\mathbb{E}_{a\sim\pi_b}\Big[\rho_s(a)^2\big(\bar{R}(s,a)^2 - 2\bar{R}(s,a)\tilde{R}(s,a) + \tilde{R}(s,a)^2 + \mathbb{V}_{D_{\hat{R}^+}}[\hat{R}^+(s,a)]\big)\Big] \tag{112}$$

where $\tilde{R}(s,a) = \mathbb{E}_{D_{\hat{R}^+}}[\hat{R}^+(s,a)]$ \hfill (113)

$$\text{Term } 2 = \mathbb{E}_{D_{\hat{R}^+}}\mathbb{E}_{s\sim d_0}[\mathbb{E}_{a\sim\pi_e}[\bar{R}(s,a) - \hat{R}^+(s,a)]^2] \tag{114}$$

$$= \mathbb{E}_{s\sim d_0}\mathbb{E}_{D_{\hat{R}^+}}\Big[\big(V^{\pi_e}(s) - \hat{R}^+(s,\pi_e)\big)^2\Big] \tag{115}$$

$$= \mathbb{E}_{s\sim d_0}\mathbb{E}_{D_{\hat{R}^+}}\Big[V^{\pi_e}(s)^2 - 2V^{\pi_e}(s)\hat{R}^+(s,\pi_e) + \hat{R}^+(s,\pi_e)^2\Big] \tag{116}$$

$$= \mathbb{E}_{s\sim d_0}\Big[V^{\pi_e}(s)^2 - 2V^{\pi_e}(s)\tilde{R}(s,\pi_e) + \tilde{R}(s,\pi_e)^2 + \mathbb{V}_{D_{\hat{R}^+}}[\hat{R}^+(s,\pi_e)]\Big] \tag{117}$$

where $\tilde{R}(s,\pi_e) = \mathbb{E}_{D_{\hat{R}^+}}[\hat{R}^+(s,\pi_e)]$ \hfill (118)

Term 1 - Term 2 = \hfill (119)

$$\mathbb{E}_{s\sim d_0}\mathbb{E}_{a\sim\pi_b}\Big[\rho_s(a)^2\big(\bar{R}(s,a)^2 - 2\bar{R}(s,a)\tilde{R}(s,a) + \tilde{R}(s,a)^2 + \mathbb{V}_{D_{\hat{R}^+}}[\hat{R}^+(s,a)]\big)\Big] \tag{120}$$

$$- \mathbb{E}_{s\sim d_0}\Big[V^{\pi_e}(s)^2 - 2V^{\pi_e}(s)\tilde{R}(s,\pi_e) + \tilde{R}(s,\pi_e)^2 + \mathbb{V}_{D_{\hat{R}^+}}[\hat{R}^+(s,\pi_e)]\Big] \tag{121}$$

If the DM estimate is unbiased, we have $\tilde{R}(s,a) = \mathbb{E}_{D_{\hat{R}^+}}[\hat{R}^+(s,a)] = \bar{R}(s,a)$. Substituting into the expression above results in cancellations and the expression becomes:

$$\mathbb{E}_{s\sim d_0}\Big[\mathbb{E}_{a\sim\pi_b}[\rho_s(a)^2\mathbb{V}_{D_{\hat{R}^+}}[\hat{R}^+(s,a)]] - \mathbb{V}_{D_{\hat{R}^+}}[\hat{R}^+(s,\pi_e)]\Big] \tag{122}$$

$$= \mathbb{E}_{s \sim d_0} \Big[ \sum_a \pi_e(a|s)^2 \frac{1}{\pi_b(a|s)} \mathbb{V}_{D_{\hat{R}^+}}[\hat{R}^+(s,a)] - \sum_a \pi_e(a|s)^2 \mathbb{V}_{D_{\hat{R}^+}}[\hat{R}^+(s,a)] \Big] \tag{123}$$

$$= \mathbb{E}_{s \sim d_0} \Big[ \sum_a \pi_e(a|s)^2 \big( \frac{1}{\pi_b(a|s)} - 1 \big) \mathbb{V}_{D_{\hat{R}^+}}[\hat{R}^+(s,a)] \Big] \tag{124}$$

$$= \mathbb{E}_{s \sim d_0} \mathbb{E}_{a \sim \pi_b} \Big[ \big( \rho_s(a)^2 - \tfrac{1}{\pi_b(a|s)} \big) \mathbb{V}_{D_{\hat{R}^+}}[\hat{R}^+(s,a)] \Big] \tag{125}$$

If the DM estimate is biased, let's say $\tilde{R}(s,a) = \mathbb{E}_{D_{\hat{R}^+}}[\hat{R}^+(s,a)] = \bar{R}(s,a) + \varepsilon_R(s,a)$, those terms no longer cancel and we get additional terms as follows:

$$\mathbb{E}_{s \sim d_0} \mathbb{E}_{a \sim \pi_b} \Big[ \rho_s(a)^2 \big( \bar{R}(s,a)^2 - 2\bar{R}(s,a)\tilde{R}(s,a) + \tilde{R}(s,a)^2 \big) \Big] \tag{126}$$

$$- \mathbb{E}_{s \sim d_0} \Big[ V^{\pi_e}(s)^2 + 2V^{\pi_e}(s)^2 \tilde{R}(s,\pi_e) + \tilde{R}(s,\pi_e)^2 \Big] \tag{127}$$

$$= \mathbb{E}_{s \sim d_0} \Big[ \mathbb{E}_{a \sim \pi_b}[\rho_s(a)^2 \varepsilon(s,a)^2] - \varepsilon^{\pi_e}(s)^2 \Big] \tag{128}$$

$$\text{where } \varepsilon^{\pi_e}(s) = \mathbb{E}_{D_{\hat{R}^+}}[\hat{R}(s,\pi_e)] - V^{\pi_e}(s) \tag{129}$$

$$= \mathbb{E}_{s \sim d_0} \Big[ \sum_a \pi_e(a|s)^2 \frac{1}{\pi_b(a|s)} \varepsilon(s,a)^2 - \Big( \sum_a \pi_e(a|s)\varepsilon(s,a) \Big)^2 \Big] \tag{130}$$

If we assume the bias of DM is constant across all $(s,a)$ such that $\varepsilon(s,a) = \varepsilon$, then the above expression becomes

$$= \varepsilon^2 \mathbb{E}_{s \sim d_0} \Big[ \big( \sum_a \pi_e(a|s)^2 \frac{1}{\pi_b(a|s)} \big) - 1 \Big] \tag{131}$$

$$\geq 0 \text{ by Sedrakyan's lemma, a special case of Cauchy-Schwarz inequality} \tag{132}$$

The whole variance term is now:

$$N \cdot \mathbb{V}[\hat{V}^{DM^+ - IS}] = \mathbb{V}_{s \sim d_0}[v^{\pi_e}(s)] + \mathbb{E}_{s \sim d_0}\mathbb{E}_{a \sim \pi_b(s)}[\rho_s(a)^2 \sigma_R^2(s,a)] \tag{133}$$

$$+ \mathbb{E}_{s \sim d_0} \Big[ \mathbb{E}_{a \sim \pi_b}[\rho_s(a)^2 \varepsilon(s,a)^2] - \varepsilon^{\pi_e}(s)^2 \Big] \tag{134}$$

$$+ \mathbb{E}_{s \sim d_0}\mathbb{E}_{a \sim \pi_b} \Big[ \big( \rho_s(a)^2 - \tfrac{1}{\pi_b(a|s)} \big) \mathbb{V}_{D_{\hat{R}^+}}[\hat{R}^+(s,a)] \Big] \tag{135}$$

In the case that the annotations are perfect, the whole variance term is altered to reflect the fact that $\varepsilon = 0$.

**Proposition 13** (Variance of DM$^+$-IS)**.** *If Assumptions 2 and 5 hold,*

$$N \cdot \mathbb{V}[\hat{V}^{DM^+ - IS}] = \mathbb{V}_{s \sim d_0}[v^{\pi_e}(s)] + \mathbb{E}_{s \sim d_0}\mathbb{E}_{a \sim \pi_b(s)}[\rho_s(a)^2 \sigma_R^2(s,a)] \tag{136}$$

$$+ \mathbb{E}_{s \sim d_0}\mathbb{E}_{a \sim \pi_b} \Big[ \big( \rho_s(a)^2 - \tfrac{1}{\pi_b(a|s)} \big) \mathbb{V}_{D_{\hat{R}^+}}[\hat{R}^+(s,a)] \Big] \tag{137}$$

## G.2 Expectation and variance of DM-IS$^+$

The DR estimator is defined as

$$\hat{V}^{\text{DM-IS}^+} = \frac{1}{N} \sum_{i=1}^N \Big( \hat{R}(s_i, \pi_e) + \sum_{a \in A} w_i^a \frac{\pi_e(a|s_i)}{\pi_{b^+}(a|s_i)} (c_i^a - \hat{R}(s_i, a)) \Big)$$

### G.2.1 Expectation

The expectation of the estimator assuming biased annotations is summarized in Theorem 2 which is restated below.

**Theorem** (Expectation of DM-IS$^+$ and DM$^+$-IS$^+$ under imperfect annotations). *Under Assumptions 3 and 5, the two estimators have the same expectation:*

$$\mathbb{E}[\hat{V}^{\text{DM-IS}^+}] = \mathbb{E}[\hat{V}^{\text{DM}^+\text{-IS}^+}] = v(\pi_e) + \mathbb{E}_{s_i \sim d_0}[\mathbb{E}_{a \sim \pi_e(s_i)}[(1 - \frac{\bar{W}(a|s_i,a)\pi_b(a|s_i)}{\pi_b^+(a|s_i)})\epsilon_G(s_i,a)]]$$

Proof: We use the linearity of expectation, and the definition of the expectation.

$$\mathbb{E}_{D_0, D_{\hat{R}} \sim \mathcal{D}}[\hat{V}^{\text{DM-IS}^+}] \tag{138}$$

$$= \mathbb{E}_{\substack{D_{\hat{R}}, s_i \sim d_0 a_i \sim \pi_b(s_i), \\ r \sim R(s_i, a_i), w \sim W(s_i, a_i)}} \left[\frac{1}{N}\sum_{i=1}^{N}\left(\hat{R}(s_i, \pi_e) + \sum_{a \in A} w_i^a \frac{\pi_e(a|s_i)}{\pi_{b^+}(a|s_i)}(c_i^a - \hat{R}(s_i, a))\right)\right] \tag{139}$$

$$= \mathbb{E}_{\substack{D_{\hat{R}}, s_i \sim d_0 a_i \sim \pi_b(s_i), \\ r \sim R(s_i, a_i), w \sim W(s_i, a_i), \\ g_i^a \sim G(s_i, a)}} [\frac{1}{N}\sum_{i=1}^{N}(\hat{R}(s_i, \pi_e) + w_i^{a_i} \frac{\pi_e(a_i|s_i)}{\pi_{b^+}(a_i|s_i)}(r_i - \hat{R}(s_i, a_i)) \tag{140}$$

$$+ \sum_{a \in A\setminus\{a_i\}} w_i^a \frac{\pi_e(a|s_i)}{\pi_{b^+}(a|s_i)}(g_i^a - \hat{R}(s_i, a)))] \tag{141}$$

$$= \frac{1}{N}\sum_{i=1}^{N}\mathbb{E}_{D_{\hat{R}}, s_i \sim d_0}[\hat{R}(s_i, \pi_e)] \tag{142}$$

$$+ \underbrace{\mathbb{E}_{D_{\hat{R}}, s_i \sim d_0 a_i \sim \pi_b(s_i), r_i \sim R(s_i, a_i), w \sim W(s_i, a_i)}[w_i^{a_i} \frac{\pi_e(a_i|s_i)}{\pi_{b^+}(a_i|s_i)}(r_i - \hat{R}(s_i, a_i))]}_{2} \tag{143}$$

$$+ \underbrace{\mathbb{E}_{D_{\hat{R}}, s_i \sim d_0 a_i \sim \pi_b(s_i), w \sim W(s_i, a_i), g_i^a \sim G(s_i, a)}[\sum_{a \in A\setminus\{a_i\}} w_i^a \frac{\pi_e(a|s_i)}{\pi_{b^+}(a|s_i)}(g_i^a - \hat{R}(s_i, a))]}_{3} \tag{144}$$

Now we simplify term 2 using the expectation of the weights and the expectation of scalar reward, and the definition of expectation:

$$\text{Term 2} = \mathbb{E}_{D_{\hat{R}}, s \sim d_0 a_i \sim \pi_b, r \sim R, w \sim W}[w_i^{a_i} \frac{\pi_e(a_i|s_i)}{\pi_{b^+}(a_i|s_i)}(r_i - \hat{R}(s_i, a_i))] \tag{145}$$

$$= \mathbb{E}_{D_{\hat{R}}, s \sim d_0 a_i \sim \pi_b, r \sim R}[\mathbb{E}_{w \sim W}[w_i^{a_i}] \frac{\pi_e(a_i|s_i)}{\pi_{b^+}(a_i|s_i)}(\mathbb{E}_{r \sim R}[r_i] - \hat{R}(s_i, a_i))] \tag{146}$$

$$= \mathbb{E}_{D_{\hat{R}}, s \sim d_0 a_i \sim \pi_b, r \sim R}[\bar{W}(a_i|s_i, a_i) \frac{\pi_e(a_i|s_i)}{\pi_{b^+}(a_i|s_i)}(\bar{R}(s_i, a_i) - \hat{R}(s_i, a_i))] \tag{147}$$

$$\tag{148}$$

Now we simplify term 3 using a similar procedure:

$$\text{Term 3} = \mathbb{E}_{D_{\hat{R}}, s \sim d_0 a \sim \pi_b, w \sim W, g \sim G}[\sum_{a \in A\setminus\{a_i\}} w_i^a \frac{\pi_e(a|s_i)}{\pi_{b^+}(a|s_i)}(g_i^a - \hat{R}(s_i, a))] \tag{149}$$

$$= \mathbb{E}_{D_{\hat{R}}, s \sim d_0 a \sim \pi_b}[\sum_{a \in A\setminus\{a_i\}} \mathbb{E}_{w \sim W}[w_i^a] \frac{\pi_e(a|s_i)}{\pi_{b^+}(a|s_i)}(\mathbb{E}_{g \sim G}[g_i^a] - \hat{R}(s_i, a))] \tag{150}$$

$$= \mathbb{E}_{D_{\hat{R}}, s \sim d_0 a \sim \pi_b}[\sum_{a \in A\setminus\{a_i\}} \bar{W}(a|s_i, a) \frac{\pi_e(a|s_i)}{\pi_{b^+}(a|s_i)}(\bar{R}(s_i, a) + \epsilon_G - \hat{R}(s_i, a))] \tag{151}$$

$$\tag{152}$$

Putting together the terms:

$$\mathbb{E}_{s \sim d_0}[\hat{R}(s_i, \pi_e)] + \mathbb{E}_{s \sim d_0 a_i \sim \pi_b, r \sim R}[\bar{W}(a_i|s_i, a_i) \frac{\pi_e(a_i|s_i)}{\pi_{b^+}(a_i|s_i)}(\bar{R}(s_i, a_i) - \hat{R}(s_i, a_i))] \tag{153}$$

$$+ \mathbb{E}_{s \sim d_0 a \sim \pi_b}[\sum_{a \in A \setminus \{a_i\}} \bar{W}(a|s_i, a) \frac{\pi_e(a|s_i)}{\pi_{b^+}(a|s_i)} (\bar{R}(s_i, a) + \epsilon_G - \hat{R}(s_i, a))] \tag{154}$$

$$\tag{155}$$

Note that we can simplify the terms:

$$= \mathbb{E}_{D_{\hat{R}}, s \sim d_0}[\hat{R}(s_i, \pi_e)] \tag{156}$$

$$+ \mathbb{E}_{D_{\hat{R}}, s \sim d_0 a_i \sim \pi_b, r \sim R}[\sum_{a \in A} \bar{W}(a_i|s_i, a_i) \frac{\pi_e(a_i|s_i)}{\pi_{b^+}(a_i|s_i)} (\bar{R}(s_i, a_i) - \hat{R}(s_i, a_i))] \tag{157}$$

$$+ \mathbb{E}_{D_{\hat{R}}, s \sim d_0 a \sim \pi_b}[\sum_{a \in A \setminus \{a_i\}} \bar{W}(a|s_i, a) \frac{\pi_e(a|s_i)}{\pi_{b^+}(a|s_i)} (\epsilon_G)] \tag{158}$$

The sum of the first two terms becomes the value of the target policy:

$$\mathbb{E}_{D_{\hat{R}}, s \sim d_0}[\hat{R}(s_i, \pi_e)] + \mathbb{E}_{D_{\hat{R}}, s \sim d_0 a_i \sim \pi_b, r \sim R}[\sum_{a \in A} \bar{W}(a_i|s_i, a_i) \frac{\pi_e(a_i|s_i)}{\pi_{b^+}(a_i|s_i)} (\bar{R}(s_i, a_i) - \hat{R}(s_i, a_i))] \tag{159}$$

$$= \mathbb{E}_{D_{\hat{R}}, s \sim d_0}[\hat{R}(s_i, \pi_e)] \tag{160}$$

$$+ \mathbb{E}_{D_{\hat{R}}, s \sim d_0 r \sim R}[\sum_{a_i \in A} \pi_b(a_i|s_i) \sum_{a \in A} \bar{W}(a_i|s_i, a_i) \frac{\pi_e(a_i|s_i)}{\pi_{b^+}(a_i|s_i)} (\bar{R}(s_i, a_i) - \hat{R}(s_i, a_i))] \tag{161}$$

$$= \mathbb{E}_{D_{\hat{R}}, s \sim d_0}[\hat{R}(s_i, \pi_e)] + \mathbb{E}_{D_{\hat{R}}, s \sim d_0 r \sim R}[\pi_e(a_i|s_i)(\bar{R}(s_i, a_i) - \hat{R}(s_i, a_i))] \tag{162}$$

$$= \mathbb{E}_{D_{\hat{R}}, s \sim d_0}[\hat{R}(s_i, \pi_e)] + \mathbb{E}_{s \sim d_0 a_i \sim \pi_e, r \sim R}[(\bar{R}(s_i, a_i) - \hat{R}(s_i, a_i))] \tag{163}$$

$$= \mathbb{E}_{s \sim d_0 a_i \sim \pi_e}[(\bar{R}(s_i, a_i)] \tag{164}$$

$$= v(\pi_e) \tag{165}$$

The final term can be simplified as follows:

$$\mathbb{E}_{D_{\hat{R}}, s \sim d_0 a \sim \pi_b}[\sum_{\tilde{a} \in A \setminus \{a_i\}} \bar{W}(\tilde{a}|s_i, a) \frac{\pi_e(\tilde{a}|s_i)}{\pi_{b^+}(\tilde{a}|s_i)} \epsilon_G] \tag{166}$$

$$= \mathbb{E}_{D_{\hat{R}}, s \sim d_0}[\sum_{a \in A} \pi_b(a|s_i) \left( \sum_{\tilde{a} \in A \setminus \{a_i\}} \bar{W}(\tilde{a}|s_i, a) \frac{\pi_e(\tilde{a}|s_i)}{\pi_{b^+}(\tilde{a}|s_i)} \epsilon_G \right)] \tag{167}$$

$$= \mathbb{E}_{D_{\hat{R}}, s \sim d_0}[\sum_{\tilde{a} \in A} ((\sum_{a \in A} \pi_b(a|s) \bar{W}(\tilde{a}|s, a)) \frac{\pi_e(\tilde{a}|s)}{\pi_b^+(\tilde{a}|s} \epsilon_G))] \tag{168}$$

$$- \mathbb{E}_{D_{\hat{R}}, s \sim d_0}[(\sum_{a \in A} \pi_b(a|s) \bar{W}(a|s, a) \frac{\pi_e(a|s)}{\pi_b^+(a|s)} \epsilon_G)] \tag{169}$$

$$= \mathbb{E}_{D_{\hat{R}}, s \sim d_0}[\sum_{\tilde{a} \in A} \pi_e(\tilde{a}|s) \epsilon_G))] \tag{170}$$

$$- \mathbb{E}_{D_{\hat{R}}, s \sim d_0}[(\sum_{a \in A} \pi_e(a|s) \bar{W}(a|s, a) \frac{\pi_b(a|s)}{\pi_b^+(a|s)} \epsilon_G)] \tag{171}$$

$$= \mathbb{E}_{s \sim d_0}[\mathbb{E}_{a \sim \pi_e}[\left( 1 - \frac{\bar{W}(a|s, a) \pi_b(a|s)}{\pi_b^+(a|s)} \right) \epsilon_G]] \tag{172}$$

Thus, the final term for the expectation is:

$$v(\pi_e) + \mathbb{E}_{s \sim d_0} \left[ \mathbb{E}_{a \sim \pi_e} \left[ \left( 1 - \frac{\bar{W}(a|s, a) \pi_b(a|s)}{\pi_b^+(a|s)} \right) \epsilon_G \right] \right] \tag{173}$$

**Proposition 14** (Unbiasedness of DM-IS$^+$). *If both Assumptions 2 and 5 hold, the* DM-IS$^+$ *estimator is unbiased,* $\mathbb{E}[\hat{V}^{\text{DM-IS}^+}] = v(\pi_e)$.

Proof: If $\epsilon_G = 0$, then, the expectation term for the imperfect annotations case reduces to the value of the expert policy $v(\pi_e)$.

### G.2.2   Variance

We first derive the variance of this estimator in the setting in which we have perfect annotations.

**Proposition 15** (Variance of DM-IS$^+$ under perfect annotations). *If Assumptions 2 and 5 hold,*

$$\mathbb{V}_{D\sim\mathcal{D}}[\hat{V}^{\text{DM-IS}^+}] = \frac{1}{N^2} \sum_{i=1}^{N} \mathbb{V}_{s_i\sim d_0}[\hat{V}^{\pi_e}(s_i)] \tag{174}$$

$$+ \mathbb{E}_{s_i\sim d_0}\left[\mathbb{V}_{a_i\sim\pi_b(\cdot|s_i)}\left[\sum_{a\in A}\frac{\pi_e(a|s)}{\pi_b^+(a|s)}\bar{W}(a|s_i,a_i)(\bar{R}(s_i,a)-\hat{R}(s_i,a))\Big|s_i\right]\right] \tag{175}$$

$$+ \mathbb{E}_{s_i\sim d_0}\left[\mathbb{E}_{a_i\sim\pi_b}\left[\sum_{a\in A}\bar{W}(a|s_i,a_i)^2\frac{\pi_e(a|s)^2}{\pi_b^+(a|s)}\sigma_R(s_i,a)^2\Big|s_i\right]\right] \tag{176}$$

$$+ \mathbb{E}_{s_i\sim d_0}[\mathbb{E}_{a_i\sim\pi_b(\cdot|s_i)}[\sum_{a\in A}\frac{\pi_e(a|s_i)^2}{\pi_b^+(a|s_i)}\mathbb{V}_{w_i^a\sim W(s_i,a_i)}[w_i^a] \tag{177}$$

$$\times \left(\sigma_R(s_i,a)^2 + (\bar{R}(s_i,a)-\hat{R}(s_i,a)^2)\right) \tag{178}$$

$$+ 2\sum_{\tilde{a}_j,\tilde{a}_k}^{\tilde{a}_j\neq\tilde{a}_k}\frac{\pi_e(\tilde{a}_j|s_i)}{\pi_b^+(\tilde{a}_j|s_i)}\frac{\pi_e(\tilde{a}_k|s_i)}{\pi_b^+(\tilde{a}_k|s_i)} \tag{179}$$

$$\times Cov_{w_i^a\sim W(s_i,a)}(w_i^{\tilde{a}_j},w_i^{\tilde{a}_k})(\bar{R}(s_i,\tilde{a}_j)-\hat{R}(s_i,\tilde{a}_j))(\bar{R}(s_i,\tilde{a}_k)-\hat{R}(s_i,\tilde{a}_k))|a_i]|s_i]] \tag{180}$$

Proof: We start by moving the variance inside the summation because the variance of each term that the summation is over is independent. This is similar to what the IS$^+$ variance proof does.

$$\mathbb{V}_{D\sim\mathcal{D}}[\hat{V}^{\text{DM-IS}^+}] \tag{181}$$

$$= \mathbb{V}_{\substack{s_i\sim d_0 a_i\sim\pi_b(\cdot|s_i),\\c_i^a\sim R(s_i,a),w_i^a\sim W(s_i,a_i)}}\left[\frac{1}{N}\sum_{i=1}^{N}\left(\hat{R}(s_i,\pi_e)+\sum_{a\in A}w_i^a\frac{\pi_e(a|s_i)}{\pi_{b^+}(a|s_i)}(c_i^a-\hat{R}(s_i,a))\right)\right] \tag{182}$$

$$= \frac{1}{N^2}\sum_{i=1}^{N}\mathbb{V}_{\substack{s_i\sim d_0 a_i\sim\pi_b(\cdot|s_i),\\c_i^a\sim R(s_i,a),w_i^a\sim W(s_i,a_i)}}\left[\hat{R}(s_i,\pi_e)+\sum_{a\in A}w_i^a\frac{\pi_e(a|s_i)}{\pi_{b^+}(a|s_i)}(c_i^a-\hat{R}(s_i,a))\right] \tag{183}$$

Now, we use the law of total variance to separate the joint distribution that the variance is over:

$$= \frac{1}{N^2}\sum_{i=1}^{N}\mathbb{E}_{s_i\sim d_0}\left[\underbrace{\mathbb{V}_{\substack{a_i\sim\pi_b(\cdot|s_i),\\c_i^a\sim R(s_i,a),\\w_i^a\sim W(s_i,a_i)}}\left[\hat{R}(s_i,\pi_e)+\sum_{a\in A}w_i^a\frac{\pi_e(a|s_i)}{\pi_b^+(a|s_i)}(c_i^a-\hat{R}(s_i,a))\Big|s_i\right]}_{2}\right] \tag{184}$$

$$+ \mathbb{V}_{s_i\sim d_0}\left[\underbrace{\mathbb{E}_{\substack{a_i\sim\pi_b(\cdot|s_i),\\c_i^a\sim R(s_i,a),\\w_i^a\sim W(s_i,a_i)}}\left[\hat{R}(s_i,\pi_e)+\sum_{a\in A}w_i^a\frac{\pi_e(a|s_i)}{\pi_b^+(a|s_i)}(c_i^a-\hat{R}(s_i,a))\Big|s_i\right]}_{1}\right] \tag{185}$$

Note that Term 1 is just the value of the target policy conditioned on a specific $s_i$. This makes the whole variance now:

$$= \frac{1}{N^2} \sum_{i=1}^{N} \mathbb{E}_{s_i \sim d_0} \left[ \underbrace{\mathbb{V}_{\substack{a_i \sim \pi_b(\cdot|s_i), \\ c_i^a \sim R(s_i,a), \\ w_i^a \sim W(s_i,a_i)}} \left[ \hat{R}(s_i, \pi_e) + \sum_{a \in A} w_i^a \frac{\pi_e(a|s_i)}{\pi_b^+(a|s_i)} (c_i^a - \hat{R}(s_i, a)) \middle| s_i \right]}_{2} \right] \tag{186}$$

$$+ \mathbb{V}_{s_i \sim d_0} \left[ v^{\pi_e}(s_i) \right] \tag{187}$$

Now we decompose term 2 using the law of total variance:

$$\text{Term } 2 = \mathbb{E}_{s_i \sim d_0} \left[ \mathbb{V}_{\substack{a_i \sim \pi_b(\cdot|s_i), \\ c_i^a \sim R(s_i,a), w_i^a \sim W(s_i,a_i)}} \left[ \hat{R}(s_i, \pi_e) + \sum_{a \in A} w_i^a \frac{\pi_e(a|s_i)}{\pi_b^+(a|s_i)} (c_i^a - \hat{R}(s_i, a)) \middle| s_i \right] \right] \tag{188}$$

$$= \mathbb{E}_{s_i \sim d_0} [\mathbb{E}_{a_i \sim \pi_b(\cdot|s_i)} \underbrace{\left[ \mathbb{V}_{\substack{c_i^a \sim R(s_i,a), \\ w_i^a \sim W(s_i,a_i)}} , [\hat{R}(s_i, \pi_e) + \sum_{a \in A} w_i^a \frac{\pi_e(a|s_i)}{\pi_b^+(a|s_i)} (c_i^a - \hat{R}(s_i, a)) \middle| a_i] | s_i \right]}_{3}] \tag{189}$$

$$+ \underbrace{\mathbb{V}_{a_i \sim \pi_b(\cdot|s_i)} \left[ \mathbb{E}_{\substack{c_i^a \sim R(s_i,a), \\ w_i^a \sim W(s_i,a_i)}} , [\hat{R}(s_i, \pi_e) + \sum_{a \in A} w_i^a \frac{\pi_e(a|s_i)}{\pi_b^+(a|s_i)} (c_i^a - \hat{R}(s_i, a)) \middle| a_i] | s_i \right]}_{4} \tag{190}$$

Now we decompose term 4 by distributing the inner-most expectation. Note that the inner-most expectation only depends on the distribution over $c_i$, which is not in many of the terms. :

$$\text{Term } 4 = \mathbb{E}_{s_i \sim d_0} \left[ \mathbb{V}_{a_i \sim \pi_b(\cdot|s_i)} \left[ \mathbb{E}_{\substack{c_i^a \sim R(s_i,a), \\ w_i^a \sim W(s_i,a_i)}} , [\hat{R}(s_i, \pi_e) + \sum_{a \in A} w_i^a \frac{\pi_e(a|s_i)}{\pi_b^+(a|s_i)} (c_i^a - \hat{R}(s_i, a)) \middle| a_i] \middle| s_i \right] \right] \tag{191}$$

$$= \mathbb{E}_{s_i \sim d_0} \left[ \mathbb{V}_{a_i \sim \pi_b(\cdot|s_i)} \left[ \hat{R}(s_i, \pi_e) + \mathbb{E}_{\substack{c_i^a \sim R(s_i,a), \\ w_i^a \sim W(s_i,a_i)}} , [\sum_{a \in A} w_i^a \frac{\pi_e(a|s_i)}{\pi_b^+(a|s_i)} (c_i^a - \hat{R}(s_i, a)) \middle| a_i] \middle| s_i \right] \right] \tag{192}$$

Because $c, w$ are conditionally independent given $s, a$:

$$= \mathbb{E}_{s_i \sim d_0} \left[ \mathbb{V}_{a_i \sim \pi_b(\cdot|s_i)} \left[ \hat{R}(s_i, \pi_e) + \sum_{a \in A} \frac{\pi_e(a|s_i)}{\pi_b^+(a|s_i)} \mathbb{E}_{\substack{c_i^a \sim R(s_i,a), \\ w_i^a \sim W(s_i,a_i)}} , [w_i^a (c_i^a - \hat{R}(s_i, a)) | a_i] \middle| s_i \right] \right] \tag{193}$$

$$= \mathbb{E}_{s_i \sim d_0} \left[ \mathbb{V}_{a_i \sim \pi_b(\cdot|s_i)} \left[ \hat{R}(s_i, \pi_e) + \sum_{a \in A} \frac{\pi_e(a|s_i)}{\pi_b^+(a|s_i)} \bar{W}(a|s_i, a_i)(\bar{R}(s_i, a) - \hat{R}(s_i, a)) \middle| s_i \right] \right] \tag{194}$$

$$= \mathbb{E}_{s_i \sim d_0} \left[ \mathbb{V}_{a_i \sim \pi_b(\cdot|s_i)} \left[ \sum_{a \in A} \frac{\pi_e(a|s_i)}{\pi_b^+(a|s_i)} \bar{W}(a|s_i, a_i)(\bar{R}(s_i, a) - \hat{R}(s_i, a)) \middle| s_i \right] \right] \tag{195}$$

We cannot decompose this term any further because each weight is dependent on $a_i$, and the variance is over the distribution of $a_i$. Now we decompose term 3 using the law of total variance:

$$\text{Term } 3 = \mathbb{E}_{s_i \sim d_0} \left[ \mathbb{E}_{a_i \sim \pi_b(\cdot|s_i)} \left[ \mathbb{V}_{\substack{c_i^a \sim R(s_i,a), \\ w_i^a \sim W(s_i,a_i)}} \left[ \hat{R}(s_i, \pi_e) + \sum_{a \in A} w_i^a \frac{\pi_e(a|s_i)}{\pi_b^+(a|s_i)} (c_i^a - \hat{R}(s_i, a)) \middle| a_i \right] \middle| s_i \right] \right] \tag{196}$$

$$= \mathbb{E}_{s_i \sim d_0} [\mathbb{E}_{a_i \sim \pi_b(\cdot|s_i)} [\mathbb{E}_{c_i^a \sim R(s_i,a)} \underbrace{[\mathbb{V}_{w_i^a \sim W(s_i,a_i)} [\hat{R}(s_i, \pi_e) + \sum_{a \in A} w_i^a \frac{\pi_e(a|s_i)}{\pi_b^+(a|s_i)} (c_i^a - \hat{R}(s_i, a)) | c_i] | a_i]}_{5} \tag{197}$$

$$+ \underbrace{\mathbb{V}_{c_i^a \sim R(s_i,a)} [\mathbb{E}_{w_i^a \sim W(s_i,a_i)} [\hat{R}(s_i, \pi_e) + \sum_{a \in A} w_i^a \frac{\pi_e(a|s_i)}{\pi_b^+(a|s_i)} (c_i^a - \hat{R}(s_i, a)) | c_i] | a_i]}_{6} | s_i]] \tag{198}$$

$$\tag{199}$$

Now we decompose term 6 by distributing the inner-most expectation. Most terms do not have a weight in them:

$$\text{Term 6} = \mathbb{E}_{s_i \sim d_0}\left[\mathbb{E}_{a_i \sim \pi_b}\left[\mathbb{V}_{c_i^a \sim R(s_i,a)}\left[\mathbb{E}_{w_i^a \sim W(s_i,a_i)}\left[\hat{R}(s_i,\pi_e) + \sum_{a \in A} w_i^a \frac{\pi_e(a|s_i)}{\pi_b^+(a|s_i)}(c_i^a - \hat{R}(s_i,a))\Big|c_i\right]\Big|a_i\right]\Big|s_i\right]\right]\tag{200}$$

$$= \mathbb{E}_{s_i \sim d_0}\left[\mathbb{E}_{a_i \sim \pi_b}\left[\mathbb{V}_{c_i^a \sim R(s_i,a_i)}\left[\hat{R}(s_i,\pi_e) + \sum_{a \in A} \mathbb{E}_{w_i^a \sim W(s_i,a_i)}[w_i^a]\frac{\pi_e(a|s_i)}{\pi_b^+(a|s_i)}(c_i^a - \hat{R}(s_i,a))\Big|a_i\right]\Big|s_i\right]\right]\tag{201}$$

$$= \mathbb{E}_{s_i \sim d_0}\left[\mathbb{E}_{a_i \sim \pi_b}\left[\mathbb{V}_{c_i^a \sim R(s_i,a)}\left[\hat{R}(s_i,\pi_e) + \sum_{a \in A} \bar{W}(a|s_i,a_i)\frac{\pi_e(a|s_i)}{\pi_b^+(a|s_i)}(c_i^a - \hat{R}(s_i,a))\Big|a_i\right]\Big|s_i\right]\right]\tag{202}$$

Notice that the inner-most variance term now is over $c_i^a$, so all terms that do not consider $c_i^a$ are considered constants.

$$= \mathbb{E}_{s_i \sim d_0}\left[\mathbb{E}_{a_i \sim \pi_b}\left[\sum_{a \in A} \bar{W}(a|s_i,a_i)^2 \frac{\pi_e(a|s_i)}{\pi_b^+(a|s_i)}^2 (\mathbb{V}_{c_i^a \sim R(s_i,a)}[c_i|a_i])|s_i\right]\right]\tag{203}$$

$$= \mathbb{E}_{s_i \sim d_0}\left[\mathbb{E}_{a_i \sim \pi_b}\left[\sum_{a \in A} \bar{W}(a|s_i,a_i)^2 \frac{\pi_e(a|s_i)}{\pi_b^+(a|s_i)}^2 \sigma_R^2(s_i,a)\Big|s_i\right]\right]\tag{204}$$

Now we decompose term 5. Because the weights are not independent (they need to sum to 1 across all the actions/annotations for a given context), we need to consider the covariance between them. First, we move the variance term outside the summation because $\hat{R}(s_i,\pi_e)$ is considered a constant.

$$\text{Term 5} = \mathbb{E}_{s_i \sim d_0}\left[\mathbb{E}_{a_i \sim \pi_b(\cdot|s_i)}\left[\mathbb{E}_{c_i^a \sim R(s_i,a)}\left[\mathbb{V}_{w_i^a \sim W(s_i,a_i)}\left[\hat{R}(s_i,\pi_e)+\right.\right.\right.\right.\tag{205}$$

$$\left.\left.\left.\left.\sum_{a \in A} w_i^a \frac{\pi_e(a|s_i)}{\pi_b^+(a|s_i)}(c_i^a - \hat{R}(s_i,a))\Big|c_i\right]\Big|a_i\right]\Big|s_i\right]\right]\tag{206}$$

$$= \mathbb{E}_{s_i \sim d_0}\left[\mathbb{E}_{a_i \sim \pi_b(\cdot|s_i)}\left[\mathbb{E}_{c_i^a \sim R(s_i,a)}\left[\mathbb{V}_{w_i^a \sim W(s_i,a_i)}\left[\sum_{a \in A} w_i^a \frac{\pi_e(a|s_i)}{\pi_b^+(a|s_i)}(c_i^a - \hat{R}(s_i,a))\Big|c_i\right]\Big|a_i\right]\Big|s_i\right]\right]\tag{207}$$

When we move the variance inside the summation if we also consider the covariance of each term in the summation. Then, we simplify and consider all terms that don't contain a weight as constant.

$$= \mathbb{E}_{s_i \sim d_0}[\mathbb{E}_{a_i \sim \pi_b(\cdot|s_i)}[\mathbb{E}_{c_i^a \sim R(s_i,a)}[\sum_{a \in A} \mathbb{V}_{w_i^a \sim W(s_i,a_i)}[w_i^a \frac{\pi_e(a|s)}{\pi_b^+(a|s)}(c_i^a - \hat{R}(s_i,a))|c_i]\tag{208}$$

$$+ 2\sum_{\tilde{a}_j,\tilde{a}_k}^{\tilde{a}_j \neq \tilde{a}_k} Cov_{w_i^a \sim W(s_i,a)}(w_i^{\tilde{a}_j}\frac{\pi_e(\tilde{a}_j|s_i)}{\pi_b^+(\tilde{a}_j|s_i)}(c_i^{\tilde{a}_j} - \hat{R}(s_i,\tilde{a}_j)), w_i^{\tilde{a}_k}\frac{\pi_e(\tilde{a}_k|s_i)}{\pi_b^+(\tilde{a}_k|s_i)}(c_i^{\tilde{a}_k} - \hat{R}(s_i,\tilde{a}_k)))|a_i]|s_i]]\tag{209}$$

$$= \mathbb{E}_{s_i \sim d_0}[\mathbb{E}_{a_i \sim \pi_b(\cdot|s_i)}[\mathbb{E}_{c_i^a \sim R(s_i,a)}[\sum_{a \in A} \frac{\pi_e(a|s_i)}{\pi_b^+(a|s_i)}^2 (c_i^a - \hat{R}(s_i,a))^2 \mathbb{V}_{w_i^a \sim W(s_i,a_i)}[w_i^a]\tag{210}$$

$$+ 2\sum_{\tilde{a}_j,\tilde{a}_k}^{\tilde{a}_j \neq \tilde{a}_k} \frac{\pi_e(\tilde{a}_j|s_i)}{\pi_b^+(\tilde{a}_j|s_i)}(c_i^{\tilde{a}_j} - \hat{R}(s_i,\tilde{a}_j))\frac{\pi_e(\tilde{a}_k|s_i)}{\pi_b^+(\tilde{a}_k|s_i)}(c_i^{\tilde{a}_k} - \hat{R}(s_i,\tilde{a}_k))\tag{211}$$

$$\times Cov_{w_i^a \sim W(s_i,a)}(w_i^{\tilde{a}_j}, w_i^{\tilde{a}_k})|a_i]|s_i]]\tag{212}$$

Then, we distribute the inner most expectation, which is over the annotation/reward.

$$= \mathbb{E}_{s_i \sim d_0}[\mathbb{E}_{a_i \sim \pi_b(\cdot|s_i)}[\sum_{a \in A} \frac{\pi_e(a|s_i)}{\pi_b^+(a|s_i)}^2 \mathbb{V}_{w_i^a \sim W(s_i,a_i)}[w_i^a]\mathbb{E}_{c_i^a \sim R(s_i,a)}[(c_i^a - \hat{R}(s_i,a))^2]\tag{213}$$

$$+ 2 \sum_{\tilde{a}_j, \tilde{a}_k}^{\tilde{a}_j \neq \tilde{a}_k} \frac{\pi_e(\tilde{a}_j|s_i)}{\pi_b^+(\tilde{a}_j|s_i)} \frac{\pi_e(\tilde{a}_k|s_i)}{\pi_b^+(\tilde{a}_k|s_i)} \tag{214}$$

$$\times Cov_{w_i^a \sim W(s_i,a)}(w_i^{\tilde{a}_j}, w_i^{\tilde{a}_k}) \mathbb{E}_{c_i^a \sim R(s_i,a)}[(c_i^{\tilde{a}_j} - \hat{R}(s_i, a_j))(c_i^{\tilde{a}_k} - \hat{R}(s_i, \tilde{a}_k))]|a_i]|s_i]] \tag{215}$$

$$\tag{216}$$

Notice how the second expectation term is the product of two independent terms. Thus, the expectation of the product is the product of the expectations of the individual terms.

$$= \mathbb{E}_{s_i \sim d_0}[\mathbb{E}_{a_i \sim \pi_b(\cdot|s_i)}[\sum_{a \in A} \frac{\pi_e(a|s_i)^2}{\pi_b^+(a|s_i)} \mathbb{V}_{w_i^a \sim W(s_i,a_i)}[w_i^a] \mathbb{E}_{c_i^a \sim R(s_i,a)}[(c_i^a - \hat{R}(s_i, a))^2] \tag{217}$$

$$+ 2 \sum_{\tilde{a}_j, \tilde{a}_k}^{\tilde{a}_j \neq \tilde{a}_k} \frac{\pi_e(\tilde{a}_j|s_i)}{\pi_b^+(\tilde{a}_j|s_i)} \frac{\pi_e(\tilde{a}_k|s_i)}{\pi_b^+(\tilde{a}_k|s_i)} \tag{218}$$

$$\times Cov_{w_i^a \sim W(s_i,a)}(w_i^{\tilde{a}_j}, w_i^{\tilde{a}_k})(\bar{R}(s_i, \tilde{a}_j) - \hat{R}(s_i, \tilde{a}_j))(\bar{R}(s_i, \tilde{a}_k) - \hat{R}(s_i, \tilde{a}_k))|a_i]|s_i]] \tag{219}$$

Now we use the definition of variance to simplify the first expectation squared term.

$$= \mathbb{E}_{s_i \sim d_0}[\mathbb{E}_{a_i \sim \pi_b(\cdot|s_i)}[\sum_{a \in A} \frac{\pi_e(a|s_i)^2}{\pi_b^+(a|s_i)} \mathbb{V}_{w_i^a \sim W(s_i,a_i)}[w_i^a] \tag{220}$$

$$\times \left( \mathbb{V}_{c_i^a \sim R(s_i,a)}[(c_i^a - \hat{R}(s_i, a))] + \mathbb{E}_{c_i^a \sim R(s_i,a)}[c_i - \hat{R}(s_i, a)]^2 \right) \tag{221}$$

$$+ 2 \sum_{\tilde{a}_j, \tilde{a}_k}^{\tilde{a}_j \neq \tilde{a}_k} \frac{\pi_e(\tilde{a}_j|s_i)}{\pi_b^+(\tilde{a}_j|s_i)} \frac{\pi_e(\tilde{a}_k|s_i)}{\pi_b^+(\tilde{a}_k|s_i)} \tag{222}$$

$$\times Cov_{w_i^a \sim W(s_i,a)}(w_i^{\tilde{a}_j}, w_i^{\tilde{a}_k})(\bar{R}(s_i, \tilde{a}_j) - \hat{R}(s_i, \tilde{a}_j))(\bar{R}(s_i, \tilde{a}_k) - \hat{R}(s_i, \tilde{a}_k))|a_i]|s_i]] \tag{223}$$

$$= \mathbb{E}_{s_i \sim d_0}[\mathbb{E}_{a_i \sim \pi_b(\cdot|s_i)}[\sum_{a \in A} \frac{\pi_e(a|s_i)^2}{\pi_b^+(a|s_i)} \mathbb{V}_{w_i^a \sim W(s_i,a_i)}[w_i^a] \tag{224}$$

$$\times \left( \sigma_R^2(s_i, a) + (\bar{R}(s_i, a) - \hat{R}(s_i, a)^2) \right) \tag{225}$$

$$+ 2 \sum_{\tilde{a}_j, \tilde{a}_k}^{\tilde{a}_j \neq \tilde{a}_k} \frac{\pi_e(\tilde{a}_j|s_i)}{\pi_b^+(\tilde{a}_j|s_i)} \frac{\pi_e(\tilde{a}_k|s_i)}{\pi_b^+(\tilde{a}_k|s_i)} \tag{226}$$

$$\times Cov_{w_i^a \sim W(s_i,a)}(w_i^{\tilde{a}_j}, w_i^{\tilde{a}_k})(\bar{R}(s_i, \tilde{a}_j) - \hat{R}(s_i, \tilde{a}_j))(\bar{R}(s_i, \tilde{a}_k) - \hat{R}(s_i, \tilde{a}_k))|a_i]|s_i]] \tag{227}$$

$$\tag{228}$$

Putting together the variance term:

$$\mathbb{V}_{D \sim \mathcal{D}}[\hat{V}^{\text{DM-IS}^+}] = \frac{1}{N^2} \sum_{i=1}^{N} \mathbb{V}_{s_i \sim d_0}[\hat{V}^{\pi_e}(s_i)] \tag{229}$$

$$+ \mathbb{E}_{s_i \sim d_0} \left[ \mathbb{V}_{a_i \sim \pi_b(\cdot|s_i)} \left[ \sum_{a \in A} \frac{\pi_e(a|s)}{\pi_b^+(a|s)} \bar{W}(a|s_i, a_i)(\bar{R}(s_i, a_i) - \hat{R}(s_i, a)) \middle| s_i \right] \right] \tag{230}$$

$$+ \mathbb{E}_{s_i \sim d_0} \left[ \mathbb{E}_{a_i \sim \pi_b} \left[ \sum_{a \in A} \bar{W}(a|s_i, a_i)^2 \frac{\pi_e(a|s)^2}{\pi_b^+(a|s)} \sigma_R^2(s_i, a) \middle| s_i \right] \right] \tag{231}$$

$$+ \mathbb{E}_{s_i \sim d_0}[\mathbb{E}_{a_i \sim \pi_b(\cdot|s_i)}[\sum_{a \in A} \frac{\pi_e(a|s_i)^2}{\pi_b^+(a|s_i)} \mathbb{V}_{w_i^a \sim W(s_i,a_i)}[w_i^a] \tag{232}$$

$$\times \left( \sigma_R^2(s_i, a) + (\bar{R}(s_i, a) - \hat{R}(s_i, a)^2) \right) \tag{233}$$

$$+ 2 \sum_{\tilde{a}_j, \tilde{a}_k}^{\tilde{a}_j \neq \tilde{a}_k} \frac{\pi_e(\tilde{a}_j|s_i)}{\pi_b^+(\tilde{a}_j|s_i)} \frac{\pi_e(\tilde{a}_k|s_i)}{\pi_b^+(\tilde{a}_k|s_i)} \tag{234}$$

$$\times Cov_{w_i^a \sim W(s_i,a)}(w_i^{\tilde{a}_j}, w_i^{\tilde{a}_k})(\bar{R}(s_i,\tilde{a}_j) - \hat{R}(s_i,\tilde{a}_j))(\bar{R}(s_i,\tilde{a}_k) - \hat{R}(s_i,\tilde{a}_k))|a_i]|s_i]] \tag{235}$$

$$\tag{236}$$

Now, we study the variance of the estimator under imperfect annotations.

**Proposition 16** (Variance of DM-IS$^+$ under imperfect annotations)**.** *In the case that Assumptions 3 to 5 hold,*

$$\mathbb{V}_{D \sim \mathcal{D}}[\hat{V}^{\text{DM-IS}^+}] = \frac{1}{N^2} \sum_{i=1}^{N} \mathbb{V}_{s_i \sim d_0} \left[ v(\pi_e) + \mathbb{E}_{\substack{s_i \sim d_0 \\ a \sim \pi_e(\cdot|s_i)}} \left[ (1 - \frac{\bar{W}(a|s_i,a)\pi_b(a|s_i)}{\pi_b^+(a|s_i)})\epsilon_G(s_i,a) \right] \right] \tag{237}$$

$$+ \mathbb{E}_{s_i \sim d_0} \left[ \mathbb{V}_{a_i \sim \pi_b(\cdot|s_i)} \left[ \frac{\pi_e(s_i|a_i)}{\pi_b^+(s_i|a_i)} \bar{W}(a_i|s_i,a_i)(\bar{R}(s_i,a_i) - \hat{R}(s_i,a_i)) \right. \right. \tag{238}$$

$$\left. \left. + \sum_{a \in A \setminus \{a_i\}} \frac{\pi_e(a|s_i)}{\pi_b^+(a|s_i)} \bar{W}(a|s_i,a_i)(\bar{R}(s_i,a) + \epsilon_R(s_i,a) - \hat{R}(s_i,a)) \Big| a_i \right] \Big| s_i \right] \tag{239}$$

$$+ \mathbb{E}_{s_i \sim d_0} \left[ \mathbb{E}_{a_i \sim \pi_b} \left[ \sum_{a \in A} \sum_{a \in A} \bar{W}(a|s_i,a_i)^2 \frac{\pi_e(a|s_i)^2}{\pi_b^+(a|s_i)} \sigma_R^2(s_i,a) \right. \right. \tag{240}$$

$$\left. \left. + \sum_{a \in A \setminus \{a_i\}} \bar{W}(a|s_i,a_i)^2 \frac{\pi_e(a|s_i)^2}{\pi_b^+(a|s_i)} \Delta_G(s_i,a) \Big| s_i \right] \right] \tag{241}$$

$$+ \mathbb{E}_{s_i \sim d_0} \left[ \mathbb{E}_{a_i \sim \pi_b(\cdot|s_i)} \left[ \frac{\pi_e(a_i|s_i)^2}{\pi_b^+(a_i|s_i)} \mathbb{V}_{w_i^a \sim W(s_i,a)}[w_i^{a_i}] \right. \right. \tag{242}$$

$$\times \left( \sigma_R^2(s_i,a) + (\bar{R}(s_i,a) - \hat{R}(s_i,a))^2 \right) \tag{243}$$

$$+ \sum_{a \in A \setminus \{a_i\}} \frac{\pi_e(a|s_i)^2}{\pi_b^+(a|s_i)} \mathbb{V}_{w_i^a \sim W(s_i,a)}[w_i^a] \tag{244}$$

$$\times \left( \sigma_R^2(s_i,a) + \Delta_G(s_i,a) + (\bar{R}(s_i,a) + \epsilon_R(s_i,a) - \hat{R}(s_i,a))^2 \right) \tag{245}$$

$$+ 2\bar{\psi}_R(s_i,a_i) \sum_{\tilde{a}_j \neq \tilde{a}_i} \bar{\psi}_G(s_i,\tilde{a}_j) \times Cov_{w_i^a \sim W(s_i,a)} \left( w_i^{\tilde{a}_j}, w_i^{a_i} \right) \tag{246}$$

$$+ \sum_{\tilde{a}_j \neq a_i} \sum_{\tilde{a}_k \neq a_i, \tilde{a}_k \neq \tilde{a}_j} \bar{\psi}_G(s_i,\tilde{a}_j)\bar{\psi}_G(s_i,\tilde{a}_k) \tag{247}$$

$$\times Cov_{w_i^a \sim W(s_i,a)} \left( w_i^{\tilde{a}_j}, w_i^{\tilde{a}_k} \right) \Big| a_i \right] \Big| s_i \right] \tag{248}$$

Proof: We start by moving the variance inside the summation because the variance of each term that the summation is over is independent. This is a similar strategy proposed in the Let $c_i^a \sim C(s_i,a_i)$ be $r_i^a \sim R(s_i,a)$ if $c_i^a$ is a factual reward and $g_i^a \sim G(s_i,a)$ if $c_i^a$ is a counterfactual annotation.

$$\mathbb{V}_{D \sim \mathcal{D}}[\hat{V}^{\text{DM-IS}^+}] \tag{249}$$

$$= \mathbb{V}_{\substack{s_i \sim d_0, a_i \sim \pi_b(\cdot|s_i), \\ c_i^a \sim C(s_i,a), w_i^a \sim W(s_i,a)}} \left[ \frac{1}{N} \sum_{i=1}^{N} \left( \hat{R}(s_i, \pi_e) + \sum_{a \in A} w_i^a \frac{\pi_e(a|s_i)}{\pi_{b^+}(a|s_i)}(c_i^a - \hat{R}(s_i,a)) \right) \right] \tag{250}$$

$$= \frac{1}{N^2} \sum_{i=1}^{N} \mathbb{V}_{\substack{s_i \sim d_0 a_i \sim \pi_b(\cdot|s_i), \\ c_i^a \sim C(s_i,a), w_i^a \sim W(s_i,a)}} \left[ \hat{R}(s_i, \pi_e) + \sum_{a \in A} w_i^a \frac{\pi_e(a|s_i)}{\pi_{b^+}(a|s_i)} (c_i^a - \hat{R}(s_i, a)) \right] \tag{251}$$

Now, we use the law of total variance to separate the joint distribution that the variance is over:

$$= \frac{1}{N^2} \sum_{i=1}^{N} \mathbb{E}_{s_i \sim d_0} \left[ \underbrace{\mathbb{V}_{\substack{a_i \sim \pi_b(\cdot|s_i), \\ c_i^a \sim C(s_i,a), \\ w_i^a \sim W(s_i,a)}} \left[ \hat{R}(s_i, \pi_e) + \sum_{a \in A} w_i^a \frac{\pi_e(a|s_i)}{\pi_b^+(a|s_i)} (c_i^a - \hat{R}(s_i, a)) \middle| s_i \right]}_{2} \right] \tag{252}$$

$$+ \mathbb{V}_{s_i \sim d_0} \left[ \underbrace{\mathbb{E}_{\substack{a_i \sim \pi_b(\cdot|s_i), \\ c_i^a \sim C(s_i,a), \\ w_i^a \sim W(s_i,a)}} \left[ \hat{R}(s_i, \pi_e) + \sum_{a \in A} w_i^a \frac{\pi_e(a|s_i)}{\pi_b^+(a|s_i)} (c_i^a - \hat{R}(s_i, a)) \middle| s_i \right]}_{1} \right] \tag{253}$$

Note that Term 1 is just the value of the target policy conditioned on a specific $s_i$. We have already proved the bias of the estimator under imperfect annotations. This makes the whole variance now:

$$= \frac{1}{N^2} \sum_{i=1}^{N} \mathbb{E}_{s_i \sim d_0} \left[ \underbrace{\mathbb{V}_{\substack{a_i \sim \pi_b(\cdot|s_i), \\ c_i^a \sim C(s_i,a), \\ w_i^a \sim W(s_i,a)}} \left[ \hat{R}(s_i, \pi_e) + \sum_{a \in A} w_i^a \frac{\pi_e(a|s_i)}{\pi_b^+(a|s_i)} (c_i^a - \hat{R}(s_i, a)) \middle| s_i \right]}_{2} \right] \tag{254}$$

$$+ \mathbb{V}_{s_i \sim d_0} \left[ v(\pi_e) + \mathbb{E}_{\substack{s_i \sim d_0 \\ a \sim \pi_e(\cdot|s_i)}} \left[ \left( 1 - \frac{\bar{W}(a|s_i, a)\pi_b(a|s_i)}{\pi_b^+(a|s_i)} \right) \epsilon_G(s_i, a) \right] \right] \tag{255}$$

Now we decompose term 2 using the law of total variance:

$$\text{Term 2} = \mathbb{E}_{s_i \sim d_0} \left[ \mathbb{V}_{\substack{a_i \sim \pi_b(\cdot|s_i), \\ c_i^a \sim C(s_i,a), \\ w_i^a \sim W(s_i,a)}} \left[ \hat{R}(s_i, \pi_e) + \sum_{a \in A} w_i^a \frac{\pi_e(a|s_i)}{\pi_b^+(a|s_i)} (c_i^a - \hat{R}(s_i, a)) \middle| s_i \right] \right] \tag{256}$$

$$= \mathbb{E}_{s_i \sim d_0} \left[ \mathbb{E}_{a_i \sim \pi_b(\cdot|s_i)} \left[ \underbrace{\mathbb{V}_{\substack{c_i^a \sim C(s_i,a), \\ w_i^a \sim W(s_i,a)}} \left[ \hat{R}(s_i, \pi_e) + \sum_{a \in A} w_i^a \frac{\pi_e(a|s_i)}{\pi_b^+(a|s_i)} (c_i^a - \hat{R}(s_i, a)) \middle| a_i \right]}_{3} \middle| s_i \right] \right] \tag{257}$$

$$+ \mathbb{V}_{a_i \sim \pi_b(\cdot|s_i)} \left[ \underbrace{\mathbb{E}_{\substack{c_i^a \sim C(s_i,a), \\ w_i^a \sim W(s_i,a)}} \left[ \hat{R}(s_i, \pi_e) + \sum_{a \in A} w_i^a \frac{\pi_e(a|s_i)}{\pi_b^+(a|s_i)} (c_i^a - \hat{R}(s_i, a)) \middle| a_i \right] \middle| s_i}_{4} \right] \tag{258}$$

Now we decompose term 4 by distributing the inner-most expectation. Note that the inner-most expectation only depends on the distribution over $c_i$ and $w_i$, which is not in many of the terms:

$$\text{Term 4} = \mathbb{E}_{s_i \sim d_0} \left[ \mathbb{V}_{a_i \sim \pi_b(\cdot|s_i)} \left[ \mathbb{E}_{\substack{c_i^a \sim C(s_i,a), \\ w_i^a \sim W(s_i,a)}} \left[ \hat{R}(s_i, \pi_e) + \sum_{a \in A} w_i^a \frac{\pi_e(a|s_i)}{\pi_b^+(a|s_i)} (c_i^a - \hat{R}(s_i, a)) \middle| a_i \right] \middle| s_i \right] \right] \tag{259}$$

$$= \mathbb{E}_{s_i \sim d_0} \left[ \mathbb{V}_{a_i \sim \pi_b(\cdot|s_i)} \left[ \hat{R}(s_i, \pi_e) + \mathbb{E}_{\substack{c_i^a \sim C(s_i,a), \\ w_i^a \sim W(s_i,a)}} \left[ \sum_{a \in A} w_i^a \frac{\pi_e(a|s_i)}{\pi_b^+(a|s_i)} (c_i^a - \hat{R}(s_i, a)) \middle| a_i \right] \middle| s_i \right] \right] \tag{260}$$

Applying linearity of expectation and the fact that $c, w$ are conditionally independent given $s, a$:

$$= \mathbb{E}_{s_i \sim d_0} \left[ \mathbb{V}_{a_i \sim \pi_b(\cdot|s_i)} \left[ \hat{R}(s_i, \pi_e) + \sum_{a \in A} \frac{\pi_e(a|s_i)}{\pi_b^+(a|s_i)} \mathbb{E}_{\substack{c_i^a \sim C(s_i,a), \\ w_i^a \sim W(s_i,a)}} \left[ w_i^a (c_i^a - \hat{R}(s_i, a)) \middle| a_i \right] \middle| s_i \right] \right] \tag{261}$$

$$= \mathbb{E}_{s_i \sim d_0} \left[ \mathbb{V}_{a_i \sim \pi_b(\cdot|s_i)} \left[ \sum_{a \in A} \frac{\pi_e(a|s_i)}{\pi_b^+(a|s_i)} \mathbb{E}_{\substack{c_i^a \sim C(s_i,a), \\ w_i^a \sim W(s_i,a)}} \left[ w_i^a (c_i^a - \hat{R}(s_i,a)) \Big| a_i \right] \Big| s_i \right] \right] \tag{262}$$

$$\tag{263}$$

Now, separating into factual and counterfactual:

$$= \mathbb{E}_{s_i \sim d_0} \left[ \mathbb{V}_{a_i \sim \pi_b(\cdot|s_i)} \left[ \frac{\pi_e(s_i|a_i)}{\pi_b^+(s_i|a_i)} \bar{W}(a_i|s_i,a_i)(\bar{R}(s_i,a_i) - \hat{R}(s_i,a_i)) \right. \right. \tag{264}$$

$$\left. \left. + \sum_{a \in A \backslash \{a_i\}} \frac{\pi_e(a|s_i)}{\pi_b^+(a|s_i)} \bar{W}(a|s_i,a_i)(\bar{R}(s_i,a) + \epsilon_R(s_i,a) - \hat{R}(s_i,a)) \Big| a_i, s_i \right] \right] \tag{265}$$

$$\tag{266}$$

We cannot decompose this term any further because each weight is dependent on $a_i$, and the variance is over the distribution of $a_i$. Now we decompose term 3 using the law of total variance:

$$\text{Term } 3 = \mathbb{E}_{s_i \sim d_0} \left[ \mathbb{E}_{a_i \sim \pi_b(\cdot|s_i)} \left[ \mathbb{V}_{\substack{c_i^a \sim C(s_i,a), \\ w_i^a \sim W(s_i,a)}} \left[ \hat{R}(s_i,\pi_e) + \sum_{a \in A} w_i^a \frac{\pi_e(a|s_i)}{\pi_b^+(a|s_i)} (c_i^a - \hat{R}(s_i,a)) \Big| a_i \right] \Big| s_i \right] \right] \tag{267}$$

$$= \mathbb{E}_{s_i \sim d_0} \left[ \mathbb{E}_{a_i \sim \pi_b(\cdot|s_i)} \left[ \underbrace{\mathbb{E}_{c_i^a \sim C(s_i,a)} \left[ \mathbb{V}_{w_i^a \sim W(s_i,a)} \left[ \hat{R}(s_i,\pi_e) + \sum_{a \in A} w_i^a \frac{\pi_e(a|s_i)}{\pi_b^+(a|s_i)} (c_i^a - \hat{R}(s_i,a)) \Big| c_i \right] \Big| a_i \right]}_{5} \right. \right. \tag{268}$$

$$\left. \left. + \underbrace{\mathbb{V}_{c_i^a \sim C(s_i,a)} \left[ \mathbb{E}_{w_i^a \sim W(s_i,a)} \left[ \hat{R}(s_i,\pi_e) + \sum_{a \in A} w_i^a \frac{\pi_e(a|s_i)}{\pi_b^+(a|s_i)} (c_i^a - \hat{R}(s_i,a)) \Big| c_i \right] \Big| a_i \right]}_{6} \Big| s_i \right] \right] \tag{269}$$

Now we decompose term 6 by distributing the inner-most expectation. Most terms do not have a weight in them:

$$\text{Term } 6 = \tag{270}$$

$$\mathbb{E}_{s_i \sim d_0} \left[ \mathbb{E}_{a_i \sim \pi_b(\cdot|s_i)} \left[ \mathbb{V}_{c_i^a \sim C(s_i,a)} \left[ \mathbb{E}_{w_i^a \sim W(s_i,a)} \left[ \hat{R}(s_i,\pi_e) + \sum_{a \in A} w_i^a \frac{\pi_e(a|s_i)}{\pi_b^+(a|s_i)} (c_i^a - \hat{R}(s_i,a)) \Big| c_i \right] \Big| a_i \right] \Big| s_i \right] \right] \tag{271}$$

$$= \mathbb{E}_{s_i \sim d_0} \left[ \mathbb{E}_{a_i \sim \pi_b(\cdot|s_i)} \left[ \mathbb{V}_{c_i^a \sim R(s_i,a_i)} \left[ \hat{R}(s_i,\pi_e) + \sum_{a \in A} \mathbb{E}_{w_i^a \sim W(s_i,a)} [w_i^a] \frac{\pi_e(a|s_i)}{\pi_b^+(a|s_i)} (c_i^a - \hat{R}(s_i,a)) \Big| a_i \right] \Big| s_i \right] \right] \tag{272}$$

$$= \mathbb{E}_{s_i \sim d_0} \left[ \mathbb{E}_{a_i \sim \pi_b(\cdot|s_i)} \left[ \mathbb{V}_{c_i^a \sim C(s_i,a)} \left[ \hat{R}(s_i,\pi_e) + \sum_{a \in A} \bar{W}(a|s_i,a_i) \frac{\pi_e(a|s_i)}{\pi_b^+(a|s_i)} (c_i^a - \hat{R}(s_i,a)) \Big| a_i \right] \Big| s_i \right] \right] \tag{273}$$

Notice that the inner-most variance term now is over $c_i^a$, so all terms that do not consider $c_i^a$ are considered constants.

$$= \mathbb{E}_{s_i \sim d_0} \left[ \mathbb{E}_{a_i \sim \pi_b(\cdot|s_i)} \left[ \sum_{a \in A} \bar{W}(a|s_i,a_i)^2 \frac{\pi_e(a|s_i)^2}{\pi_b^+(a|s_i)} (\mathbb{V}_{c_i^a \sim C(s_i,a)} [c_i|a_i]) | s_i \right] \right] \tag{274}$$

Now, because the annotations are assumed to be imperfect, we must split the variance term into a factual sample and several possible counterfactual annotations.

$$= \mathbb{E}_{s_i \sim d_0} \left[ \mathbb{E}_{a_i \sim \pi_b(\cdot|s_i)} \left[ \bar{W}(a_i|s_i,a_i)^2 \frac{\pi_e(a_i|s_i)^2}{\pi_b^+(a_i|s_i)} \sigma_R^2(s_i,a_i) \right. \right. \tag{275}$$

$$+ \sum_{a \in A \setminus \{a_i\}} \bar{W}(a|s_i, a_i)^2 \frac{\pi_e(a|s_i)^2}{\pi_b^+(a|s_i)} \left( \sigma_R^2(s_i, a) + \Delta_G(s_i, a) \right) \Big| s_i \Big]\Big] \tag{276}$$

$$= \mathbb{E}_{s_i \sim d_0} \left[ \mathbb{E}_{a_i \sim \pi_b(\cdot|s_i)} \left[ \sum_{a \in A} \bar{W}(a|s_i, a_i)^2 \frac{\pi_e(a|s_i)^2}{\pi_b^+(a|s_i)} \sigma_R^2(s_i, a) \right. \right. \tag{277}$$

$$+ \sum_{a \in A \setminus \{a_i\}} \bar{W}(a|s_i, a_i)^2 \frac{\pi_e(a|s_i)^2}{\pi_b^+(a|s_i)} \Delta_G(s_i, a) \Big| s_i \Big]\Big] \tag{278}$$

$$\tag{279}$$

Now we decompose term 5. Because the weights are not independent (they need to sum to 1 across all the actions/annotations for a given context), we need to consider the covariance between them. First, we move the variance term outside the summation because $\hat{R}(s_i, \pi_e)$ is considered a constant.

$$\text{Term } 5 = \mathbb{E}_{s_i \sim d_0} \left[ \mathbb{E}_{a_i \sim \pi_b(\cdot|s_i)(\cdot|s_i)} \left[ \mathbb{E}_{c_i^a \sim C(s_i, a)} \left[ \mathbb{V}_{w_i^a \sim W(s_i, a)} \left[ \hat{R}(s_i, \pi_e) \right. \right. \right. \right. \tag{280}$$

$$+ \sum_{a \in A} w_i^a \frac{\pi_e(a|s_i)}{\pi_b^+(a|s_i)} (c_i^a - \hat{R}(s_i, a)) \Big| c_i \Big] \Big| a_i \Big] \Big| s_i \Big]\Big] \tag{281}$$

$$= \mathbb{E}_{s_i \sim d_0} \left[ \mathbb{E}_{a_i \sim \pi_b(\cdot|s_i)(\cdot|s_i)} \left[ \mathbb{E}_{c_i^a \sim C(s_i, a)} \left[ \mathbb{V}_{w_i^a \sim W(s_i, a)} \left[ \sum_{a \in A} w_i^a \frac{\pi_e(a|s_i)}{\pi_b^+(a|s_i)} (c_i^a - \hat{R}(s_i, a)) \Big| c_i \Big] \Big| a_i \Big] \Big| s_i \Big]\Big] \tag{282}$$

When we move the variance inside the summation if we also consider the covariance of each term in the summation. Then, we simplify and consider all terms that don't contain a weight as constant.

$$= \mathbb{E}_{s_i \sim d_0} \left[ \mathbb{E}_{a_i \sim \pi_b(\cdot|s_i)} \left[ \mathbb{E}_{c_i^a \sim C(s_i, a)} \left[ \sum_{a \in A} \mathbb{V}_{w_i^a \sim W(s_i, a)} \left[ w_i^a \frac{\pi_e(a|s)}{\pi_b^+(a|s)} (c_i^a - \hat{R}(s_i, a)) \Big| c_i \right] \right. \right. \right. \tag{283}$$

$$+ 2 \sum_{\tilde{a}_j, \tilde{a}_k}^{\tilde{a}_j \neq \tilde{a}_k} Cov_{w_i^a \sim W(s_i, a)} \left( w_i^{\tilde{a}_j} \frac{\pi_e(\tilde{a}_j|s_i)}{\pi_b^+(\tilde{a}_j|s_i)} (c_i^{\tilde{a}_j} - \hat{R}(s_i, \tilde{a}_j)), w_i^{\tilde{a}_k} \frac{\pi_e(\tilde{a}_k|s_i)}{\pi_b^+(\tilde{a}_k|s_i)} \left( c_i^{\tilde{a}_k} - \hat{R}(s_i, \tilde{a}_k) \right) \right) \Big| a_i \Big] \Big| s_i \Big]\Big] \tag{284}$$

$$= \mathbb{E}_{s_i \sim d_0} \left[ \mathbb{E}_{a_i \sim \pi_b(\cdot|s_i)} \left[ \mathbb{E}_{c_i^a \sim C(s_i, a)} \left[ \sum_{a \in A} \frac{\pi_e(a|s_i)^2}{\pi_b^+(a|s_i)} (c_i^a - \hat{R}(s_i, a))^2 \mathbb{V}_{w_i^a \sim W(s_i, a)} \left[ w_i^a \right] \right. \right. \right. \tag{285}$$

$$+ 2 \sum_{\tilde{a}_j, \tilde{a}_k}^{\tilde{a}_j \neq \tilde{a}_k} \frac{\pi_e(\tilde{a}_j|s_i)}{\pi_b^+(\tilde{a}_j|s_i)} (c_i^{\tilde{a}_j} - \hat{R}(s_i, \tilde{a}_j)) \frac{\pi_e(\tilde{a}_k|s_i)}{\pi_b^+(\tilde{a}_k|s_i)} (c_i^{\tilde{a}_k} - \hat{R}(s_i, \tilde{a}_k)) \tag{286}$$

$$\times Cov_{w_i^a \sim W(s_i, a)} \left( w_i^{\tilde{a}_j}, w_i^{\tilde{a}_k} \right) \Big| a_i \Big] \Big| s_i \Big]\Big] \tag{287}$$

Now, we separate the terms to account for different distributions of the factual rewards and counterfactual annotations. Let $\psi(s_i, a) = \frac{\pi_e(a|s_i)}{\pi_b^+(a|s_i)} (c_i^a - \hat{R}(s_i, a))$. Then, we can re-write the expression as:

$$\mathbb{E}_{s_i \sim d_0} \left[ \mathbb{E}_{a_i \sim \pi_b(\cdot|s_i)} \left[ \mathbb{E}_{c_i^a \sim C(s_i, a)} \left[ \sum_{a \in A} \frac{\pi_e(a|s_i)^2}{\pi_b^+(a|s_i)} (c_i^a - \hat{R}(s_i, a))^2 \mathbb{V}_{w_i^a \sim W(s_i, a)} \left[ w_i^a \right] \right. \right. \right. \tag{288}$$

$$+ 2 \sum_{\tilde{a}_j \neq \tilde{a}_k} \psi(s_i, \tilde{a}_j) \psi(s_i, \tilde{a}_k) \tag{289}$$

$$\times Cov_{w_i^a \sim W(s_i, a)} \left( w_i^{\tilde{a}_j}, w_i^{\tilde{a}_k} \right) \Big| a_i \Big] \Big| s_i \Big]\Big] \tag{290}$$

Now, splitting into factual and counterfactual:

$$\mathbb{E}_{s_i \sim d_0}\left[\mathbb{E}_{a_i \sim \pi_b(\cdot|s_i)}\left[\frac{\pi_e(a_i|s_i)}{\pi_b^+(a_i|s_i)}^2 \mathbb{E}_{r_i^{a_i} \sim R(s_i,a_i)}\left[(r_i^a - \hat{R}(s_i,a))^2\right]\mathbb{V}_{w_i^a \sim W(s_i,a)}\left[w_i^{a_i}\right]\right.\right. \tag{291}$$

$$+ \sum_{a \in A\setminus\{a_i\}} \frac{\pi_e(a|s_i)}{\pi_b^+(a|s_i)}^2 \mathbb{E}_{g_i^a \sim G(s_i,a)}\left[(g_i^a - \hat{R}(s_i,a))^2\right]\mathbb{V}_{w_i^a \sim W(s_i,a)}\left[w_i^a\right] \tag{292}$$

$$+ 2\mathbb{E}_{r_i^{a_i} \sim R(s_i,a_i)}\left[\psi(s_i,a_i)\right] \sum_{\tilde{a}_j \neq \tilde{a}_i} \mathbb{E}_{g_i^a \sim G(s_i,a)}\left[\psi(s_i,\tilde{a}_j)\right] \times Cov_{w_i^a \sim W(s_i,a)}\left(w_i^{\tilde{a}_j}, w_i^{a_i}\right) \tag{293}$$

$$+ \sum_{\tilde{a}_j \neq \tilde{a}_i} \sum_{\tilde{a}_k \neq \tilde{a}_i, \tilde{a}_k \neq \tilde{a}_j} \mathbb{E}_{g_i^a \sim G(s_i,a)}\left[\psi(s_i,\tilde{a}_j)\psi(s_i,\tilde{a}_k)\right] \tag{294}$$

$$\left.\left.\times Cov_{w_i^a \sim W(s_i,a)}\left(w_i^{\tilde{a}_j}, w_i^{\tilde{a}_k}\right)\middle| a_i\right]\middle| s_i\right] \tag{295}$$

Recall that $\mathbb{E}_{r_i^{a_i} \sim R(s_i,a_i)}[\psi(s_i,a)] = \bar{\psi}_R(s_i,a) = \frac{\pi_e(a|s_i)}{\pi_b^+(a|s_i)}(\bar{R}(s_i,a) - \hat{R}(s_i,a))$ and $\mathbb{E}_{g_i^a \sim G(s_i,a)}[\psi(s_i,a)] = \bar{\psi}_G(s_i,a) = \frac{\pi_e(a|s_i)}{\pi_b^+(a|s_i)}(\bar{R}(s_i,a) + \epsilon_R(s_i,a) - \hat{R}(s_i,a))$. Also recall that the product of expectation of two independent terms is the expectation of the product. Thus:

$$\mathbb{E}_{s_i \sim d_0}\left[\mathbb{E}_{a_i \sim \pi_b(\cdot|s_i)}\left[\frac{\pi_e(a_i|s_i)}{\pi_b^+(a_i|s_i)}^2 \mathbb{E}_{r_i^{a_i} \sim R(s_i,a_i)}\left[(r_i^a - \hat{R}(s_i,a))^2\right]\mathbb{V}_{w_i^a \sim W(s_i,a)}\left[w_i^{a_i}\right]\right.\right. \tag{296}$$

$$+ \sum_{a \in A\setminus\{a_i\}} \frac{\pi_e(a|s_i)}{\pi_b^+(a|s_i)}^2 \mathbb{E}_{g_i^a \sim G(s_i,a)}\left[(g_i^a - \hat{R}(s_i,a))^2\right]\mathbb{V}_{w_i^a \sim W(s_i,a)}\left[w_i^a\right] \tag{297}$$

$$+ 2\bar{\psi}_R(s_i,a_i) \sum_{\tilde{a}_j \neq \tilde{a}_i} \bar{\psi}_G(s_i,\tilde{a}_j) \times Cov_{w_i^a \sim W(s_i,a)}\left(w_i^{\tilde{a}_j}, w_i^{a_i}\right) \tag{298}$$

$$+ \sum_{\tilde{a}_j \neq \tilde{a}_i} \sum_{\tilde{a}_k \neq \tilde{a}_i, \tilde{a}_k \neq \tilde{a}_j} \bar{\psi}_G(s_i,\tilde{a}_j)\bar{\psi}_G(s_i,\tilde{a}_k) \tag{299}$$

$$\left.\left.\times Cov_{w_i^a \sim W(s_i,a)}\left(w_i^{\tilde{a}_j}, w_i^{\tilde{a}_k}\right)\middle| a_i\right]\middle| s_i\right] \tag{300}$$

Now we use the definition of variance to simplify the first two expectation squared terms.

$$\mathbb{E}_{s_i \sim d_0}\left[\mathbb{E}_{a_i \sim \pi_b(\cdot|s_i)}\left[\frac{\pi_e(a_i|s_i)}{\pi_b^+(a_i|s_i)}^2 \mathbb{V}_{w_i^a \sim W(s_i,a)}\left[w_i^{a_i}\right]\right.\right. \tag{301}$$

$$\times \left(\mathbb{V}_{r_i^{a_i} \sim R(s_i,a_i)}\left[(r_i^a - \hat{R}(s_i,a))\right] + \mathbb{E}_{r_i^a \sim R(s_i,a)}\left[(r_i^a - \hat{R}(s_i,a))\right]^2\right) \tag{302}$$

$$+ \sum_{a \in A\setminus\{a_i\}} \frac{\pi_e(a|s_i)}{\pi_b^+(a|s_i)}^2 \mathbb{V}_{w_i^a \sim W(s_i,a)}\left[w_i^a\right] \tag{303}$$

$$\times \left(\mathbb{V}_{g_i^a \sim G(s_i,a)}\left[(g_i^a - \hat{R}(s_i,a))\right] + \mathbb{E}_{g_i^a \sim G(s_i,a)}\left[(g_i^a - \hat{R}(s_i,a))\right]^2\right) \tag{304}$$

$$+ 2\bar{\psi}_R(s_i,a_i) \sum_{\tilde{a}_j \neq \tilde{a}_i} \bar{\psi}_G(s_i,\tilde{a}_j) \times Cov_{w_i^a \sim W(s_i,a)}\left(w_i^{\tilde{a}_j}, w_i^{a_i}\right) \tag{305}$$

$$+ \sum_{\tilde{a}_j \neq \tilde{a}_i} \sum_{\tilde{a}_k \neq \tilde{a}_i, \tilde{a}_k \neq \tilde{a}_j} \bar{\psi}_G(s_i,\tilde{a}_j)\bar{\psi}_G(s_i,\tilde{a}_k) \tag{306}$$

$$\left.\left.\times Cov_{w_i^a \sim W(s_i,a)}\left(w_i^{\tilde{a}_j}, w_i^{\tilde{a}_k}\right)\middle| a_i\right]\middle| s_i\right] \tag{307}$$

Simplifying:

$$\mathbb{E}_{s_i \sim d_0}\Bigg[\mathbb{E}_{a_i \sim \pi_b(\cdot|s_i)}\Bigg[\frac{\pi_e(a_i|s_i)}{\pi_b^+(a_i|s_i)}^2 \mathbb{V}_{w_i^a \sim W(s_i,a)}\Big[w_i^{a_i}\Big] \tag{308}$$

$$\times \left(\sigma_R^2(s_i,a) + (\bar{R}(s_i,a) - \hat{R}(s_i,a))^2\right) \tag{309}$$

$$+ \sum_{a \in A \setminus \{a_i\}} \frac{\pi_e(a|s_i)}{\pi_b^+(a|s_i)}^2 \mathbb{V}_{w_i^a \sim W(s_i,a)}\Big[w_i^a\Big] \tag{310}$$

$$\times \left(\sigma_R^2(s_i,a) + \Delta_G(s_i,a) + (\bar{R}(s_i,a) + \epsilon_R(s_i,a) - \hat{R}(s_i,a))^2\right) \tag{311}$$

$$+ 2\bar{\psi}_R(s_i,a_i) \sum_{\tilde{a}_j \neq \tilde{a}_i} \bar{\psi}_G(s_i,\tilde{a}_j) \times Cov_{w_i^a \sim W(s_i,a)}\left(w_i^{\tilde{a}_j}, w_i^{a_i}\right) \tag{312}$$

$$+ \sum_{\tilde{a}_j \neq a_i} \sum_{\tilde{a}_k \neq a_i, \tilde{a}_k \neq \tilde{a}_j} \bar{\psi}_G(s_i,\tilde{a}_j)\bar{\psi}_G(s_i,\tilde{a}_k) \tag{313}$$

$$\times Cov_{w_i^a \sim W(s_i,a)}\left(w_i^{\tilde{a}_j}, w_i^{\tilde{a}_k}\right)\Big|a_i\Bigg]\Bigg|s_i\Bigg] \tag{314}$$

Putting together the variance term:

$$\mathbb{V}_{D \sim \mathcal{D}}[\hat{V}^{\text{DM-IS}^+}] = \tag{315}$$

$$\frac{1}{N^2}\sum_{i=1}^N \mathbb{V}_{s_i \sim d_0}\Bigg[v(\pi_e) + \mathbb{E}_{\substack{s_i \sim d_0 \\ a \sim \pi_e(\cdot|s_i)}}\Big[(1 - \frac{\bar{W}(a|s_i,a)\pi_b(a|s_i)}{\pi_b^+(a|s_i)})\epsilon_G(s_i,a)\Big]\Bigg] \tag{316}$$

$$+ \mathbb{E}_{s_i \sim d_0}\Bigg[\mathbb{V}_{a_i \sim \pi_b(\cdot|s_i)}\Big[\frac{\pi_e(s_i|a_i)}{\pi_b^+(s_i|a_i)}\bar{W}(a_i|s_i,a_i)(\bar{R}(s_i,a_i) - \hat{R}(s_i,a_i)) \tag{317}$$

$$+ \sum_{a \in A \setminus \{a_i\}} \frac{\pi_e(a|s_i)}{\pi_b^+(a|s_i)}\bar{W}(a|s_i,a_i)(\bar{R}(s_i,a) + \epsilon_R(s_i,a) - \hat{R}(s_i,a))\Big|a_i\Big]\Big|s_i\Bigg] \tag{318}$$

$$+ \mathbb{E}_{s_i \sim d_0}\Bigg[\mathbb{E}_{a_i \sim \pi_b}\Big[\sum_{a \in A}\sum_{a \in A}\bar{W}(a|s_i,a_i)^2 \frac{\pi_e(a|s_i)}{\pi_b^+(a|s_i)}^2 \sigma_R^2(s_i,a) \tag{319}$$

$$+ \sum_{a \in A \setminus \{a_i\}} \bar{W}(a|s_i,a_i)^2 \frac{\pi_e(a|s_i)}{\pi_b^+(a|s_i)}^2 \Delta_G(s_i,a)\Big|s_i\Big]\Bigg] \tag{320}$$

$$+ \mathbb{E}_{s_i \sim d_0}\Bigg[\mathbb{E}_{a_i \sim \pi_b(\cdot|s_i)}\Big[\frac{\pi_e(a_i|s_i)}{\pi_b^+(a_i|s_i)}^2 \mathbb{V}_{w_i^a \sim W(s_i,a)}[w_i^{a_i}] \tag{321}$$

$$\times \left(\sigma_R^2(s_i,a) + (\bar{R}(s_i,a) - \hat{R}(s_i,a))^2\right) \tag{322}$$

$$+ \sum_{a \in A \setminus \{a_i\}} \frac{\pi_e(a|s_i)}{\pi_b^+(a|s_i)}^2 \mathbb{V}_{w_i^a \sim W(s_i,a)}[w_i^a] \tag{323}$$

$$\times \left(\sigma_R^2(s_i,a) + \Delta_G(s_i,a) + (\bar{R}(s_i,a) + \epsilon_R(s_i,a) - \hat{R}(s_i,a))^2\right) \tag{324}$$

$$+ 2\bar{\psi}_R(s_i,a_i) \sum_{\tilde{a}_j \neq \tilde{a}_i} \bar{\psi}_G(s_i,\tilde{a}_j) \times Cov_{w_i^a \sim W(s_i,a)}\left(w_i^{\tilde{a}_j}, w_i^{a_i}\right) \tag{325}$$

$$+ \sum_{\tilde{a}_j \neq a_i} \sum_{\tilde{a}_k \neq a_i, \tilde{a}_k \neq \tilde{a}_j} \bar{\psi}_G(s_i,\tilde{a}_j)\bar{\psi}_G(s_i,\tilde{a}_k) \tag{326}$$

$$\times Cov_{w_i^a \sim W(s_i, a)}\left(w_i^{\tilde{a}_j}, w_i^{\tilde{a}_k}\right)\Bigg|a_i\Bigg]\Bigg|s_i\Bigg] \tag{327}$$

### G.3 Expectation and variance of DM$^+$-IS$^+$

The DR estimator is defined as

$$\hat{V}^{\text{DM}^+\text{-IS}^+} = \frac{1}{N}\sum_{i=1}^{N}\left(\hat{R}^+(s_i, \pi_e) + \sum_{a \in A} w_i^a \frac{\pi_e(a|s_i)}{\pi_{b^+}(a|s_i)}(c_i^a - \hat{R}^+(s_i, a))\right)$$

#### G.3.1 Expectation

The expectation without making any assumptions about the quality of the counterfactual annotations is summarized in Theorem 2 which is restated below.

**Theorem** (Expectation of DM-IS$^+$ and DM$^+$-IS$^+$ under imperfect annotations). *Under Assumptions 3 and 5, the two estimators have the same expectation:*

$$\mathbb{E}[\hat{V}^{\text{DM-IS}^+}] = \mathbb{E}[\hat{V}^{\text{DM}^+\text{-IS}^+}] = v(\pi_e) + \mathbb{E}_{s_i \sim d_0}[\mathbb{E}_{a \sim \pi_e(s_i)}[(1 - \frac{\bar{W}(a|s_i, a)\pi_b(a|s_i)}{\pi_b^+(a|s_i)})\epsilon_G(s_i, a)]]$$

The proof is nearly identical to the one for the DM-IS$^+$ estimator, except that the reward function $\hat{R}$ is replaced by $\hat{R}^+$.

**Proposition 17** (Unbiasedness of DM$^+$-IS$^+$). *If both Assumptions 2 and 5 hold, the DM$^+$-IS$^+$ estimator is unbiased, $\mathbb{E}[\hat{V}^{\text{DM}^+\text{-IS}^+}] = v(\pi_e)$.*

Proof: The proof is identical to the unbiasedness proof for the DM-IS$^+$ estimator.

#### G.3.2 Variance

We first derive the variance under the perfect annotation setting.

**Proposition 18** (Variance of DM$^+$-IS$^+$ under perfect annotations). *If Assumptions 2 and 5 holds,*

$$\mathbb{V}_{D \sim \mathcal{D}}[\hat{V}^{\text{DM}^+\text{-IS}^+}] = \frac{1}{N^2}\sum_{i=1}^{N}\mathbb{V}_{s_i \sim d_0}[\hat{V}^{\pi_e}(s_i)] \tag{328}$$

$$+ \mathbb{E}_{s_i \sim d_0}\left[\mathbb{V}_{a_i \sim \pi_b(\cdot|s_i)}\left[\sum_{a \in A}\frac{\pi_e(a|s)}{\pi_b^+(a|s)}\bar{W}(a|s_i, a_i)(\bar{R}(s_i, a) - \hat{R}^+(s_i, a))\Bigg|s_i\right]\right] \tag{329}$$

$$+ \mathbb{E}_{s_i \sim d_0}\left[\mathbb{E}_{a_i \sim \pi_b}\left[\sum_{a \in A}\bar{W}(a|s_i, a_i)^2\frac{\pi_e(a|s)}{\pi_b^+(a|s)}^2\sigma_R^2(s_i, a)\Bigg|s_i\right]\right] \tag{330}$$

$$+ \mathbb{E}_{s_i \sim d_0}[\mathbb{E}_{a_i \sim \pi_b(\cdot|s_i)}[\sum_{a \in A}\frac{\pi_e(a|s_i)}{\pi_b^+(a|s_i)}^2\mathbb{V}_{w_i^a \sim W(s_i, a_i)}[w_i^a] \tag{331}$$

$$\times \left(\sigma_R^2(s_i, a) + (\bar{R}(s_i, a) - \hat{R}^+(s_i, a)^2)\right) \tag{332}$$

$$+ 2\sum_{\tilde{a}_j, \tilde{a}_k}^{\tilde{a}_j \neq \tilde{a}_k}\frac{\pi_e(\tilde{a}_j|s_i)}{\pi_b^+(\tilde{a}_j|s_i)}\frac{\pi_e(\tilde{a}_k|s_i)}{\pi_b^+(\tilde{a}_k|s_i)} \tag{333}$$

$$\times Cov_{w_i^a \sim W(s_i, a)}(w_i^{\tilde{a}_j}, w_i^{\tilde{a}_k})(\bar{R}(s_i, \tilde{a}_j) - \hat{R}^+(s_i, \tilde{a}_j))(\bar{R}(s_i, \tilde{a}_k) - \hat{R}^+(s_i, \tilde{a}_k))|a_i]|s_i]] \tag{334}$$

The proof is identical to the one used to derive variance under perfect conditions for the DM-IS$^+$ estimator, except with $\hat{R}^+$ replacing $\hat{R}$.

Now, we discuss the variance under imperfect annotations.

**Proposition 19** (Variance of DM$^+$-IS$^+$ under imperfect annotations).

$$\mathbb{V}_{D\sim\mathcal{D}}\left[\hat{V}^{\text{DM}^+\text{-IS}^+}\right] = \tag{335}$$

$$\frac{1}{N^2}\sum_{i=1}^{N}\mathbb{V}_{s_i\sim d_0}\left[v(\pi_e) + \mathbb{E}_{\substack{s_i\sim d_0 \\ a\sim\pi_e(\cdot|s_i)}}\left[(1 - \frac{\bar{W}(a|s_i,a)\pi_b(a|s_i)}{\pi_b^+(a|s_i)})\epsilon_G(s_i,a)\right]\right] \tag{336}$$

$$+ \mathbb{E}_{s_i\sim d_0}\left[\mathbb{V}_{a_i\sim\pi_b(\cdot|s_i)}\left[\frac{\pi_e(s_i|a_i)}{\pi_b^+(s_i|a_i)}\bar{W}(a_i|s_i,a_i)(\bar{R}(s_i,a_i) - \hat{R}^+(s_i,a_i))\right.\right. \tag{337}$$

$$+ \sum_{a\in A\backslash\{a_i\}}\frac{\pi_e(a|s_i)}{\pi_b^+(a|s_i)}\bar{W}(a|s_i,a_i)(\bar{R}(s_i,a) + \epsilon_R(s_i,a) - \hat{R}^+(s_i,a))\Big|a_i\Big]\Big|s_i\Big] \tag{338}$$

$$+ \mathbb{E}_{s_i\sim d_0}\left[\mathbb{E}_{a_i\sim\pi_b}\left[\sum_{a\in A}\sum_{a\in A}\bar{W}(a|s_i,a_i)^2\frac{\pi_e(a|s_i)}{\pi_b^+(a|s_i)}^2\sigma_R^2(s_i,a)\right.\right. \tag{339}$$

$$+ \sum_{a\in A\backslash\{a_i\}}\bar{W}(a|s_i,a_i)^2\frac{\pi_e(a|s_i)}{\pi_b^+(a|s_i)}^2\Delta_G(s_i,a)\Big|s_i\Big]\Big] \tag{340}$$

$$+ \mathbb{E}_{s_i\sim d_0}\left[\mathbb{E}_{a_i\sim\pi_b(\cdot|s_i)}\left[\frac{\pi_e(a_i|s_i)}{\pi_b^+(a_i|s_i)}^2\mathbb{V}_{w_i^a\sim W(s_i,a)}[w_i^{a_i}]\right.\right. \tag{341}$$

$$\times\left(\sigma_R^2(s_i,a) + (\bar{R}(s_i,a) - \hat{R}^+(s_i,a))^2\right) \tag{342}$$

$$+ \sum_{a\in A\backslash\{a_i\}}\frac{\pi_e(a|s_i)}{\pi_b^+(a|s_i)}^2\mathbb{V}_{w_i^a\sim W(s_i,a)}[w_i^a] \tag{343}$$

$$\times\left(\sigma_R^2(s_i,a) + \Delta_G(s_i,a) + (\bar{R}(s_i,a) + \epsilon_R(s_i,a) - \hat{R}^+(s_i,a))^2\right) \tag{344}$$

$$+ 2\bar{\psi}_R(s_i,a_i)\sum_{\tilde{a}_j\neq\tilde{a}_i}\bar{\psi}_G(s_i,\tilde{a}_j)\times Cov_{w_i^a\sim W(s_i,a)}\left(w_i^{\tilde{a}_j},w_i^{a_i}\right) \tag{345}$$

$$+ \sum_{\tilde{a}_j\neq a_i}\sum_{\tilde{a}_k\neq a_i,\tilde{a}_k\neq\tilde{a}_j}\bar{\psi}_G(s_i,\tilde{a}_j)\bar{\psi}_G(s_i,\tilde{a}_k) \tag{346}$$

$$\times Cov_{w_i^a\sim W(s_i,a)}\left(w_i^{\tilde{a}_j},w_i^{\tilde{a}_k}\right)\Big|a_i\Big]\Big|s_i\Big] \tag{347}$$

The proof is identical to that of Proposition 16, except the reward model $\hat{R}$ is replaced with $\hat{R}^+$.

## H  Equivalence of IS$^+$, DM-IS$^+$, DM$^+$-IS$^+$ under equal weights

**Corollary 20** (Equivalence of IS$^+$, DM$^+$-IS, DM$^+$-IS$^+$ under equal weights). *If we assume that all the weights are equal, and that all OPE methods have access to the same set of counterfactual annotations, $\hat{V}^{\text{IS}^+}$, $\hat{V}^{\text{DM-IS}^+}$, $\hat{V}^{\text{DM}^+\text{-IS}^+}$ are equivalent by definition.*

Proof: First we re-state the definition of the three methods for a single sample $s_i, a_i, r_i$, and all counterfactual annotations $c_i^a$, under the assumption that all weights are equal. Under this assumption, each weight is $\frac{1}{|A|}$:

1.

$$\hat{V}^{\text{IS}^+} = \sum_{a\in\mathcal{A}}w_i^a\frac{\pi_e(a|s_i)}{\pi_{b^+}(a|s_i)}c_i^a \tag{348}$$

$$\hat{V}^{\text{IS}^+*} = \sum_{a \in A} \pi_e(a|s_i)c_i^a \tag{349}$$

$$\hat{V}^{\text{DM-IS}^+} = \hat{R}(s_i, \pi_e) + \sum_{a \in A} \pi_e(a|s_i)(c_i^a - \hat{R}(s_i, a)) \tag{350}$$

$$\hat{V}^{\text{DM}^+\text{-IS}^+*} = \hat{R}^+(s_i, \pi_e) + \sum_{a \in A} \pi_e(a|s_i)(c_i^a - \hat{R}^+(s_i, a)) \tag{351}$$

$\hat{V}^{\text{IS}^+*}$ is $\hat{V}^{\text{IS}^+}$ under equal weights. The following is a derivation of how we can reach $\hat{V}^{\text{IS}^+*}$ under equal weights using the definition of the augmented behavior policy:

$$\hat{V}^{\text{IS}^+} = \sum_{a \in \mathcal{A}} w_i^a \frac{\pi_e(a|s_i)}{\pi_{b^+}(a|s_i)} c_i^a \tag{352}$$

$$= \sum_{a \in \mathcal{A}} \bar{W}(a|s_i, a_i) \frac{\pi_e(a|s_i)}{\pi_{b^+}(a|s_i)} c_i^a \tag{353}$$

$$= \sum_{a \in \mathcal{A}} \pi_e(a|s_i)c_i^a \tag{354}$$

$$= \hat{V}^{C*-IS} \tag{355}$$

2. Now, we write the definition for $\hat{R}(s_i, \pi_e)$ and $\hat{R}^+(s_i, \pi_e)$.

$$\hat{R}(s_i, \pi_e) = \sum_{a \in A} \pi_e(a|s_i)\hat{R}(s_i, a) \tag{356}$$

$$\hat{R}^+(s_i, \pi_e) = \sum_{a \in A} \pi_e(a|s_i)\hat{R}^+(s_i, a) \tag{357}$$

3. Let us start by re-writing the definition of $\hat{V}^{\text{DM-IS}^+}$ using the definition of $\hat{R}(s_i, \pi_e)$:

$$\hat{V}^{\text{DM-IS}^+} = \sum_{a \in A} \pi_e(a|s_i)\hat{R}(s_i, a) + \sum_{a \in A} \pi_e(a|s_i)(c_i^a - \hat{R}(s_i, a)) \tag{358}$$

$$= \sum_{a \in A} \pi_e(a|s_i)\hat{R}(s_i, a) + \sum_{a \in A} \pi_e(a|s_i)c_i^a - \sum_{a \in A} \pi_e(a|s_i)\hat{R}(s_i, a) \tag{359}$$

$$\tag{360}$$

4. Note that the second term is the definition of *AugIS* for a single sample. Also note that subtracting term 3 from term 1 is equal to 0 by definition.

This shows that IS$^+$ is equivalent to DM-IS$^+$. Note that we can replace the estimate of the reward function with $\hat{R}^+$ and produce the same derivation for DM$^+$-IS$^+$. Thus, we have shown that the three estimators IS$^+$, DM-IS$^+$ DM$^+$-IS$^+$ are equivalent when the weights are equal and all three methods have access to the same counterfactual annotations.

**Corollary 21** (Equivalence of Variance of IS$^+$).

Now, we want to verify that our variance decomposition for $\hat{V}^{\text{IS}^+}, \hat{V}^{\text{DM-IS}^+*}, \hat{V}^{\text{DM}^+\text{-IS}^+*}$ are equivalent to that of IS$^+*$. Recall that we are in the setting where the weights are equal. Here, we also assume that $\hat{R}, \hat{R}^+$ are constant and have been learned already.

1. First we note that the variance decomposition for DM-IS$^+$ and DM$^+$-IS$^+$ each have 7 terms. The last 4 terms reason about the covariance of the weights. Because our weights are constant and equal, the last four terms become 0.

2. Next, we note that the first term is identical in all of the variance decompositions. Namely, this is $\mathbb{V}_{s_i \sim d_0}[V^{\pi_e}(s_i)]$.

3. Now, we note that if $\hat{R}, \hat{R}^+$ are constants, the second term in the variance of DM-IS$^+$ and DM$^+$-IS$^+$ becomes 0.

4. All we need to do is establish that that the third term is equivalent to the third term for the variance decomposition. This is true by definition.

Thus, all three variance decompositions are the same under this setting.

