# OpenReview forum: "CANDOR: Counterfactual ANnotated DOubly Robust Off-Policy Evaluation"
_TMLR — Rejected by TMLR_

### Review · Reviewer_YL5w · 2025-09-14

**Summary Of Contributions:**

This paper tackles off-policy evaluation with counterfactual annotations—extra labels for unobserved actions that may be biased and/or noisy—and introduces three doubly-robust estimators that specify where such annotations should (and should not) enter the pipeline: DM+-IS (annotations only in the model-based/DM term), DM-IS+ (annotations only in the IS term), and DM+-IS+ (annotations in both). It proves a bias analysis: with known propensities and support, DM+-IS remains unbiased even when annotations are biased, whereas any estimator that feeds annotations into the IS correction (method of Tang & Wiens (2023)) inherits bias proportional to the annotation bias and reweighting. The paper derives a variance decomposition for DM+-IS that makes clear how annotation noise affects the learned reward model and overall estimator variance. Extensive simulations on synthetic settings (Two context bandit, heartsteps, sepsis) and a semi-synthetic setting (MIMIC-IV case study with LLM-generated annotations) corroborate the idea that DM+-IS provides the most robust estimation of the policy value under reward mispecification and imperfect annotation.

Strengths:
- Well-scoped, realistic problem: Clearly frames OPE with imperfect counterfactual annotations (biased/noisy)—a practical setting of relevance for the ML community. This paper is a natural follow up to Tang & Wiens (2023).
- Useful bias analysis: allows us to understand why DM+-IS seems to be a good estimator. It is an interesting result to see that DM-IS+ is biased because it pushes the biased annotations through the IS correction.
- Extensive empirical evaluation: the experiments are conducted on a wide range of settings.
- Clarity: overall the paper is well written (see however my remark below on section 2.3).

Weaknesses:
- Known propensities assumption. The author frame the paper as a doubly robust method. However, doubly robust refers to being robust to either DM or the IPS nuisance parameter estimations. That means the estimator is unbiased if at least one of them is well-specified. In the case of known propensities, the estimator is trivially non biased (DM+-IPS) - since propensities are known. This makes the bias result quite straightforward... It would have been an extremely helpful contribution to extend the IS+ methodology of Tang & Wiens (2023) to unknown propensities.
- Partial/vacuous variance analysis: the authors fully derive variance only for **DM+-IS**. The IS-side variants (**DM-IS+**, **DM+-IS+**) are left to empirics. The authors argue that those estimators are biased, but what if they have less variance? The bias–variance trade-off is not theoretically pinned down there. In that same directions, the authors provide empirical experiments in which RMSE only is discussed in the main text, but what about the bias/variance decomposition?
- Vacuous section 5 on selecting an OPE estimator: the name of section 5, with all due respect, feels naive. The authors discuss the (practical?) strategy of choice of the estimator when the reward is either specified/mispecified, the annotations are either perfect/imperfect. Apart from bringing the (useful) insight that DM+-IS estimator is close to optimal performances in all 4 cases, no practitioner in real-world is going to know if their reward are well-specified nor their annotations perfect. You could call that section robustness analysis of DM+-IS. Talking about how to select an OPE estimator, you do not discuss offline model selection (how to tune your DM estimator for example) in your setting, nor do you discuss any principled weight/selection strategy for annotations. (how many annotations to gather, performance w.r.t to M, the number of annotations, how to weight them optimally...).
- Empirical results are presented with heatmaps. This is too qualitative, this does not even allow to compare estimators with rigorous quantitative metrics (let alone do paired two sample tests, to rigorously conclude that a method is statistically better than another one...).

**Additional Comments:**

It would have been nice to connect the doubly robust estimators proposed here with all the literature on one-step estimators, semiparametric efficient statistics, efficient influence functions. Possibly discuss the asymptotic RMSE properties of the estimators with regards to the optimal properties of the one-steps estimators. (Your estimator is most likely a one-step estimator, so you are likely to inherit those optimal properties and accelerated convergence rates of the excess risks as well...). Discussing the bias and variance of the doubly robust estimators is only the tip of the iceberg of the great properties of such estimators.

Hines, Oliver et al. “Demystifying Statistical Learning Based on Efficient Influence Functions.” The American Statistician 76 (2021): 292 - 304.
Kennedy, Edward H. "Semiparametric doubly robust targeted double machine learning: a review." Handbook of statistical methods for precision medicine (2024): 207-236.

**Audience:**

Yes

**Audience Explanation:**

Yes, the idea of looking into imperfect annotations is really useful and I thank the authors for bringing the useful case where the method of Tang & Wiens (2023) fails.

**Claims And Evidence:**

No

**Claims Explanation:**

The paper provides a bias and a variance analysis of the introduced estimators. But the variance analysis is not complete, and the bias and variance theoretical insights are not well connected to the empirical sections.
The paper itself states it relies on empirical studies to assess their introduced estimators, but the experiments show qualitative heatmaps that do not allow to compare rigorously the estimators. Only in the semi-synthetic MIMIC-IV setting can we see more quantitative values of the RMSE, but again in those plots as well error bars would be welcome.

**Requested Changes:**

I urge the authors to look into the following:
- providing quantitative metrics for empirical results, to have clearer comparisons of the methods.
- add the variance analysis of the other estimators and provide clearer empirical results to compare bias and variances of the all three variants: provide at least a high-level variance bound for DM-IS+ and DM+-IS+, even on a simplified setting (finite actions, bounded rewards, fixed weights).
- improve the writing of section 2.3, which would make the paper more self-contained and not require the readers to go to read Tang & Wiens (2023).
- improve section 5, either rename it or discuss the points raised above (weaknesses).

Minor:
- Analysis - dependence on sample-splitting. Unbiasedness arguments for DM(+)-IS implicitly rely on training the reward model on disjoint data (or cross-fitting). This operational requirement isn’t front-and-center, but it matters in practice.
- Experiments - annotation noise sensitivity. Even though DM+-IS is bias-robust, noisy/heteroskedastic annotations can inflate variance via the learned reward model; the paper offers limited guidance on diagnosing or mitigating that in practice.

---

> ### Author Response · Authors · 2025-10-29
>
> We thank the reviewer for their suggestions and are thrilled to hear that our work is relevant for the TMLR audience. We respond to individual concerns below.
>
> **Re: Imperfect IPS Ratios**
>
> As we mention in Section 3, our goal is to define an estimator that is “doubly robust” to two sources of error (the error of the annotation and the error of the reward model). As such, we do not account for imperfect IPS ratios and assume that the estimates of IPS ratios are fairly accurate.
>
> In the case that the IPS ratio is inaccurate, all proposed estimators will be biased. In particular, [1] uses the term $\delta(s, a, h) = 1- \pi_b(a|s,h)/\hat{\pi_b}(a|s,h)$ to describe the estimation error of the learned behavior policy $\hat{\pi_b}$. This estimation error will propagate through the bias and variance reductions introduced in our work.
>
> To further illustrate this, we report the performance of $DM^+-IS$ with varying degrees of incorrectly estimated $\pi_b$ (Figure 11 in the updated manuscript). We note in this plot that when the degree of estimation error increases, the estimator exhibits higher RMSE, indicating that it is biased. We leave further evaluation of this setting to future work.
>
> **Re: Characterization of variance**
>
> We include variance derivations for all three estimators under perfect annotations. We only perform the variance derivation under imperfect annotations for one estimator. We analyze the variance for the other two estimators empirically because their derivations are not directly comparable to that of $DM^+-IS$.
>
> In particular, the proof of the variance of $DM-IS^{+}$ and $DM^{+}-IS^{+}$ under imperfect annotations starts similar to the corresponding proof for $DM^+-IS$ (Appendix G.1.2) until Equation 98. However, the second term (variance of the expectation of the estimator) does not simplify to $v(\pi_e)$ and instead is $$V[v^{\pi_e}(s) + E_{s_i \sim d_0}[E_{a \sim \pi_e(s_i)}[(1- \frac{\bar{W}(a|s_i,a) \pi_b(a|s_i)}{\pi_b^+(a|s_i)}) \epsilon_G(s_i, a)]]]$$.
>
> This term simplifies to $V[v^{\pi_e}(s_i)] + V[E_{s_i \sim d_0}[E_{a \sim \pi_e(s_i)}[(1- \frac{\bar{W}(a|s_i,a) \pi_b(a|s_i)}{\pi_b^+(a|s_i)}) \epsilon_G(s_i, a)]]]] + Cov(v(s_i), E_{s_i \sim d_0}[E_{a \sim \pi_e(s_i)}[(1- \frac{\bar{W}(a|s_i,a) \pi_b(a|s_i)}{\pi_b^+(a|s_i)}) \epsilon_G(s_i, a)]])$.
>
> The last covariance term is difficult to analyze; without additional assumptions on the weights or annotation quality, we do not know its sign or magnitude. However, we will include the variance derivations in the manuscript for completeness.
>
> **Section 5**
>
> We are happy to revise the section. We agree that practitioners rarely know the quality of their reward model or annotations, and by showing that $DM^+{-}IS$ performs near optimally in such cases, we demonstrate its practical utility. We will also add details on tuning the DM estimator (e.g., via cross-fitting). In our work, we make no assumptions about how to weight counterfactual annotations, leaving this choice to practitioners. Finally, Figure 4b examines the effect of the number of counterfactual annotations, and we defer more comprehensive analysis to future work.
>
> **Dependence on sample splitting**
>
> We update the writing to note that we do split the data into two: one split is used for learning the OPE estimate and the other is used to fit the reward model.
>
> **Empirical Results presented heatmaps**
>
> Figure 2 involves four variables (mean RMSE, estimator, annotation bias, and variance), making bar plots difficult, though we include one in Appendix A. In Figure 3, we collapse the variance dimension since RMSE varies more with bias, and present these results as a line plot with standard deviation error bars in Appendix A.
>
> **Re: writing of Section 2.3**
>
> We will revise to include sufficient detail so readers need not reference Tang et. al. In particular, we will add a full description of the weight assignment strategy.
>
> **Mitigating noise of annotations in reward model for \hat{R} in $DM^+-IS$**
>
> While an unbiased estimator can still exhibit inflated variance under noisy annotations, Figures 3 and 4 show that $DM^+{-}IS$ is less affected than other annotation-based estimators. Exploring methods to further mitigate this remains an important direction for future work.
>
> **Re: Related Literature**
>
> We will add a discussion of one-step estimators, semiparametric efficiency, and influence functions, noting that the doubly robust estimator is the one-step correction to DM and connects to asymptotic variance analyses in OPE.
>
> **Re: Asymptotic RMSE**
>
> We focus on the finite-sample setting and derive corresponding bias and variance bounds, which yield more informative and realistic comparisons for our applications. Using these results, we can also show weak consistency, which is true likely only for the $DM^+{-}IS$ estimator among the three we propose.
>
> [1] Dudík, Miroslav et al. “Doubly Robust Policy Evaluation and Learning.” International Conference on Machine Learning (2011).

---

> ### Comment · Reviewer_YL5w · 2025-11-04
> **Follow-up review**
>
> I have reviewed the updated manuscript. While the authors have improved minor aspects of presentation, several of the core concerns raised in my original review remain unresolved. The revision may not yet meet the level of precision and methodological clarity required for publication in *TMLR*.
>
> 1. Definition and use of “doubly robust”
>
> The paper continues to define *doubly robust* as robustness to **reward-model** and **annotation** errors under the assumption of perfectly known propensities. This departs from the established definition—robustness to misspecification of either the propensity model or the outcome model—which is grounded in Neyman-orthogonality of the efficient influence function (EIF). In that sense, the revision added in the manuscript is unprecise "A related line of literature is in one-step estimators, which strongly resemble DR estimators (Kennedy, 2023). The asymptotic properties of these estimators is studied in semiparametric efficient statistics and efficient influence functions."
> The manuscript should either
> (a) adopt accurate terminology (e.g., *annotation- and model-robust given known* $\pi(b)$, or
> (b) ground the paper with a genuine DR result by deriving the EIF and proving orthogonality to all nuisance directions, including that of the annotations.
>
> 2. Variance analysis remains incomplete
>
> The authors' claim that the covariance term in their variance decomposition is "difficult to analyze" is not convincing.
> The term
> $$
> \mathrm{Cov} \left(
> v^{\pi_e}(S),\,
> \mathbb{E}_{A\sim\pi_e(\cdot\mid S)} \left[
> \Bigl(1 -
> \frac{\bar W(A\mid S,A)\,\pi_b(A\mid S)}{\pi_b^+(A\mid S)}
> \Bigr)\epsilon_G(S,A)
> \right]
> \right)
> $$
>
> is tractable under mild and interpretable conditions.
> In particular, if the annotation noise satisfies the mean-zero condition
>
> $$
> \mathbb{E}[\epsilon_G(S,A)\mid S,A]=0,
> $$
>
> the inner expectation vanishes and the covariance term is exactly zero.
> More generally, by the Cauchy–Schwarz and Jensen inequalities,
>
> $$
> |\mathrm{Cov}(\cdot)|
> \le B\,\sqrt{\mathrm{Var} \big[v^{\pi_e}(S)\big]\,
> \mathbb{E} \big[\epsilon_G(S,A)^2\big]}.
> $$
>
> where
>
> $$
> B=\sup_{s,a}\Bigl|1-
> \frac{\bar W(s,a)\pi_b(a\mid s)}{\pi_b^+(a\mid s)}\Bigr|.
> $$
>
> Under standard bounded-weight and overlap assumptions
>
> $$
> |\bar W|\le C, \qquad
> \pi_b^+(a\mid s)\ge c\,\pi_b(a\mid s),
> $$
>
> we have $B\le 1+C/c$, yielding a finite and interpretable bound.
> Hence, the covariance term can either be shown to vanish or be tightly bounded using familiar arguments from semiparametric variance analysis (e.g., Kennedy 2024; van der Laan & Rubin 2006). Declaring it ``difficult to analyze'' without introducing such regularity assumptions is therefore not accurate and weakens the completeness of the theoretical treatment.
>
> 3. Operational diagnostic or sufficient condition
>
> The statement that DM⁺–IS improves variance “when the purple + green terms in Theorem 3 are smaller than …” is circular, as these terms depend on unknown quantities.
>
> 4. Connection to EIF and one-step estimators
>
> The authors in their response note that "the doubly robust estimator is the one-step correction to DM" but again, their definition of DR departs from that classically used in semiparametric efficiency theory. Please rigorously define your statistical target parameter so as to possibly specify an orthogonal moment or pathwise derivative, to then explain which nuisances are held fixed when “double robustness” is claimed.
>
> This is essential for interpreting asymptotic behaviour and situating the work within the semiparametric/DR OPE literature.
>
>
> 5. Quantitative evaluation
>
> The experiments remain largely qualitative.
>
>
> Summary recommendation
>
> The current updated revision leaves theoretical and empirical issues unresolved. Unless the authors deliver a **clear and rigorous update** addressing the points above I would **not be able to recommend acceptance** of this manuscript in its current form.

---

> > ### Author Response · Authors · 2025-11-04
> >
> > We thank the reviewer for their detailed feedback. We address individual concerns below.
> >
> > **Re: Definition and use of “doubly robust”**
> >
> > We thank the reviewer for this comment and wish to clarify our claims. In the current manuscript, we state that our estimators are “inspired by the doubly robust principle” because they share the same structural form as the standard doubly robust estimator. However, we do not claim that our estimators are doubly robust in the classical semiparametric sense.
> >
> > We agree with the reviewer that “doubly robust” has an established meaning. Our estimators do not provide this property, as we assume access to accurate propensity ratios and instead focus on robustness to two different nuisance sources (annotation error and reward model misspecification).
> >
> > To avoid confusion, we will revise the manuscript to (i) remove language suggesting classical double robustness, (ii) clearly state that our goal is to mitigate two nuisance components under known propensities, and (iii) emphasize that EIF-based literature apply only to the standard doubly robust estimator. We will also adopt the terminology suggested by the reviewer (e.g., annotation- and model- robust) to make the distinction explicit. If helpful for clarity, we are willing to adjust the title accordingly to reflect that the estimators are doubly robust–inspired, rather than doubly robust in the traditional sense.
> >
> > **Re: Variance analysis**
> >
> > We appreciate the reviewer’s detailed feedback on the variance derivation. First, we note that in the special case where $\epsilon_G=0$, the reviewer’s observation is correct: the covariance term vanishes, which simplifies the analysis of the variance for both the $DM–IS^+$ and $DM^+–IS^+$ estimators. This is similar to the setting under Assumption 2, for which proofs are provided in Appendices G.2.2 and G.3.3.
> >
> > As the reviewer highlights, there is also a setting in which the annotations are unbiased but exhibit higher variance than the factual samples. This effect can be studied through the variance derivation of $DM–IS^+$ (Appendix G.2.2). In this scenario, the additional variance from counterfactual annotations increases the magnitude of the term in Equation 204 (accounting for a non-zero $\Delta_G$), which in turn raises the overall variance of the estimator. A similar argument applies to $DM^+–IS^+$ under these conditions.
> >
> > We observe that the difficulty in analyzing the variance of $DM–IS^+$ and $DM^+–IS^+$ under imperfect annotations arises primarily from biased annotations (Assumption 3). In this case, the covariance term introduced by the bias has an indeterminate sign, as noted by the reviewer. In contrast, when the annotations are unbiased but have higher variance, the variance derivation is tractable, and we will include this derivation in the final manuscript.
> >
> > **Re: Operational diagnostic or sufficient condition**
> >
> > We thank the reviewer for this comment and agree that our original phrasing was unclear. First, we restate that the DM^+–IS yields lower variance than DM–IS when
> > $$E_{s \sim d_0} E_{a \sim \pi_b} [ (\rho_{s}(a)^2 - \frac{1}{\pi_b(a|s)} ) \Delta_{\hat{R}^+}(s,a)]  + {E_{s \sim d_0} [E_{a \sim \pi_b}[\rho_{s}(a)^2 \varepsilon_{\hat{R}^+}(s,a)^2] - \varepsilon_{\hat{R}^+}^{\pi_e}(s)^2 ]} < E_{s \sim d_0} E_{a \sim \pi_b}[ (\rho_{s}(a)^2 - \frac{1}{\pi_b(a|s)}) \Delta_{\hat{R}}(s,a)]
> > $$
> >
> > The terms on the LHS demonstrate that $DM^+–IS$ improves performance when the annotations reduce reward-model variance more than they introduce additional bias.
> > However, as the reviewer notes, these terms depend on the unknown true reward function and are therefore not directly testable in practice. We will revise the manuscript to emphasize that this criterion is intended as a conceptual characterization of when $DM^+–IS$ is beneficial, not as an estimator selection criterion.
> >
> > In practice, we therefore rely on empirical evaluation rather than this inequality. As shown in Figures 6 and 10 in Appendix A, $DM^+–IS$ consistently achieves lower RMSE across a range of annotation quality.
> >
> > **Re: Connection to EIF and one-step estimators**
> >
> > As discussed above, our estimators are not doubly robust in the classical semiparametric sense, because we do not seek robustness to incorrect IPS ratios. We will revise the manuscript to clarify that the EIF-based theory applies only to the standard doubly robust estimator.
> >
> > **Re: Quantitative Evaluation**
> >
> > We agree that heatmaps emphasize qualitative patterns, and we have taken further steps to address this. Specifically, we have reproduced all heatmap-based results as line plots in the appendix. In particular:
> > Figure 6 and Figure 10 (Appendix A) now present the results corresponding to Figures 2 and 3 in line-plot form, including standard deviation error bars. We are also happy to provide numbers in a table if the reviewer recommends that.

---

> > > ### Comment · Action_Editor_3x6Y · 2025-12-15
> > > **please update manuscript**
> > >
> > > Dear authors,
> > >
> > > Please update your manuscript with the proposed changes (to reviewer YL5w above but also reviewer 4VFJ) in order to finalize the review process.
> > >
> > > -AE

---

> > > > ### Author Response · Authors · 2025-12-22
> > > >
> > > > Dear action editor and reviewers,
> > > >
> > > > We thank you for your support. We have updated the manuscript with the following (all updates are in the text color Brown):
> > > >
> > > > - A variance derivation for both the $DM-IS^+$ and $DM^+-IS^+$ estimators under imperfect annotations (Propositions 16 and 19).
> > > > - We have improved the writing of Section 2.3 to avoid having a reader reference the original Tang & Wiens work. In particular, we clarify how the weighting procedure can affect the estimators.
> > > > - We also clarify our theoretical results in Section 4 and justify why we choose to compare estimators empirically rather than purely from a theoretical perspective.
> > > >
> > > > We are happy to make any other adjustments to our manuscript, and once again we thank all reviewers and our action editor.
> > > >
> > > > Best,
> > > >
> > > > Authors

---

### Review · Reviewer_TPKW · 2025-09-25

**Summary Of Contributions:**

This paper studied off-policy evaluation in contextual bandit. The authors mentioned potential application to high-risk scenarios like healthcare. Because data is scarce or not available from these scenarios, OPE is required to estimate the performance of a new policy given data collected by some other behavior policies. The authors introduced limited data coverage and imperfect counterfactual annotations as a key challenge to OPE in the scenarios of interest. The authors combined DM and IS to propose three new estimators to handle counterfactual annotations. Theoretical analysis and experiments from healthcare domains were provided.

**Audience:**

Yes

**Audience Explanation:**

Again I cannot fully evaluate correctness of the technical part but I am excited to see that the authors tried out their proposed methods on several healthcare applications. It would be nicer if the authors could address my questions about technical assumptions, but that does not outweigh the contribution of the paper imo.

**Broader Impact Concerns:**

I don't find this paper have any societal impact at this point as the paper deals with technical questions.

**Claims And Evidence:**

Yes

**Claims Explanation:**

Disclaimer: I am not an expert in OPE or bandit, and I cannot evaluate correctness of the paper. But I do feel excited that the authors proposed new methods on healthcare applications like Sepsis and MIMIC.

From a technical side, I am most interested in the assumptions the authors made. For example, the authors assumed the IPS ratio was known following prior works. But I wonder how your analysis would change given an imperfect IPS ratio? Because data coverage is a key challenge and that naturally brings biased IPS ratio, it is interesting to examine how that would affect the current analysis.

For annotations, the authors listed perfect, biased and noisy annotations. But I wonder whether the current analysis works for more practical scenarios like systematic bias? I would say it is more natural that an expert shows systematic bias e.g. towards only one direction.

**Requested Changes:**

I would suggest the authors to include new insights on systematic bias and imperfect IPS. If there are no new insights yet, the current draft could benefit from a thorough discussion on these points.

---

> ### Author Response · Authors · 2025-10-29
>
> We thank the reviewer for their response. We are thrilled to hear that the reviewer finds that our work is valuable to the TMLR audience. We address specific comments in individual points below.
>
> **Re: Imperfect IPS Ratios**
>
> As we mention in Section 3, our goal is to define an estimator that is “doubly robust” to two sources of error, namely the error of the annotation and the error of the reward model. As such, we do not account for imperfect IPS ratios and assume that the estimates of IPS ratios are fairly accurate.
>
> In the case that the IPS ratio is inaccurate, all proposed estimators will be biased. In particular, [1] uses the term $\delta(s, a, h) = 1- \pi_b(a|s,h)/\hat{\pi_b}(a|s,h)$ to describe the estimation error of the learned behavior policy $\hat{\pi_b}$. This estimation error will propagate through the bias and variance reductions introduced in our work.
>
> To further illustrate this, we report the performance of $DM^+-IS$ with varying degrees of incorrectly estimated behavior policies (Figure 11 in the updated manuscript). We note in this plot that when the degree of estimation error increases, the estimator achieves higher RMSE, indicating that it is biased. We leave further evaluation of this setting to future work.
>
> **Re:Practical Scenarios like systematic bias**
>
> We believe that scenarios like systematic bias can be represented in our framework. In particular, we define the annotation bias as $\epsilon_G(s,a)$, which allows per-sample bias. If the bias is shared across samples, as in the case with systematic bias, we can specify $\epsilon_G(s,a)$ to be a constant and shared across all or a subset of samples.
>
> [1] Dudík, Miroslav et al. “Doubly Robust Policy Evaluation and Learning.” International Conference on Machine Learning (2011).

---

### Review · Reviewer_4VFJ · 2025-10-21

**Summary Of Contributions:**

This paper addresses the problem of off-policy evaluation of a target policy within a contextual bandit setting, using data collected by a separate behavior policy. The authors' central contribution is the development of estimators for the target policy's expected reward that leverage annotated counterfactual data—samples whose rewards were not observed but inferred by an expert. Building on prior work, the authors introduce three novel estimators that rely on combinations of two main techniques: importance sampling and estimating a reward model using the counterfactual data. They theoretically characterize the bias and variance of these estimators and complement their theoretical results with experiments on synthetic settings and a real-world clinical dataset.

**Additional Comments:**

Due to bad coordination from my side, I have significantly delayed submitting my review for this manuscript. I apologize to the authors for any inconvenience this may have caused.

**Audience:**

Yes

**Audience Explanation:**

The paper should be of interest to TMLR's audience. Off-policy evaluation is a central problem in reinforcement learning that is attracting significant interest, and the paper’s focus on the use of counterfactual (and potentially biased) data is a good addition to the literature. Researchers working on contextual bandits and causal inference may find the proposed estimators and their theoretical analysis valuable.

**Broader Impact Concerns:**

This work does not have any direct ethical implications.

**Claims And Evidence:**

Yes

**Claims Explanation:**

The proposed estimators effectively extend and build upon prior work. The authors provide rigorous theoretical results characterizing the bias and variance for each of the three estimators they introduce, and these results appear solid. The experimental evaluation is extensive, using three synthetic environments and a real-world clinical dataset. The empirical results align closely with the theory. Most importantly, one of the three estimators introduced by the authors outperforms all competitive baselines on the real dataset in terms of the error in estimating the expected reward, while in the worst case it matches the best performance in the synthetic data. That said, I believe the paper sufficiently supports its main claims. While the evidence is convincing, there are a few aspects where there is room for improvement, which I elaborate on under “Requested Changes”.

**Requested Changes:**

The following (more major) points relate mainly to the completeness of the theoretical and experimental results:
1. While all three proposed estimators are unbiased under perfect counterfactual annotations, there doesn't seem to be a characterization of their variance. I would expect that the use of annotations provides a variance reduction compared to the estimators that do not use the annotated counterfactual data. This result may be straightforward to show but I think it would add intuition on why using the counterfactual annotations is useful in the first place.
1. The authors conclude Section 3 by noting that imperfect annotations are expected to increase the variance of all three estimators. Connecting to my previous point, this raises a question that doesn't seem to be properly addressed in the paper: Under what conditions (e.g., in relation to sample size or annotation quality) does using imperfect counterfactual data still yield an overall benefit (i.e., decreased variance) compared to ignoring them entirely? A more formal characterization of this trade-off would make the paper much stronger.
1. Based on Figure 2, the authors conclude that the RMSE is not (significantly) affected by noisy counterfactual annotations. However, it is not specified what sample size the authors use in that figure---I presume something very large because the figure looks very smooth. That said, I am not sure if one can make conclusions about the effects of the variance in the annotations. To draw robust conclusions, I believe one would need to do the same analysis as in Figure 2 across multiple sample sizes and also consider multiple values for the variance of the underlying reward distribution.

The following points are minor and relate mainly to the clarity and presentation of the paper:
1. The context space $\mathcal{S}$ is defined as discrete, but the paper does not discuss why this assumption is made and how it affects the theoretical results. Also, the contexts used in the experiments appear to be continuous (vector-valued).
1. Assumption 4 is referenced twice in Section 2.1 but is only presented much later in the text. Maybe it is worth introducing it in 2.1 so that it is easier for readers to follow the flow.
1. Are the DR estimator in Equation 1 and the DM-IS estimator discussed in the experiments the same? If so, it would be best to use a single consistent abbreviation throughout the paper.
1. It would be helpful to clarify in Equations 2 to 4 exactly how the terms depend on the factual ($N$ points) and counterfactual ($M$ points) data, as the current presentation suggests that all estimators depend only on $N$. Furthermore, on Page 5, I think the authors should clarify how the reward model estimation datasets ($D_{\hat{R}}$ and $D_{\hat{R}^+}$) relate to the factual ($D$) and counterfactual ($D^+$) datasets, as this distinction is rather confusing.
1. Assumptions 1 and 2 are in direct contradiction. While subsequent theoretical results do not assume both simultaneously, stating them sequentially in the main text is somewhat confusing. It would be clearer to state these assumptions directly within the theorems where they are applied, rather than listing them together out of context.
1. I found it somewhat strange that the reader is pointed to the Appendix right as the theoretical results begin. I would encourage the authors to bring their first results (i.e., the unbiasedness of the three estimators under perfect counterfactual annotations) into the main body of the paper.
1. In the experimental setup of Section 4.1.2, is the distinction between "renal" and "non-renal" patients used for defining the target and behavior policies captured by the context? It would be helpful to confirm that the two policies still maintain common support.
1. In several plots the notation for the $DM^+ - IS$ estimator is inconsistent (written as $DM+-IS$) and should be fixed.
1. In Figure 3, the $DM$, $DM-IS$, and $DM^+-IS$ estimators appear to perform equally well. It would be helpful if the authors could provide some intuition for this observation.
1. The final result of the paper, discussed at the end of Page 9, points the reader to the Appendix for the corresponding figure. It would be better to bring that figure to the main body.

---

> ### Author Response · Authors · 2025-10-29
>
> We thank the reviewer for their suggestions and are thrilled that our work is suitable for the TMLR audience. We respond to individual concerns below.
>
> **Re: Characterization of variance**
>
> We include variance derivations for all three estimators under perfect annotations. We only perform the variance derivation under imperfect annotations for one estimator. We analyze the variance for the other two estimators empirically because their derivations are not directly comparable to that of $DM^+-IS$.
>
> In particular, the proof of the variance of $DM-IS^{+}$ and $DM^{+}-IS^{+}$ under imperfect annotations starts similar to the corresponding proof for $DM^+-IS$ (Appendix G.1.2) until Equation 98. However, the second term (variance of the expectation of the estimator) does not simplify to $v(\pi_e)$ and instead is $$V[v^{\pi_e}(s) + E_{s_i \sim d_0}[E_{a \sim \pi_e(s_i)}[(1- \frac{\bar{W}(a|s_i,a) \pi_b(a|s_i)}{\pi_b^+(a|s_i)}) \epsilon_G(s_i, a)]]]$$.
>
> This term simplifies to $V[v^{\pi_e}(s_i)] + V[E_{s_i \sim d_0}[E_{a \sim \pi_e(s_i)}[(1- \frac{\bar{W}(a|s_i,a) \pi_b(a|s_i)}{\pi_b^+(a|s_i)}) \epsilon_G(s_i, a)]]]] + Cov(v(s_i), E_{s_i \sim d_0}[E_{a \sim \pi_e(s_i)}[(1- \frac{\bar{W}(a|s_i,a) \pi_b(a|s_i)}{\pi_b^+(a|s_i)}) \epsilon_G(s_i, a)]])$.
>
> The last covariance term is difficult to analyze; without additional assumptions on the weights or annotation quality, we do not know its sign or magnitude. However, we will include the variance derivations in the manuscript for completeness.
>
> **Re: Conditions for Improvement with Imperfect Annotations**
>
> The formal conditions for when $DM^+-IS$ performs better than $DM-IS$ with imperfect annotations can be found by analyzing Theorem 3. In particular, when the sum of the the purple and the green terms in Theorem 3 are less than $$E_{s \sim d_0} E_{a \sim \pi_b} [ (\rho_{s}(a)^2 - \frac{1}{\pi_b(a|s)}) \Delta_{\hat{R}}(s,a)] $$, the variance of $DM^+-IS$ improves.
>
> However, we cannot make more claims without making more assumptions on the systematic nature of bias and variance across the annotations. While we can provide a formal characterization, it is not informative without more assumptions on the annotation quality.
>
> **Re: Figure 2**
>
> The results reported in Figure 2 use N=400 factual samples and M=400 counterfactual annotations. We appreciate the suggestion to do the same analysis across multiple sample sizes. We perform a similar analysis in Figure 4b, and find that across various numbers of counterfactual annotations that the RMSE of $DM^+-IS$ improves in comparison to baselines.
>
> **Re: Discrete context space**
>
> The discrete context space is used for simplicity in our theoretical results when we analyze the DM component of our estimators, in line with prior work. In particular, the DM component is calculated using a sample average, which relies on a discrete context space. However, in our empirical results, we investigate more complex continuous but finite context spaces. Our theoretical results can also be used to describe these settings in a manner similar to prior work [1].
>
> **Re: Renal vs non-renal captured by context?**
>
> We define the renal and non-renal sub-cohorts based on the known comorbidity status of the patients. There is coverage between the cohorts in terms of the action space (Figure 14).
>
> **Clarity on where the annotations are used and what is the difference between $D_{R}, D_{R+}$**
>
> In equations 2-4, we use annotations in three distinct ways. The annotations in Equation 2 and 4 are used within the reward model, and the annotations in Equation 3 and 4 are used within $\sum_a \in A$, where we aggregate all annotations for the i’th factual sample. We assume that $D_{R}, D_{R+}$ are identically distributed.
>
> **Formatting Concerns**
>
> We note that Figure 9 (Figure 12 in the updated manuscript) contains a similar analysis to that presented in Figure 4b, just represented differently. As such, we will leave it in the Appendix. We will also refer to the DR estimator always as DR rather than using DM-IS, since these are the same estimator. We will also separate the introduction of the assumptions and add a sentence saying that Assumptions 1 and 2 are never invoked simultaneously. We are also happy to move the theoretical results referenced in Section 3 to the main text.
>
> **DM/DM-IS/$DM^+-IS$ performance is equal in Figure 2**
>
> The performance of DM, DM-IS, and $DM^+-IS$ is comparable for the HeartSteps setting. This is in part due to the fact that the reward function is simple and does not benefit substantially from additional counterfactual annotations. Furthermore, if the reward model is accurate, the correction term in the DR estimator does not contribute much. We note this phenomenon is only observed in the HeartSteps setting. In other settings,  $DM^+-IS$ outperforms the standard DR estimator and the DM estimator.
>
> [1] Jiang, Nan and Tengyang Xie. “Offline Reinforcement Learning in Large State Spaces: Algorithms and Guarantees.” (2025).

---

### Decision · Action_Editor_3x6Y · 2026-01-13

**Recommendation:** Reject

**Additional Comments:**

Here is the final comment provided by reviewer YL5w after your latest manuscript update, for your reference:
>The revised version addresses some of the previous theoretical and empirical concerns unaddressed (derived variance of DM-IS+ and DM+-IS+ under imperfect annotations in Proposition 16 and Proposition 19) but the comparison in the updated text remains vacuous and uninformative, the authors did not try to discuss the terms involved with regards to simplified cases to build intuition and intepretation.
>
>The writing overall has not changed, the section 5 for example has not been updated at all despite my recommendations. The references to DR and semiparametric efficiency theory is only vaguely mentioned but the paper is not well positioned w.r.t to that literature.
>
>Overall, I still believe this paper is below acceptance threshold and that the scientific message is approximative and not well grounded.

If you do aim for a major revision, please take this into account by making the theoretical analysis more informative, or by focusing on experiments with more thorough empirical studies.

**Audience:**

Yes

**Audience Explanation:**

The reviewers found the setting and the proposed estimators interesting for the TMLR audience.

**Claims And Evidence:**

No

**Claims Explanation:**

Two of the reviewers expressed concerns regarding the theoretical analysis of the proposed estimators, in particular the variance terms, which are not fully analyzed, thus failing to properly show the benefits of the proposed methods in certain situations. This is concerning as one of the primary goals of the paper is to obtain a method with "better statistical guarantees". The experimental results also would benefit from more clarity with more details to clarify the relative performance of different methods.

**Resubmission Of Major Revision:**

The authors may consider submitting a major revision at a later time.